# Causal discovery for time series with endogenous context variables

Oana-Iuliana Popescu [* 1]   Wiebke Günther [* 2]   Martin Rabel [1]   Jakob Runge [1]

## Abstract

Many real-world systems exhibit both context- and time-dependent causal dynamics, where the dynamical system state also influences its context. For instance, soil moisture is driven by precipitation, yet also provides the context for heat-flux realization. We capture such dynamics in Structural Causal Models (SCMs) by introducing endogenous and time-dependent discrete context variables, also allowing for lagged dependencies with the system variables. While context variables are discrete, they may also be proxies of continuous variables. The enabling assumptions for causal discovery of our model are either persistence of the context or sparsity of the context–system dependencies. We design two novel PCMCI-based algorithms for causal discovery with endogenous context variables for time series and prove their soundness. A systematic evaluation on synthetic benchmarks and an application to a real-world example from climate science demonstrate their effectiveness and applicability.

In many dynamical real-world systems, causal relationships change over time, a phenomenon we call *context shifts*. Such shifts may arise from internal system states (endogenous contexts), not only from external forcing (exogenous contexts). Understanding these shifts is crucial for prediction, yet most causal discovery (CD) methods with context shifts assume exogeneity. This assumption fails in many applications. In climate science, for example, soil moisture is driven by precipitation but also modulates how surface temperatures respond: moist soils favor evaporative cooling, whereas dry soils increase sensible heat flux and amplify heat extremes (Seneviratne et al., 2010). Treating soil moisture as an exogenous switch cannot capture this feedback,

and Regime-PCMCI (Runge, 2020), which assumes exogenous contexts, has struggled in this setting (Popescu et al., 2024). Analogous endogenous regimes arise in financial markets, where volatility regimes emerge from trading dynamics and alter the effects of economic indicators (Yao et al., 2025), and in neuroscience, where brain states such as attention emerge from neural activity while shaping sensory processing (Cohen & Maunsell, 2009). Recently, Günther et al. (2024) adapted the PC algorithm to handle discrete endogenous context variables for i.i.d. data. Extending this idea to time series introduces fundamental challenges, starting with how to model context shifts. We consider observed, discrete context variables that modulate structural dependencies over time, following prior work (Mooij et al., 2020), but, unlike existing methods, explicitly allowing contexts to be endogenous. Unlike in the i.i.d. setting, context switches in time series may become effective only after a delay[1], e.g., vegetation responses after a transition to dry soil conditions.

This flexibility creates algorithmic challenges. Because the context at a given time is defined by context values over a time window up to some maximal lag, the number of possible contexts grows exponentially, rendering naive context-specific CD computationally infeasible and statistically ill-posed. We address this by exploiting two system properties: temporal stability (context persistence) and structural simplicity (sparsity). A further challenge is selection bias: conditioning on contexts as if they were exogenous can bias estimates, especially when the context is a collider. Furthermore, naive masking is statistically inefficient and obscures causal effects mediated by the context itself since it removes all variation in the context variable. Following Günther et al. (2024), we solve this via an adaptive conditional independence testing framework combining pooled and context-specific data, improving sample efficiency and allowing the relevant context lag to be learned from data. These benefits hold even for exogenous contexts.

Our contributions are: (1) identifying context persistence and sparsity as testable properties enabling tractable context-specific CD for time series motivated by real-world examples, (2) proposing two PCMCI[+]-based algorithms (Runge, 2020) that recover context-specific causal graphs without

---
[*]Equal contribution  [1]University of Potsdam, Department of Computer Science, Potsdam, Germany [2]Technische Universität Berlin, Institute of Computer Engineering and Microelectronics, Berlin, Germany. Correspondence to: Oana-Iuliana Popescu <oana-iuliana.popescu@uni-potsdam.de>.

*Proceedings of the 43$^{rd}$ International Conference on Machine Learning*, Seoul, South Korea. PMLR 306, 2026. Copyright 2026 by the author(s).

---
[1]The true system state is instantaneous but unobserved; lagged indicators may serve as near-deterministic proxies, see Sec. 3.1.

selection bias and exploit these properties, (3) providing theoretical guarantees under suitable assumptions, and (4) systematically evaluating the methods on synthetic benchmarks and real-world applications, showing when and why they outperform approaches assuming exogenous contexts.

## 1. Related Work

**CD for non-stationary time-series** Three main approaches dominate CD for non-stationary time-series: (i) Mixture-based models, where each sample comes from one of finitely many SCMs (Varambally et al., 2024; Saggioro et al., 2020), typically using a (latent) discrete regime variable, usually exogenous and contemporaneous, and inferred via variational objectives (Mameche et al., 2025; Rodas et al., 2021; 2023), invariant causal prediction (Christiansen & Peters, 2020), or expectation maximization (Rodas et al., 2024; Rahmani & Frossard, 2023). While expressive, such methods involve many hyperparameters which require careful tuning. (ii) Context-based approaches with explicit context variables interacting with system variables (Günther et al., 2023; Mameche et al., 2025; Sadeghi et al., 2024). These methods often assume structure on the context, e.g., (smooth) time trends (Huang et al., 2020; Sadeghi et al., 2024), periodicity (Gao et al., 2023), or (piecewise) constant/persistent regimes (Günther et al., 2023; Mameche et al., 2025; Saggioro et al., 2020), to recover hidden contexts and either context-specific graphs (Saggioro et al., 2020; Gao et al., 2023) or changing modules/union graphs (Sadeghi et al., 2024; Günther et al., 2023). Most prior work jointly infers contexts and graphs under exogenous contexts (see Günther et al. (2024); Rabel et al. (2024)). (iii) Graph-difference methods directly learn changes between two causal graphs (Malik et al., 2024; Assaad, 2025).

**Endogenous context variables** Context variables have been studied under different names, including F-variables (Pearl, 2009), S-variables (Bareinboim & Pearl, 2013), and the JCI-framework (Mooij et al., 2020). These approaches typically treat the context as exogenous and known. For context-specific CD with i.i.d. data, Günther et al. (2024) recently addressed the topic of endogenous context variables. The closest work to ours is Rodas et al. (2024), which uses hidden switching Markov models to infer latent (possibly endogenous) system states. As an optimization-based approach, it suffers from the typical drawbacks of such methods related to hyperparameter tuning.

**Positioning our work** We are the first to systematically study endogenous context in time-series CD. Our approach differs from prior work in the following key aspects. We explicitly model context as endogenous time-series with lagged dependencies. Moreover, as a constraint-based method, our approach requires fewer hyperparameters. Moreover, due to the adaptive testing approach, it requires

fewer CITs than other constraint-based methods, thus being more sample efficient. This efficiency comes at the cost of relying on the persistence and sparsity properties, introduced below, both empirically testable and common in practice. We further relate our approach to the other context-specific variant of PCMCI$^+$, J-PCMCI$^+$, and show how it can be combined and unified with our method in App. B.

## 2. Preliminaries

We assume the data stems from a structural causal model (SCM) (Pearl, 2009). In the time series setting, each time step is associated with an **SCM** $M_t$ which is a triple $M_t = (X, U, \mathcal{F})$, where the observed endogenous variables $X$ are determined by the **structural equations** $\mathcal{F}$ given the exogenous variables $U$ at each time step $t$. We denote the observable time-discrete **endogenous variables** at time $t$ by $X_t = \{X_t^i | i \in I\}$, and the **exogenous noises** by $U_t := \{\eta_t^i \mid i \in I\}$ with $X_t^i \in \mathcal{X}_i$ and $\eta_t^i \in \mathcal{N}_i$ for a finite index set $I$. For a time window $[t - \tau_{\max}, t]$ with $\tau_{\max}$ the maximal lag, we write $V$, and $U$, respectively, to denote the collection of variables within $[t - \tau_{\max}, t]$. Throughout this paper, we restrict the focus to uniquely solvable SCMs. Each structural equation has the form $f_i : \mathcal{X}_{\mathrm{Pa}(X_t^i)} \times \mathcal{N}_i \to \mathcal{X}_i$, where $X_t^i$ depends non-trivially on its parents $\{\mathrm{Pa}(X_t^i) \subset X | i \in I\}$. These relationships can be represented by a **causal graph**, with an edge directed into $X_t^i$ from each of its parents. While causal stationarity, i.e., the causal parents are time-invariant, is typically assumed, we allow non-stationarity by assuming that the data stems from multiple SCMs referred to as **contexts**. These contexts are indexed by discrete **context indicators** as in Günther et al. (2024), formally introduced below. Finally, we define the notion of **(hard) intervention** $\mathcal{F}_{\mathrm{do}(A=g)}$ on a subset $A \subset X$, which replaces the respective structural equations by a constant, yielding the **intervened model** $M_{\mathrm{do}(A=g)} := (X', U, \mathcal{F}_{\mathrm{do}(A=g)})$.

## 3. Challenges

### 3.1. Modeling context variables in time-series

We model the context variable as an observed, discrete time series that both influences and is influenced by system variables, potentially with time lags. Unlike in the i.i.d. case, the context variable cannot exist outside of time. Allowing lagged context effects is a deliberate design choice motivated by realistic system dynamics; App. E discusses an alternative restricted to contemporaneous effects.

*Example* 3.1 (Dam management). Consider rainfall upstream and the water level downstream of a dam. When the dam is open, the water upstream needs two time steps to reach the water level measurement station downstream. When the dam is closed, as the water arrives at time $t - 1$,

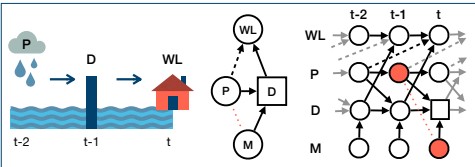

*Figure 1.* Causal system where precipitation (P) affects water level (WL), modulated by dam status (D) which is driven by P and other factors such as scheduled maintenance (M). Solid edges are always active; dashed edges vanish when the dam is closed. Black edges lie within the time window, gray edges connect to variables outside it. Red dotted edges indicate selection bias from conditioning on square nodes. Further details in Ex. 3.1.

this causal relationship is not observable. The dam state (open or closed) thus naturally defines two distinct contexts (see Figure 1). Critically, the dam state is endogenous: dam management is done based on precipitation at the previous time step. Furthermore, the dam state is also influenced by unrelated factors such as maintenance work. If we condition on "dam closed" at time $t$ (naive masking), we select samples where $D_t = 0$, but these samples have different values for $D_{t-1}$, leading to lag inconsistency and selection bias; modeling the context as a time series with explicit lagged dependencies yields the correct, lag-consistent datasets.

**Three requirements for the context variable in the time-series case** This example illustrates three features that the context should fulfill for a realistic modeling of real-world problems. We thus define the three key requirements for context variables in time-series: (1) *Contexts can be endogenous*: Dam state is determined by past precipitation, not externally imposed. (2) *Context effects are lagged*: Dam state at time $t - 1$ determines causal relationships at time $t$. (3) *Context windows span multiple lags*: To fully characterize the system, we must consider dam state over the entire causal window $[t - \tau_{\max}, t]$, not just a single time point, leading to exponentially many possible context sequences.

**From continuous states to discrete proxies** We typically observe *proxy variables*, which approximate the true latent and continuous context state that modulates the causal mechanisms. In Example 3.1, the true context is the current operational configuration of the dam, but we observe it through the binary indicator "dam open/closed". Its effect becomes visible at the measurement station after one time step. This proxy has two key properties: (1) it is *discretized*, and (2) it is *lagged*, i.e., the value at $t - 1$ indicates the context governing relationships at $t$. Explicitly modeling context states as continuous would introduce interpretability issues, as it is unclear which values belong to which regime. We thus work directly with observed discrete proxy variables, which is common in practice: climate scientists discretize soil moisture into "dry/normal/moist" regimes; financial analysts categorize market states as "low/high volatility"; medical

researchers bin biomarkers into "normal/elevated" ranges. Definition 3.1 of a context variable below addresses all three requirements: it allows for endogenous context variables which can be time-lagged, as well as to treat $C_{t-\tau_{\max}}, \ldots, C_t$ as multiple context indicators for the system. We assume known context variables, from domain expertise or change-point detection. This is a standard assumption in literature, for example, JCI (Mooij et al., 2020), CD-NOD (Huang et al., 2020), and Regime-PCMCI (Saggioro et al., 2020).

**Definition 3.1** (Context variable). A *context variable* $C$ is a discrete-valued time series. For some maximal time lag $\tau_{\max} \geq 0$, the values $(C_{t-\tau_{\max}}, \ldots, C_t)$ determine the context, i.e., the dataset to which the observation of all time-series $\mathbf{X}$ at time $t$ belongs.

### 3.2. Algorithmic challenges

#### 3.2.1. CONTEXT EXPLOSION

A direct consequence of Def. 3.1 is that the number of context values grows exponentially in the length of the time window to $K^{\tau_{\max}+1}$, where $K$ is the number of possible context values at each time step, because, to uniquely specify a context given multiple indicators, a value must be fixed for each indicator. We name this problem "context explosion", and note that we are not the first to recognize this problem: Tajeuna et al. (2021) and Rodas et al. (2024) discuss this problem too in the context of time-series. Tajeuna et al. (2021) have a conceptually similar setup, but do not tackle the problem of possibly endogenous context variables directly, while, as noted earlier, Rodas et al. (2024) propose an approach that is fundamentally different from ours. The context explosion problem poses evident challenges both from a statistical perspective and for interpretability. To simplify the problem of context explosion, we can make use of two properties which a real-world dynamical system can possess, namely persistence and sparsity of the contexts, two realistic, and testable properties. As we will describe in detail in Sec. 5, these two assumptions are complementary, and must not hold both at the same time.

**Persistent contexts** The assumption of a persistent time-series model is often encountered in literature (Saggioro et al., 2020) and holds in many real-world cases (Hoffmann et al., 2021; Ives & Carpenter, 2007). In a persistent time-series model, $C$ varies slowly, so for most $t$, $C_{t-\tau_{\max}} = \ldots = C_t$ holds. In Example 3.1, this corresponds to the dam remaining open or closed for extended periods of time longer than $\tau_{\max}$. In such cases, we can reduce the complexity of the problem by focusing on the subset of context values $\mathcal{C}_{\text{common}} \subset \mathcal{C}$ where $C_{t-\tau_{\max}} = \ldots = C_t = c$ retains most of the per-context data. The output no longer captures *all* possible context-specific knowledge, but still conveys information about the most frequent contexts (often those of greatest interest). We define persistence as below,

and note that, given $(w = \tau_{\max} + 1, \delta)$-persistent data, we can effectively use $\delta N$ samples, a substantial improvement if $\delta \gg K^{-\tau_{\max}-1}$. Our theoretical results demonstrate how the method benefits from large $\delta$.

**Property 1** $((w, \delta)$-Persistence$)$**.** *We assume the context variable* $(C_t)_{t=1}^{T}$ *to have a* $(w, \delta)$**-persistent** *value* $c \in \{1, \ldots, K\}$ *where $K$ is the number of possible contexts, i.e. for a window length $w > 0$ and a $\delta > 0$ it holds* $\mathbb{P}(C_t = C_{t+1} = \ldots = C_{t+w-1} = c) > \delta.$

**Testability** Persistence can be tested by counting the length of persistent windows, i.e., for each $t$, count the number of time steps for which $C_t = C_{t+1} = \ldots$, and verify that the smallest window length $> w$. Alternatively, the number of windows for which $C_t = C_{t+1} = \cdots = C_{t+w-1}$ can be compared to the maximum number of possible windows, for details see App. C and App. K.

**Sparse contexts** A second property that allows us to restrict the search space to certain context-specific graphs of interest is that of *sparse context-dependence*. Here, sparseness refers to a few mechanisms changing or context dependence being driven by a single or a few context parents. Notably, each variable with changing parent set might have a different context parent. Sparsity is a common assumption in causal inference. For instance, Invariant Causal Prediction (Peters et al., 2015) restricts the set of causal parents to reduce computational demands. Our flavor of sparsity ensures that the parent set of any variable does not depend on specific sequences or combinations of context values. Instead, changes in parent sets are triggered by the value of a single context variable, not by switching patterns or joint configurations of multiple contexts.[2] The sparsity property enables the construction of context summary graphs, analogous to time series summary graphs, by annotating each context-dependent link in the union graph with its (single) context parent and value. In the context setting, sparsity can also be induced by aggregating multiple context variables.

**Property 2** (Sparse context dependence)**.** *Each system variable has at most one context variable at a specific (possibly unknown) lag as a parent.*

**Testability** Sparsity is implicitly tested during the first phase of our algorithm, as, in this phase, the algorithm searches for context-system links and naturally reveals the cases in which a variable has multiple context parents, for details see App. C and App. K.

---

[2]It is possible to view the combination of multiple contexts as one context variable. However, such a reformulation has further implications due using a deterministic function to combine the multiple context variables. We leave a thorough discussion of this problem to future work.

### 3.2.2. SELECTION BIAS AND TIME DEPENDENCE

Another algorithmic challenge in the time series case is selection bias due to endogenous context variables. As discussed in Günther et al. (2024) in the i.i.d. case, selection bias occurs whenever the context variable has multiple parents. In the non-i.i.d. case, time-lagged context-system relationships resolve parts of the selection bias issue. With PCMCI (Runge et al., 2019), the links between two lagged variables $X_{t-1}$ and $Y_{t-2}$ never get tested, but are inferred using stationarity arguments, therefore, selection bias cannot arise between two lagged variables even if $X_{t-1}$ and $Y_{t-2}$ are (union) parents of $C_t$. However, as seen in Example 3.1 selection bias can occur if there is at least one contemporaneous parent of the context variable. In the situation where system variables cause context variables at some lag $\tau$, $X_t \perp\!\!\!\perp Y_{t-1} | C_{t+\tau}$ is not tested since data is not selected at a future time lag.

Günther et al. (2024) solve the problem of selection bias due to endogenous context using an adaptive testing strategy which automatically selects the correct dataset to test with depending on if a context indicator is part of a valid separating set, an approach that we also adapt to the time-series case and will introduce in more detail below. Besides solving the bias problem, this approach yields improved statistical (sample) efficiency compared to naive masking, since it leverages information across contexts rather than restricting analysis to per-context subsamples. The impact on the total number of conditional independence tests (CITs) and on runtime, however, is more nuanced and only under oracle-CIT is the number of tests reduced by combining union and context-specific tests. More details on this are provided in App. E.1. Naive masking also poses problems if the context variable acts as a lagged confounder. If $C_{t-1}$ is a parent of $X_t$ and $Y_{t-1}$, conditioning on the context $C_t = c$ introduces a spurious link that cannot be resolved when considering data with $C_t = c$, i.e. generally $X_t \not\perp\!\!\!\perp Y_{t-1} | C_t = c$. Explicitly including the context variable across its relevant lags in the conditioning sets removes this problem.

## 4. Graphical objects

Our discovery goal is to characterize how causal dependencies vary across contexts while solving the modeling and algorithmic challenges described in Sec. 3. We now introduce a set of graphical objects necessary to formulate this problem. Our primary objects of interest are the context-specific graphs, which describe the causal structure of the system at time $t$ for a particular value of the context. Let $C$ denote one or several context variables, and let $C^{\leq} := (C_{t-\tau_{\max}}, C_{t-\tau_{\max}+1}, \ldots, C_t)$ denote the sequence of context variables over the relevant time window. By fixing $C^{\leq}$ to a value $c^{\leq}$ using a do-intervention similar to Günther et al. (2024) with the available observational

support, we obtain a modified system in which the causal mechanisms are restricted to the ones active in context $c^{\leq}$. The corresponding graph $G_{C^{\leq}=c^{\leq}}^{\mathrm{descr}}$ is a window graph in the sense of Assaad et al. (2022), and represents the set of dependencies among the system variables at time $t$ under that context.

**Definition 4.1** (Context window graph)**.** For an SCM $M$ at time $t$, we define the context window graph as follows, and refer to App. F for further details:

$$\bar{G}_{C^{\leq}=c^{\leq}}^{\mathrm{descr}}[M] := G[\mathcal{F}_{\mathrm{do}(C^{\leq}=c^{\leq})}^{M}, P_M(X|C^{\leq}=c^{\leq})],$$

Context window graphs capture the observable causal structure in each context. From data, we can only discover edges involving at least one variable in $X_t$, corresponding to the SCM at time $t$. In the non-stationary setting, it is generally not possible to infer the full time causal graph from these window graphs. We denote the set of context parents of $X_t^i$ as $\mathrm{Pa}^C(X_t^i)$. Since these are always parents of $X_t^i$ in the union graph, we ignore this in the notation. Similarly, we denote the set of all lagged parents of $X_t^i$ in the union graph by $\mathrm{Pa}_t^{\mathrm{union},-}(X_t^i)$. As the number of possible contexts grows exponentially (Sec. 3.2.1), we reduce the search space using persistence, and focus on constant context window graphs, short $c$-context window graphs. These graphs summarize the causal relations that hold when the system resides in a stable regime indexed by context $c \in \{1, \ldots, K\}$.

**Definition 4.2** ($c$-context window graph)**.** We define the $c$-context window graph with constant intervention $c\mathbb{1}$, where $c\mathbb{1}$ is the vector with all elements equal to $c$, as follows:

$$\bar{G}_{C=c}^{\mathrm{descr}} := \bar{G}_{C^{\leq}=c\mathbb{1}}^{\mathrm{descr}}$$

The *union graph* aggregates all context window graphs by including an edge if it appears in at least one of the contexts, and thus encodes the full set of potential dependencies in the system. Finally, we note that some contexts might lack support in the observational distribution (see (Günther et al., 2024) and App. F for details). Thus, to prove our theoretical claims, further graphical notions are necessary. In our work, we assume that these different context-specific graphical objects all coincide (see App. G).

## 5. Time-series causal discovery with endogenous contexts

Having formalized the challenges and graphical objects, we now propose our algorithmic solution: an adaptation of the constraint-based time series causal discovery method PCMCI$^+$ (Runge, 2020) to accommodate endogenous context variables through an adaptive testing scheme inspired by Günther et al. (2024). We first sketch the overall idea. As detailed below, PCMCI$^+$ proceeds in two phases: a skeleton-discovery phase that identifies lagged parents, and a momentary conditional independence (MCI) phase that refines and orients the remaining edges. We leave the first phase untouched and modify only the second, where, on a test-by-test basis, we choose between pooled data (across all context values) and context-specific data based on whether the context variable enters the conditioning set. We instantiate this idea in two variants, PAC- and SAC-PCMCI$^+$, which differ in whether they exploit a persistence or the sparsity property of the context (cf. Sec. 3.1).

**Background: PCMCI$^+$.** PCMCI$^+$ consists of two phases. The skeleton (PC1) phase identifies, for each variable $X_t^j$, the superset of lagged parents $\hat{\mathbf{B}}^-(X_t^j)$, by running conditional independence tests (CITs) between pairs $(X_{t-\tau}^i, X_t^j)$ for $\tau > 0$ conditioned on subsets of adjacencies ranked by strength and incrementally increasing the conditioning set size. This removes edges arising from indirect paths and common causes. The output of PC1 is the superset of lagged parents for the union graph, which is used as input for the second phase. The second (MCI) phase then iterates over all adjacent pairs in the resulting skeleton, including contemporaneous ones, and tests $X_{t-\tau}^i \perp\!\!\!\perp X_t^j \mid \mathbf{S}, \hat{\mathbf{B}}^-(X_t^j), \hat{\mathbf{B}}^-(X_{t-\tau}^i)$ for subsets $\mathbf{S}$ of contemporaneous adjacencies. Conditioning on both lagged parent supersets controls for autocorrelation and serial confounding, and is the main reason PCMCI$^+$ remains well-calibrated in time-series settings. We do not modify the PC1 phase of the algorithm; all our changes target the MCI phase.

**Our modification: adaptive testing.** We modify the second phase of PCMCI$^+$, the MCI phase, to obtain the context-specific graphs using an adaptive testing approach similar to Günther et al. (2024). CITs are applied to pooled or context-specific data, depending on whether the context variables are in the conditioning set. Concretely, to learn the context-specific graph $G_{C=c}^{\mathrm{descr}}$, if $C_{t-\tau} \in \mathbf{S} \cup \hat{\mathbf{B}}^-(X_t^j) \cup \hat{\mathbf{B}}^-(X_{t-\tau}^i)$ for any $\tau \leq \tau_{\max}$, we replace the test on pooled data with the test

$$X_{t-\tau}^i \perp\!\!\!\perp X_t^j \mid \mathbf{S}, \hat{\mathbf{B}}^-(X_t^j), \hat{\mathbf{B}}^-(X_{t-\tau}^i)$$

*on context-specific data only*, where $\mathbf{S}$ denotes a subset of contemporaneous adjacencies. Intuitively, both algorithms implement the same principle of adaptive dataset selection based on the conditioning sets, but they partition the data differently: PAC-PCMCI$^+$ uses only fully persistent segments if any context lag matters, while SAC-PCMCI$^+$ uses only the specific context lags that are parents. Throughout, we denote the subset of context variables in $\mathbf{S} \cup \hat{\mathbf{B}}^-(X_t^j) \cup \hat{\mathbf{B}}^-(X_{t-\tau}^i)$ by $Z^C$.

**PAC-PCMCI$^+$.** The Persistent Adaptive Context (PAC)-PCMCI$^+$ version, presented in Alg. 1, exploits the persis-

---

**Algorithm 1** PAC-PCMCI$^+$ (focus on adaptive MCI-phase)

---

1: **Input:** Context var. $C$ with value $c$, time series data $\mathbf{X} = (X^1, \ldots, X^N, X^C = C)$, maximal time lag $\tau_{\max}$
2: **Output:** Context window graph $G_{C=c}$ for $c$
3: Run PC1-phase of PCMCI$^+$, return supersets of lagged parents $\mathcal{B}^-(\hat{X}^i)$ for all $X^i \in \mathbf{X}$
4: Initialize $G_{C=c}$ with all lagged links $\{\mathcal{B}^-(\hat{X}^i)\}_{i=1,\ldots,N}$ and fully connect all contemporaneous variables
5: **for** $p = 0$ **to** $D - 1$ **do**
6:   **for** all adjacent pairs $(X^i_{t-\tau}, X^j_t)$ for $\tau \geq 0$ in $G_{C=c}$ that satisfies $|\hat{\mathcal{A}}(X^j_t) \setminus \{X^i_{t-\tau}\}| \geq p$ **do**
7:     **for** all $S \subseteq \hat{\mathcal{A}}(X^j_t) \setminus \{X^i_{t-\tau}\}$ with $|S| = p$ **do**
8:       Set $Z = (S, \hat{\mathbf{B}}^-_t(X^j_t) \setminus \{X^i_{t-\tau}\}, \hat{\mathbf{B}}^-_{t-\tau}(X^i_{t-\tau}))$
9:       Test $X^i_{t-\tau} \perp\!\!\!\perp X^j_t \mid Z \setminus C^{\leq}, \{C^{\leq} = c\mathbb{1}\}$ if $C_{t-\tau} \in Z$ for any $0 \leq \tau \leq \tau_{\max}$,
10:       Additionally test $X^i_{t-\tau} \perp\!\!\!\perp X^j_t \mid Z$ on pooled data if $C_{t-\tau} \in Z \setminus S$ but $C_t \in \mathcal{S}$ for any $0 \leq \tau \leq \tau_{\max}$,
11:       else test $X^i_{t-\tau} \perp\!\!\!\perp X^j_t \mid Z$ on pooled data
12:       **if** independence is found **then**
13:         Delete link $X^i_{t-\tau} \to X^j_t$ for $\tau > 0$ (or $X^i_t \circ\!\!-\!\!\circ X^j_t$ for $\tau = 0$) from $G_{C=c}$
14:       **end if**
15:     **end for**
16:   **end for**
17: **end for**
18: Orient edges as in PCMCI$^+$ algorithm
19: **return** $G_{C=c}$

---

tence property, which restricts the number of interesting context-specific graphs to $K$, where $K$ is the number of values the context indicator can take at each time step. Here, all lags of the relevant context variables within the $\tau_{\max}$ time window are included in the conditioning set whenever any single lag is selected during MCI. Specifically, when $C_{t-\tau} \in \mathbf{S} \cup \hat{\mathbf{B}}^-(X^j_t) \cup \hat{\mathbf{B}}^-(X^i_{t-\tau})$ for any $\tau \leq \tau_{\max}$, we test

$$X^i_{t-\tau} \perp\!\!\!\perp X^j_t \mid (\mathbf{S}, \hat{\mathbf{B}}^-_t(X^j_t), \hat{\mathbf{B}}^-_{t-\tau}(X^i_{t-\tau})) \setminus C^{\leq}, \{C^{\leq} = c\mathbb{1}\}.$$

In words: we keep the same system variables in the conditioning set as in standard PCMCI$^+$, but *remove* every context variable from it ($\setminus C^{\leq}$) and instead restrict the data to those time points at which the entire $\tau_{\max} + 1$-long context window equals $c$ throughout ($C^{\leq} = c\mathbb{1}$). Persistence is what justifies this swap: if the context does not flip within a window, restricting to $C^{\leq} = c\mathbb{1}$ at the test time fixes *every* context value in the time window to $c$ simultaneously, so each lagged context is effectively conditioned on without appearing in the conditioning set. The price is fewer effective time points per test (only fully persistent windows), but the number of distinct context-specific tests stays at $K$.

Otherwise (if $C^{\leq} \cap (\mathbf{S} \cup \hat{\mathbf{B}}^-(X^j_t) \cup \hat{\mathbf{B}}^-(X^i_{t-\tau})) = \emptyset$), we test on pooled data: $X^i_{t-\tau} \perp\!\!\!\perp X^j_t \mid \mathbf{S}, \hat{\mathbf{B}}^-_t(X^j_t), \hat{\mathbf{B}}^-_{t-\tau}(X^i_{t-\tau})$.

Special consideration is required when the context appears in the superset of lagged parents $\hat{\mathbf{B}}^-(X^j_t) \cup \hat{\mathbf{B}}^-(X^i_{t-\tau})$, but is not a true union parent of $X^j_t$ or $X^i_{t-\tau}$. It can happen that $C_t$ is a child of both $X^j_t$ and $X^i_{t-\tau}$ (this case is not excluded by Asm. G.6; see App. G), so there is potential for selection bias. To alleviate this, additional CITs are run on pooled data: $X^i_{t-\tau} \perp\!\!\!\perp X^j_t \mid \mathbf{S}, \hat{\mathbf{B}}^-_t(X^j_t), \hat{\mathbf{B}}^-_{t-\tau}(X^i_{t-\tau})$

if $C_{t-\tau} \in \hat{\mathbf{B}}^-_t(X^j_t) \cup \hat{\mathbf{B}}^-_{t-\tau}(X^i_{t-\tau})$ for any $0 \leq \tau \leq \tau_{\max}$ and $C_t \notin \mathbf{S}$.

**SAC-PCMCI$^+$.** The Sparse Adaptive Context (SAC) version, presented in Alg. 2, takes a different approach, leveraging the sparsity property by conditioning only on the projection of the context values $\mathcal{C}$ onto the context-parents of the target, denoted $\mathcal{C}_{\text{proj}}$. In contrast to the persistent case, SAC-PCMCI$^+$ does *not* include all possible lags of the context variables in the conditioning set when testing for context-specific independencies. This requires knowing or discovering context parents, either by taking the adjustment set $Z$ and fixing the context variables to a value of interest, or directly by first discovering the union graph. If the union graph $\mathcal{G}^{\text{union}}_{\text{alg}}$ is learned, it suffices to re-test links that contain at least one node adjacent to the context to discover the context-specific graphs. To find all $K^{\tau_{\max}+1}$ possible context-specific graphs, we run $K^{|\text{Pa}^C_{\text{alg}}(X)|}$ tests per conditioning set (without context variables) $S$, where $|\text{Pa}^C_{\text{alg}}(X)| \leq \tau_{\max} + 1$ denotes the number of context parents of $X$. Thus, we test on context-specific data only if there is at least one context variable (at some lag) $C_{t-\tau}$ in the contemporaneous conditioning set $S$ or in the lagged parent supersets; otherwise we use the pooled dataset:

$$X^i_{t-\tau} \perp\!\!\!\perp X^j_t \mid (\mathbf{S}, \hat{\mathbf{B}}^-(X^j_t), \hat{\mathbf{B}}^-(X^i_{t-\tau})) \setminus Z^C,$$
$$\bigcup_{C \in Z^C} \{C = c\} \quad (1)$$

In words: only the context variables at the lags that *actually appear* in the conditioning set, collected in $Z^C$, are absorbed into a data restriction. We remove exactly these from the conditioning set ($\setminus Z^C$) and restrict to the time

points where they jointly equal $c$ ($\bigcup_{C \in Z^c} \{C = c\}$); the remaining lags of $C$ stay outside the test entirely. Unlike PAC, we therefore do not require the full $\tau_{\max} + 1$-window of $C$ to be persistent at $c$, which preserves more samples per test. The trade-off is multiplicative in the number of distinct combinations: each subset of context-parent lags appearing in some conditioning set induces its own restricted dataset, leading to $K^{|\operatorname{Pa}_{\text{alg}}^C(X)|}$ distinct tests.

## 6. Theoretical results

We show that the PAC-PCMCI$^+$ and SAC-PCMCI$^+$ methods soundly recover the adjacencies of the nodes in $\mathbf{X}_t$ in the $c$-context window graphs $G_{C=c}^{\text{descr}}$ (Thm. 6.1). The algorithms return reliable results only under a set of assumptions described below and in App. G, which ensure that changes in causal graphs are attributable to context rather than support issues, and that there are no feedback loops between context and system variables. We follow Günther et al. (2024) for the main assumptions necessary to extract a set of interpretable causal graphs from the observational data sampled from a system with endogenous context variables. First, the strong sufficiency assumption proposed in Günther et al. (2024) ensures that changes in graphs are due to changes in the value of the context variable(s), and not support problems. We adapt the strong context sufficiency assumption to the time series case in Asm. G.2. The other (weaker) versions of the sufficiency assumption can be adapted similarly. A detailed discussion of the assumptions can be found in App. G.1.

Now, we highlight key differences between the persistent and sparse case, as well as challenges that arise when extending the results from the i.i.d. case to the time-series case. The time-series context-specific versions of the Markov property (Lemma H.2 and Lemma H.6) and faithfulness (Lemma H.3 and Lemma H.7), which link d-separation in $c$-context graphs and conditional independence in the population distribution, differ between the persistent and sparse cases. Under persistence, valid conditioning sets must either include the entire context window or exclude it entirely: $C^{\leq} \subset Z$ or $C^{\leq} \cap Z = \emptyset$. Under sparsity, it suffices that $Z$ contains some context lag (or excludes all): $C_{t-\tau} \in Z$ for some $\tau \leq \tau_{\max}$ or $C^{\leq} \cap Z = \emptyset$. Thus, there are fewer admissible $Z$ and stronger restrictions for the Markov condition in the persistent case than in the sparse case. By the same arguments, the faithfulness requirement for the persistent version is weaker than for the sparse case. These different restrictions also require different acyclicity assumptions, as our procedures rely on separating context ancestors from context descendants (the split need not be known a priori, but the two sets must not overlap). For the persistent case, this distinction needs to be possible on a summary graph level, whereas for the sparse version the distinction is for-

mulated for the unrolled time series graph, see App. G.2. Another implication is that PAC-PCMCI$^+$ can only soundly discover the $K$-many context-specific graphs for constant values over $C^{\leq}$, while SAC-PCMCI$^+$ can discover all possible context-specific graphs $G_{C^{\leq}=\mathbf{c}}^{\text{descr}}$. We discussion the output of our algorithms and their interpretation in App. F.

We prove Thm. 6.1 in App. G.1. The general proof-strategy carries over from the i.i.d. case. However, in the time-series setting, the proof must account for influences from variables outside the time window, and possible interactions among multiple lagged context variables. Furthermore, proving soundness of SAC-PCMCI$^+$ requires careful handling of non-parental context variables in the conditioning sets. To orient the edges, we apply the collider phase of PCMCI$^+$ which is based on the majority rule, which has been proven complete in Runge (2020, Thm. 3).

**Theorem 6.1** (Soundness of the method for time series). *Under suitable assumptions (Asm. 1 or 2, together with Asm. G.6 or G.7, respectively, and Asm. G.2, with minimal edge flips), the oracle versions of PAC-PCMCI$^+$ and SAC-PCMCI$^+$ recover the unoriented edges of $G_{C=c}^{\text{descr}}$ involving at least one variable in $X_t$.*

## 7. Evaluation

### 7.1. Synthetic data

**Data generation** We generate data from an additive linear SCM following the procedure from Sec. I. Briefly, we sample a linear acyclic *base graph* on up to $D+1$ nodes at sparsity $s = 0.6$ with both contemporaneous and lagged links up to $\tau_{\max} \in \{2, 4\}$, where $D = 7$, of the form $X_t^j = a_j X_{t-1}^{j-1} + \sum_{i=1}^{D+1} \sum_{\tau=0}^{\tau_{\max}} b_{i,\tau} X_{t-\tau}^i + \eta_t^j$, for $j = 1, \ldots, D+1$. Because selection bias can arise when the context has at least one contemporaneous parent (Sec. 3.1), we control the share of contemporaneous links via $f_c = 0.2$. To obtain context-specific graphs, we randomly pick one node as the *context indicator* $C$, which may be exogenous or endogenous. Using a flag $endo_c$, we enforce exogenous vs. endogenous contexts by restricting candidates to variables with no parents (exogenous) or with parents (endogenous). After generating values for all *ancestors* of $C$ and a continuous $C$, we derive the discrete context by thresholding $C$ at the $n_{\text{contexts}} - 1$ quantiles (imbalance factor $b = 1$). We set $n_{\text{contexts}} = 2$. Context-dependent descendants of $C$ are then drawn based on the value of $C$ by editing $n_{\text{change}} \in \{2, 4\}$ links of the base graph by removing edges, which preserves acyclicity. Edited nodes are children of $C$ and have an edge from $C$ to that variable. We emphasize that, although the structural equations are additive in their parents, the relationship between context and system variables in our experiments is *non-additive*, consistent with Def. G.2: the context does not enter the system equations as an additive term, but

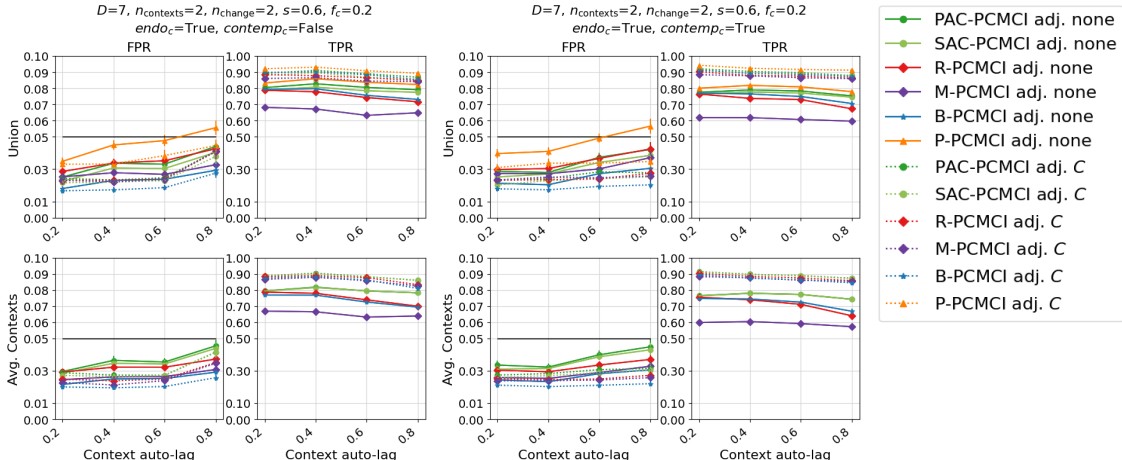

*Figure 2.* Results for PAC-PCMCI$^+$ (persistent), SAC-PCMCI$^+$ (sparse), B-PCMCI (baseline), M-PCMCI (individual datasets), R-PCMCI (Regime-PCMCI with known context) and P-PCMCI (pooled datasets) with endogenous context variables, either lagged (left panel) or contemporary (right panel). Solid lines indicate settings where no context-system links are set as link assumptions, while dotted lines indicate settings where links context-system links are known (see Sec. 7).

instead modifies the graph structure itself by activating or deactivating edges between system variables, depending on its value. Context persistence is varied via autocorrelation $a_C \in \{0.2, 0.4, 0.6, 0.8\}$. For system variables, autocorrelations are sampled as $a_j \in \{-0.4, -0.3, -0.2, 0.2, 0.3, 0.4\}$. System-system link strengths are drawn from $b_{i,\tau} \in \{0.6, 0.7, 0.5, -0.4, -0.5, -0.6, -0.7, 0.4\}$, and noises are $\eta_t^j \sim \mathcal{N}(0,1)$. We vary graph size as $D \in \{7, 9, 14\}$, sample sizes as $N \in \{200, 600, 1000, 2000\}$. We evaluate for both linear and non-linear system-system links, and, to evaluate the robustness of our method to misalignment of context variables, we introduce a controlled misspecification of the context variables. Further details appear in App. I.

**Evaluated methods** We benchmark against several PCMCI$^+$ variants. M-PCMCI applies PCMCI$^+$ separately to each context by masking, and should suffer from higher FPR due to selection bias, and lower TPR, as links involving the context indicator cannot be detected. P-PCMCI applies PCMCI$^+$ on the pooled dataset, thereby recovering only the union graph. Its TPR should be slightly higher since all links are tested on all samples. Following Günther et al. (2024), we also include B-PCMCI, which runs PCMCI$^+$ on each context by masking and on the pooled data separately, and intersects the resulting edge sets. We also evaluate Regime-PCMCI (Saggioro et al., 2020) (R-PCMCI), assuming known regime assignments. Our method is the first to address endogenous contexts in time series with constraint-based methods, therefore no direct baseline exists.

**Results** Fig. 2 shows performance varying across persistence levels. The results show that our methods achieve three key advances, as follows. (1) **Improved accuracy:** With known context-system links (dotted lines), PAC-PCMCI$^+$ and SAC-PCMCI$^+$ achieve the lowest false pos-

itive rate (FPR) while maintaining high true positive rate (TPR). This validates that our adaptive testing strategy correctly handles selection bias from endogenous contexts. **(2) Better sample efficiency:** SAC-PCMCI$^+$ achieves the highest TPR among all methods because it makes optimal use of the available data due to the adaptive testing. This is particularly valuable in domains with limited samples per context. P-PCMCI achieves the highest TPR, but also the highest FPR. This can be explained by the fact that pooling increases the chance of detecting context–system links, which increases the likelihood of larger adjacency sets and, implicitly, the likelihood of a larger number of CITs. M-PCMCI must repeat all tests for each context separately, and exhibits high FPR because it cannot distinguish between contexts that have lagged dependencies with the system. The baseline B-PCMCI performs comparably to PAC-PCMCI$^+$, our method under persistence, but with slightly higher FPR due to more tests being performed. R-PCMCI achieves low FPR but also the low TPR, since context–system links are not explicitly tested and fewer tests are run overall. FPR increases moderately with higher context autocorrelation across all methods. With strong persistence, multiple lags of the context become redundant, so conditioning sets effectively include near-duplicates of the same variable, which inflates variance in CI tests and leads to more spurious rejections. Further results, including an evaluation of computational times, are presented in App. J.

### 7.2. Real-world data

To demonstrate practical applicability, we apply PAC-PCMCI$^+$ and SAC-PCMCI$^+$ to a climate dataset of soil moisture land-atmosphere feedbacks (Popescu et al., 2024) (see App. J.8). Understanding how soil moisture contexts

*Table 1.* Link-wise comparison against ground truth. PAC-PCMCI$^+$ and SAC-PCMCI$^+$ produce more interpretable graphs using less expert-links than Regime-PCMCI. We focus on the skeleton.

| Ground Truth | PAC-PCMCI$^+$ | SAC-PCMCI$^+$ | Regime-PCMCI |
|---|---|---|---|
| Ground truth not context-dependent edges | Mostly correct | Mostly correct | Correct |
| Context-dependent edge LH - SM | Missing | Missing | Present in all regimes, no reversal |
| Proxy edge LH - TP | Present in all regimes (lag changes in moist) | Present in all regimes (lag changes and reversed in moist) | No link |
| Spurious edges in summary graph | 4 | 4 | 7 |

modulate temperature results in critical for accurate heat-wave prediction (Seneviratne et al., 2010). We summarize our findings, detailed in App. J.8: The algorithms detect a positive precipitation to latent heat link in the dry and normal contexts and a negative latent heat to precipitation link in the moist context. As latent heat is primarily driven by soil moisture, these dependencies may reflect both genuine feedbacks and confounding, but, overall, the results demonstrate that our method can identify meaningful land–atmosphere interactions in complex real-world systems. We also compare our results to the expert graph and results obtained using Regime-PCMCI in (Popescu et al., 2024) in Tab. 1.

**Code** All code and experiments can be found at https://github.com/oanaipopescu/pac_sac_pcmci.

## 8. Discussion and conclusion

**Summary** We introduced a framework for causal discovery in time series with endogenous context variables, allowing feedback between context and system and relaxing the standard exogeneity assumption. We formalized the challenges posed by temporally dynamic contexts and identified two complementary properties, persistence and sparsity, that render the problem tractable. Based on these insights, we proposed PAC-PCMCI$^+$ and SAC-PCMCI$^+$, two constraint-based extensions of PCMCI$^+$ inspired by Günther et al. (2024) that recover context-specific causal graphs while mitigating selection bias via adaptive testing. We provided theoretical guarantees and validated the methods on synthetic data and a real-world climate application.

**Strengths** Compared to existing approaches, our methods (1) handle endogenous contexts rather than assuming exogeneity, enabling applications when regimes emerge from internal dynamics; (2) improve statistical sample efficiency through adaptive testing that combines pooled and context-specific information; and (3) require few hyperparameters relative to optimization-based alternatives, advantageous in

small-sample settings. Our work represents a step toward understanding complex systems with distribution shifts driven by internal dynamics. By formalizing endogenous contexts in time series and providing practical algorithms with theoretical guarantees, we hope to encourage further research on causal modeling in domains where regime changes emerge from system behavior.

**Limitations** Our approach relies on acyclicity conditions and the persistence and sparsity properties, which are plausible and testable in many domains but may not always hold. When both fail, the problem remains exponentially hard. A key limitation is the assumption of known, discrete context variables. While common in prior work and often supported by domain knowledge (e.g., soil moisture or volatility regimes), this assumption may not always be feasible. Although our method is modular and compatible with any context assignment procedure, including changepoint detection, detected contexts can introduce deterministic relationships that violate faithfulness, see Mooij et al. (2020). Discretizing continuous signals to define contexts may hide weak effects, increasing false negatives. The trade-off between sensitivity and interpretability depends on discretization choices and should be guided by domain expertise. In addition, persistent contexts induce strong autocorrelation, and the interventions over entire time windows (as done in PAC-PCMCI$^+$) reduce the effective sample size, potentially limiting statistical power, though our real-world results show that meaningful discovery remains possible.

**Future work** Future directions include extending the framework to unobserved contexts by integrating context detection, handling deterministic contexts derived from system variables, and exploiting sparsity more aggressively by restricting conditioning sets to few context variables, potentially guided by dependence-based heuristics. Another important direction for future work is to extend the framework to allow limited violations of sparsity (e.g., bounded numbers of context parents, variables with multiple context parents, or limited interactions among the contexts).

## Acknowledgments

W.G. and J.R. have received funding from the European Research Council (ERC) Starting Grant CausalEarth under the European Union's Horizon 2020 research and innovation program (Grant Agreement No. 948112). O.-I.P. and M.R. have received funding from the European Union's Horizon 2020 research and innovation programme under grant agreement No 101003469 (XAIDA). This work used resources of the Deutsches Klimarechenzentrum (DKRZ) granted by its Scientific Steering Committee (WLA) under project ID 1083.

## Impact Statement

This paper presents work whose goal is to advance the field of Machine Learning. There are many potential societal consequences of our work, none which we feel must be specifically highlighted here.

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

## A. Pseudocode

Here, we provide the pseudocode for the SAC-PCMCI$^+$ algorithm.

---

**Algorithm 2** SAC-PCMCI$^+$ (focus on adaptive MCI-phase)

---

1: **Input:** Context indicator $C$, time series dataset $\mathbf{X} = (X^1, \ldots, X^N, X^C = C)$, maximal time lag $\tau_{\max}$, tuple of context-values $\mathbf{c}$
2: **Output:** Context-specific graph $G_{C=c}$ for $c$
3: Run PC1-phase of PCMCI$^+$, return supersets of lagged parents $\hat{\mathcal{B}}^-(X^i)$ for all $X^i \in \mathbf{X}$
4: Initialize time-series graph $G_{C=c}$ with all lagged links according to $\{\hat{\mathcal{B}}^-(X^i)\}_{i=1,\ldots,N}$ and fully connect all contemporaneous variables
5: **for** $p = 0$ **to** $D - 1$ **do**
6:    **for** all adjacent pairs $(X^i_{t-\tau}, X^j_t)$ for $\tau \geq 0$ in $G_{C=c}$ that satisfy $|\hat{\mathcal{A}}(X^j_t) \setminus \{X^i_{t-\tau}\}| \geq p$ **do**
7:       **for** all $S \subseteq \hat{\mathcal{A}}(X^j_t) \setminus \{X^i_{t-\tau}\}$ with $|S| = p$ **do**
8:          Set $Z = (S, \hat{\mathbf{B}}^-_t(X^j_t) \setminus \{X^i_{t-\tau}\}, \hat{\mathbf{B}}^-_{t-\tau}(X^i_{t-\tau}))$
9:          Set $Z^C = Z \cup C^{\leq}$
10:          Test $X^i_{t-\tau} \perp\!\!\!\perp X^j_t \mid Z \setminus Z^C, \{Z^C = \mathbf{c}\}$ if $C_{t-\tau} \in Z$ for any $0 \leq \tau \leq \tau_{\max}$,
11:          else test $X^i_{t-\tau} \perp\!\!\!\perp X^j_t \mid Z$ on pooled data
12:          **if** independence is found **then**
13:             Delete link $X^i_{t-\tau} \to X^j_t$ for $\tau > 0$ (or $X^i_t \circ\!\!-\!\!\circ X^j_t$ for $\tau = 0$) from $G$
14:          **end if**
15:       **end for**
16:    **end for**
17: **end for**
18: Orient edges as in PCMCI$^+$ algorithm
19: **return** $G_{C=c}$

---

## B. Relation to J-PCMCI$^+$

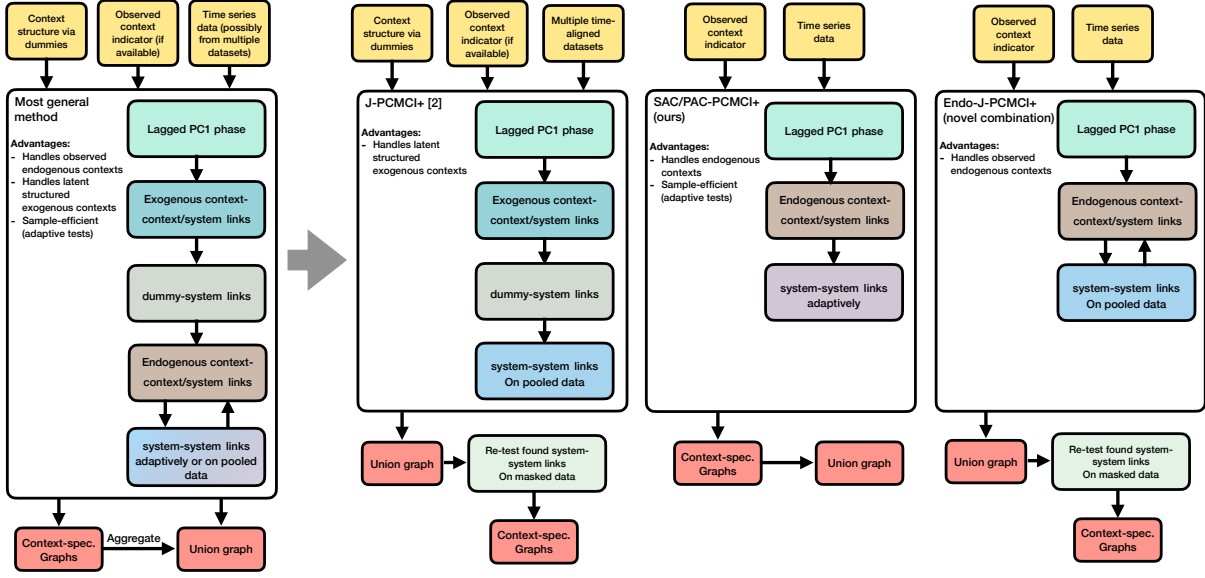

*Figure 3.* Overview of the general context-aware causal discovery framework and its derived methods. From top to bottom, the figure illustrates inputs, methodological steps, and outputs. The leftmost flowchart depicts the most general method, which serves as a set of building blocks from which the methods to the right are derived. J-PCMCI$^+$ and PAC/SAC-PCMCI$^+$ appear as special cases addressing different settings—latent structured exogenous context confounding versus endogenous (observed) contexts—and producing different outputs. As an additional example, Endo-J-PCMCI$^+$ is shown as a novel combination that applies pooled data to infer a union graph in the presence of possibly endogenous, but observed, context variables.

We illustrate how our proposed method can be unified with J-PCMCI$^+$, which is itself a variant of PCMCI$^+$ for context-dependent data, in Fig. 3. The figure is organized from top to bottom into inputs, method, and outputs. The leftmost

flowchart represents the most general PCMCI-based framework, from which the methods shown to the right are derived as special cases. In particular, J-PCMCI$^+$ and PAC/SAC-PCMCI$^+$ address different forms of context dependence—latent structured exogenous context confounding versus endogenous (observed) contexts—and therefore differ both in their assumptions and in their outputs. Additional variants can be constructed from the same building blocks; as an example, we present Endo-J-PCMCI$^+$, which uses pooled data to infer a union graph in the presence of possibly endogenous but observed context variables.

J-PCMCI$^+$ extends PCMCI$^+$ by pooling data across observed contexts and introducing time- and space-dummy variables to represent latent but structured context effects, yielding a union graph under exogenous context assumptions. In contrast, PAC/SAC-PCMCI$^+$ focuses on endogenous contexts and achieves sample efficiency via adaptive testing. The two methods thus address complementary aspects of context dependence and rely on different assumptions regarding context observability and exogeneity. Their key ideas can be combined in a modular fashion within the MCI phase: links involving observed exogenous contexts are tested conditional on lagged parents as in J-PCMCI$^+$; latent structured context effects are handled via dummy variables; and links involving endogenous contexts are discovered using adaptive pooled or context-restricted tests, depending on whether context or dummy variables appear in the conditioning set. If only a union graph is required, this strategy simplifies to testing exclusively on pooled data. Under the appropriate assumptions, J-PCMCI$^+$ and PAC/SAC-PCMCI$^+$ are recovered as special cases of this unified framework, which clarifies which components to activate in which setting.

## C. Selecting between PAC-PCMCI$^+$ and SAC-PCMCI$^+$

In real world application cases, one can asses the properties of persistence and sparsity in data. Namely persistence can directly be estimated, e.g., via frequency of context switches, and sparsity can be verified by inspecting the union graph across contexts.

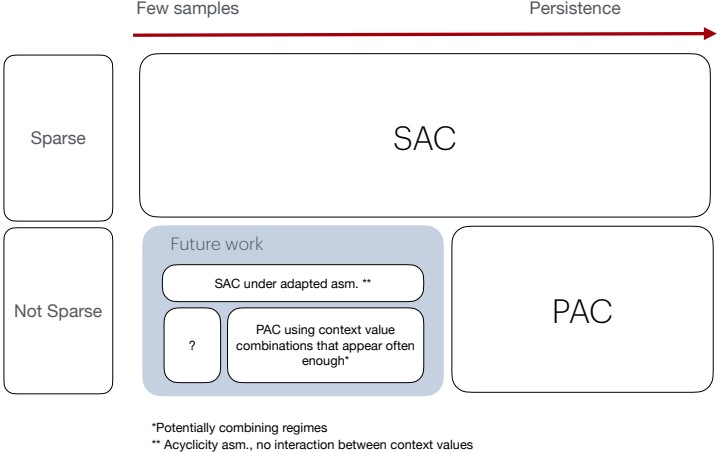

*Figure 4.* Illustration on performance trade-offs of PAC-PCMCI$^+$ and SAC-PCMCI$^+$ when neither persistence nor sparsity holds perfectly.

To make trade-offs explicit, in the situation where neither of these properties hold, the schematic figure 4 illustrates how one might proceed. The figure maps performance across two axes: (i) degree of persistence and (ii) degree of sparsity violation. This diagram summarizes both theoretical expectations and empirical findings from our synthetic experiments, and indicate which method is preferable in different regimes.

Our results suggest the following guidance. When sparsity holds, SAC-PCMCI+ should generally be preferred due to its superior finite-sample efficiency, even in the presence of persistence. In contrast, PAC-PCMCI+ remains robust when sparsity is violated, particularly in settings with persistent contexts and sufficient data to support reliable estimation. This can be seen in Fig. 14, where we show performance of the methods when they are applied outside of the ideal regime to quantify this robustness.

Importantly, the persistence assumption primarily affects finite-sample performance rather than asymptotic correctness.

Moreover, PAC-PCMCI+ can be applied to any subset of context combinations that occur sufficiently often, providing additional flexibility in practice.

Finally, contexts can sometimes be redefined (e.g., by grouping variables) to induce persistence or sparsity, thereby making either method applicable and mitigating context explosion.

## D. Examples of context-dependent systems

As described in Def. 3.1, a context variable is a discrete time series. Furthermore, the context variable can have multiple children Sec. 3.1. Further, we allow a system node to have multiple context nodes (with possibly different lags) as parents. Here, we provide some further examples of systems with endogenous context variables. Example D.1 contains a variable with two context parents that interact non-trivially, while Example D.2 presents an example of a system with two context variables, where one of the context variables has multiple parents.

*Example* D.1 (Attention in a classroom). $S_t$ is the number of sleep hours of a student the night before class, $N_t$ is the noise in the classroom, $T_t$ is the teacher engagement, and $A_t$ is the attention of the student. We assume that when the student is tired or the classroom is too noisy, no level of teacher engagement can make the student be attentive in class. Thus, we have two contexts, tiredness $C_t^1$ and noisiness $C_t^2$. We can model the SCM as follows:

$$
\begin{aligned}
S_t &= 0.1 \cdot S_{t-1} + \eta_t^S \\
N_t &= 0.2 \cdot N_{t-1} + \eta_t^N \\
T_t &= 0.3 \cdot T_{t-1} + \eta_t^T \\
C_t^1 &= \mathbb{1}(-0.5 \cdot S_{t-1} + \eta_t^{C^1} > \theta_1) \\
C_t^2 &= \mathbb{1}(0.6 \cdot N_{t-1} + \eta_t^{C^2} > \theta_2) \\
A_t &= (1 - C_t^1) \cdot (1 - C_t^2) \cdot T_{t-1}
\end{aligned}
\tag{2}
$$

*Example* D.2 (The electricity grid). $W_t$ wind speed and $T_t$ temperature. $D_t$ electricity demand is driven by temperature and wind forecasts but only when the wind is stable enough to be forecasted well. Electricity generation $G_t$ is driven by the wind, but only if there is enough wind.

$$
\begin{aligned}
T_t &= 0.4 \cdot T_{t-1} + \eta_t^T \\
W_t &= 0.2 \cdot W_{t-1} + \eta_t^W \\
C_t^1 &= \mathbf{1}(0.7 \cdot T_{t-1} + \eta_t^{C^1} > \theta_1) \\
C_t^2 &= \mathbf{1}(0.6 \cdot (W_{t-1} - W_{t-2}) + \eta_t^{C^2} > \theta_2) \\
D_t &= 0.2 \cdot D_{t-1} + 0.7 \cdot T_t + (1 - C_t^2) \cdot (0.4 \cdot W_{t-1}) + \eta_t^D \\
G_t &= 0.8 \cdot W_t + (1 - C_t^1) \cdot (0.6 \cdot D_t) + \eta_t^G
\end{aligned}
$$

## E. Alternative modeling approach for time- and context-dependence in SCMs

In this work, we represent context dependence using context variables modeled as time series. Alternative modeling choices are possible; one such alternative is described in this section.

In a time series setting, one may argue that at each time point there exists a single fixed model relating the present to the past, and that any context affecting this model therefore exists only at lag zero. Figure 5 illustrates how context dependence can be represented while restricting context to act only contemporaneously on the system.

However, the true context is typically unobserved and only accessible through proxy variables. We extend the dam example 3.1 by adding a retention basin between the dam and the measurement station whose status (full or not) is unobserved. Then, the effective context is given by the (unobserved) state of this basin. A suitable proxy for this context at time $t$ is the dam state at time $t-1$, which carries information about whether the basin is likely full. We might also include the current state of the dam as a context variable as well. This motivates the introduction of two context variables, $C_t^0 = D_t$ and $C_t^1 = D_{t-1}$, which are shifted copies of each other (Figure 5). Formally, the mechanism of the unobserved context variable ($f_1$ when expressing $C_t^1 = f_1(D_{t-1})$ in the example) is close to the identity mapping applied to an observed variable, possibly with a

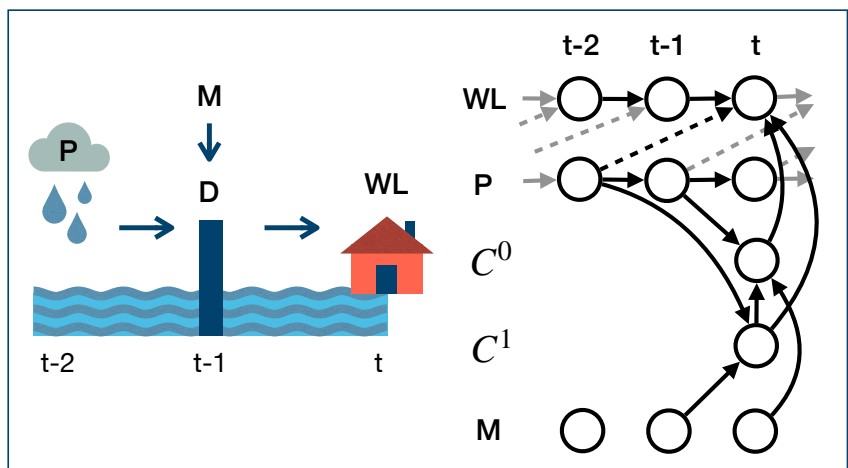

*Figure 5.* Alternative context modeling for the dam example. The effective (unobserved) context, such as the state of an intermediate retention basin, is represented using observed proxies derived from the dam state. Two context variables are introduced, $C_t^0 = D_t$ and $C_t^1 = D_{t-1}$ , which are shifted copies of each other and deterministically related. The figure illustrates how restricting context influence to be contemporaneous leads to duplicated context–system edges and obscures temporal ordering, highlighting challenges for causal structure learning.

non-trivial lag.

Although, in principle, restricting context to lag zero is sufficient to characterize the system's behavior, this modeling choice has significant practical drawbacks, especially for automated causal structure learning. First, temporal ordering cannot be fully exploited to orient context–system edges: although some context variables are derived from lagged system variables and therefore precede other variables at time $t$, this ordering information is obscured by the time shift. Second, introducing context variables as deterministic copies of existing variables induces determinism in the model, leading to violations of faithfulness that must be handled explicitly.

### E.1. Statistical efficiency vs. computational cost of PAC/SAC

We provide a more detailed discussion of the trade-offs between PAC/SAC and naive masking in terms of statistical efficiency, the number of conditional independence tests (CITs), and runtime, as the relationship between these quantities is subtler than it may initially appear.

The principal advantage of PAC/SAC over naive masking is improved statistical (sample) efficiency: rather than partitioning the dataset into per-context subsamples, PAC/SAC pools information across contexts wherever the underlying conditional independence relations are shared. The impact on the total number of CITs and on runtime, however, is more nuanced, and a clean ordering between methods only holds in an oracle-CIT regime.

In constraint-based causal discovery, the overall CIT count is strongly influenced by early edge deletions, and *incorrect* deletions (for instance, those caused by low effective sample sizes) can reduce the test count without reflecting any gain in accuracy. A smaller CIT count is therefore not, on its own, evidence of a better method. Two competing effects shape the comparison:

- Compared to masking, PAC/SAC explicitly include tests involving context variables, which tends to *increase* the number of CITs.

- Masking, conversely, operates on smaller per-context samples and frequently deletes edges early, sometimes incorrectly, thereby lowering the overall CIT count at the cost of soundness. This is consistent with earlier observations that the runtime of M-PC is lower than that of the adaptive-testing method in the i.i.d. case (Günther et al., 2024).

- On the other hand, PAC/SAC can soundly remove (i) spurious dependencies induced by selection bias upstream of

context variables and (ii) context-specific edges that are irrelevant for a given regime; these earlier deletions can in turn reduce the number of subsequent CITs.

**Illustrative example.**   Consider a simple setting with system variables $X, Y, Z, W$ and a context variable $C$, where $C$ influences $Z$ and induces a spurious association between $X$ and $Y$ via selection on $C$. In an idealized regime with oracle CI decisions and known context-system links, the three approaches behave as follows:

- A union-graph approach retains all edges across contexts and yields a relatively dense intermediate graph.

- Masking evaluates each context separately but cannot remove edges arising from selection bias.

- PAC/SAC explicitly test context-system relations and remove both selection-bias edges upstream of $C$ and edges that are only present in specific contexts, producing a sparser intermediate graph and thus requiring fewer CITs in later stages.

In summary, the primary advantage of PAC/SAC over masking lies in improved *statistical* efficiency rather than in a guaranteed reduction of runtime or CIT count. Empirical CIT counts and runtimes therefore need to be interpreted carefully, in conjunction with measures of soundness such as recall on true edges.

# F. Graphical outputs and object interpretation

### F.1. Graphical objects

**Context-specific graphical objects**   Following the main paper, we introduce graphical objects that connect context-specific independencies to structural causal models (SCMs), in the spirit of (Günther et al., 2024) for the i.i.d. case. These objects allow us to precisely specify the outputs of our algorithms, clearly state the underlying assumptions, and provide the foundation for the proofs in App. H.

The motivation is that, depending on the observational distribution, some contexts may have poor or no support. Graphs fitted from such data can then exhibit missing edges due to lack of support rather than changes in the underlying mechanisms. We therefore make the relevant distributions precise in the time–series setting.

**Setup and notation**   We assume the joint model (context and system) generates a stationary multivariate sequence $\xi = \{X_t\}_{t \in \mathbb{Z}}$. At time $t$, the observable endogenous variables are $V_t = \{V_t^i : i \in I\}$ and the exogenous noises are $U_t = \{\eta_t^i : i \in I\}$, with $V_t^i \in \mathcal{X}_i$ and $\eta_t^i \in \mathcal{N}_i$ for a finite index set $I$. We abbreviate $V_{t-\tau_{\max}:t}$ and $U_{t-\tau_{\max}:t}$ by $V$ and $U$. Together with the structural equations $\mathcal{F}$ which determine the observed endogenous variables $V$ given the exogenous variables $U$ at each time step $t$, these objects form a structural causal model SCM $M_t = (V, U, \mathcal{F})$ at each time step $t$. Let $C_t$ denote the (observable) context at time $t$, and $C^{\leq} := C_{t-\tau_{\max}:t}$ the context window. A *mask* is a concrete sequence $\mathbf{c} \in \mathcal{C}^{\tau_{\max}+1}$ assigned to $C^{\leq}$.

Given any mask $\mathbf{c}$, we define the following distributions over the variables in the window (excluding $C$):

$$P_M\big(V \mid C^{\leq} = \mathbf{c}\big) \quad \text{and} \quad P_M\big(V \mid \mathrm{do}(C^{\leq} = \mathbf{c})\big).$$

The first one is to be understood as the distribution over $V$ when $C^{\leq}$ takes exactly the sequence of values $\mathbf{c}$. The latter is interpreted as intervening on $C^{\leq}$ at each time point in the window on *separate* independent copies of $\xi$ (so that the interventions for distinct windows do not interact).

**Graphs**   We now define three context-window graphs that share the same intervened mechanisms but differ in which data distribution is used.

**Definition F.1** (Mechanism graph). The mechanism graph $G_t[\mathcal{F}]$ at time $t$ is the graph constructed from the structural parent sets $\mathrm{Pa}(X_t^i)$ for each variable $X_t^i$, as specified by the structural equations $\mathcal{F}$.

The observable graph $G[\mathcal{F}, P]$ for a set of structural equations $\mathcal{F}$ and a distribution $P(V)$ over the endogenous variables $V$ is defined as in (Günther et al., 2024). Namely, it is constructed via parent sets $\mathrm{Pa}_i' \subset \mathrm{Pa}_i$ (of $X_i$), such that $j \notin \mathrm{Pa}_i'$ if and only if for all values $\bar{\mathrm{pa}}$ of $\mathrm{Pa}_i - \{j\}$ the mapping $x_j \mapsto f_i|_{\mathrm{supp}\, P(\mathrm{Pa}_i)}(x_j, \bar{\mathrm{pa}}, \eta_i)$ is constant (where defined).

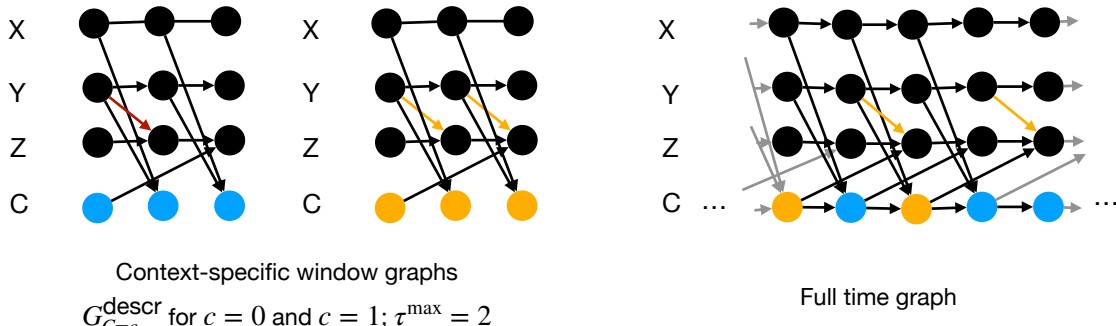

Context-specific window graphs
$G_{C=c}^{\text{descr}}$ for $c=0$ and $c=1$; $\tau^{\max}=2$

Full time graph

*Figure 6.* Illustration of the graph corresponding to the SCM given in example F.1. Colors indicate context values in $\{0,1\}$. *Red edges* mark dependencies that arise due to context parents *outside* the $\tau_{\max}$-window. Since we only select windows with the sequence $01$ in many instances of the sequence $\xi$ to obtain the distribution $P(V|C^{\leq}=01)$, there will be some with $C_{t-3}=1$. This implies, the edge $Y_{t-2} \to X_{t-1}$ appears in those cases, making this edge effectively present in $G_{C=0}^{\text{descr}}$.

**Definition F.2** (descriptive context window graph). The descriptive context window graph is defined as $\bar{G}_{C^{\leq}=\mathbf{c}}^{\text{descr}} := G[\mathcal{F}_{\text{do}(C^{\leq}=\mathbf{c})}^{M}, P_M(V|C^{\leq}=\mathbf{c})]$.

This graph retains the causal relations of the intervened model $\mathcal{F}_{(C^{\leq}=\mathbf{c})}^{M}$ that can be recovered from observational data restricted to the context $C^{\leq}=\mathbf{c}$. Since the context variables $C$ are constant per-context, we add edges involving the context variables $C^{\leq}$ from the union graph to $\bar{G}_{C^{\leq}=\mathbf{c}}^{\text{descr}}$ and denote the result $G_{C^{\leq}=\mathbf{c}}^{\text{descr}}$.

**Definition F.3** (physical context window graph). The physical context window graph is defined as $G_{C^{\leq}=\mathbf{c}}^{\text{phys}} := G[\mathcal{F}_{\text{do}(C^{\leq}=\mathbf{c})}^{M}, P_M(V)]$.

The physical graph uses *all* available data to describe the relations present under the intervened mechanisms.

**Definition F.4** (counterfactual context window graph). The counterfactual context window graph is defined as $G_{C^{\leq}=\mathbf{c}}^{\text{CF}} := G[\mathcal{F}_{\text{do}(C^{\leq}=\mathbf{c})}^{M}, P_M(V|\text{do}(C^{\leq}=\mathbf{c}))]$.

The counterfactual context window graph is a theoretical construct which employs the post-intervention distribution that would arise after intervening on all context windows separately.

### F.2. Output interpretation

We now provide an example of the output of PAC-PCMCI$^+$ on context-dependent data generated from a synthetic ground-truth SCM, and then connect the learned graphs to the context-window graphs from Def. 4.2.

*Example* F.1. We consider a time- and context-dependent SCM in which the context has a *lagged* effect on the system:

$$
\begin{aligned}
X_t &= 0.9\,X_{t-1} + \eta_t^X, \\
Y_t &= 0.8\,Y_{t-1} + \eta_t^Y, \\
C_t &= \begin{cases} 1, & \text{if } 0.6\,X_{t-1} + 0.9\,Y_{t-1} + \eta_t^C > 0, \\ 0, & \text{otherwise}, \end{cases} \\
Z_t &= 0.7\,Z_{t-1} + \mathbb{1}(C_{t-2}=1)\left(1.5\,Y_{t-1}\right) + \eta_t^Z,
\end{aligned}
$$

The $c$-context window graphs $G_{C=0}^{\text{descr}}$ and $G_{C=1}^{\text{descr}}$ (i.e., for constant masks $C^{\leq}=c\mathbf{1}$) are illustrated in Fig. 6.

For a binary context and a window of length $\tau_{\max}+1=3$, there are $2^3=8$ possible context-window graphs; we display only the two constant-mask graphs and omit the remaining 6 mixed-mask graphs (context explosion; see Sec. 3.1).

PAC-PCMCI$^+$ then outputs the graphs in Fig. 7 for $C^{\leq}=01\mathbb{1}$ and $C^{\leq}=11\mathbb{1}$, respectively.

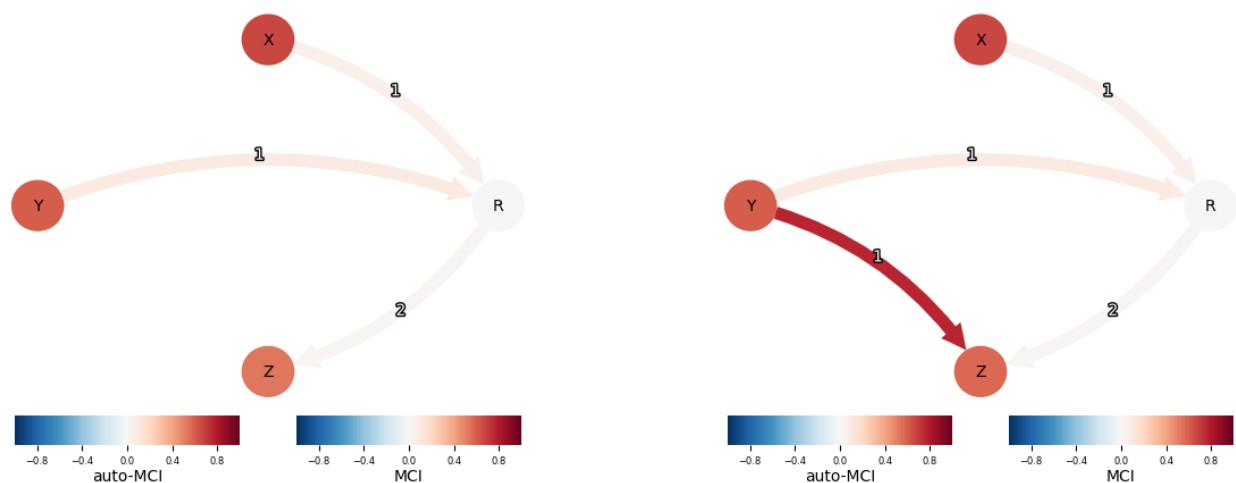

*Figure 7.* PAC-PCMCI$^+$ output for constant masks $C^{\leq} = 0\mathbb{1}$ (left) and $C^{\leq} = 1\mathbb{1}$ (right).

By design, PAC-PCMCI$^+$ returns the $c$-context *window* graphs (constant mask $C^{\leq} = c\mathbf{1}$) and only for variables at time $t$. Edges that manifest *only* under mixed masks or through context parents outside the window will therefore not appear in the output. This is intentional and matches the definition of the descriptive context-window graphs in Def. 4.2.

We illustrate how these learned graphs can be combined to form the full time graph in Fig. 8 for the following example.

*Example* F.2. We consider a time- and context-dependent SCM in which the context has a *lagged* effect on the system:

$$X_t = 0.9\,X_{t-1} + \eta_t^X,$$
$$Y_t = 0.8\,Y_{t-1} + \eta_t^Y,$$
$$C_t = \begin{cases} 1, & \text{if } 0.6\,X_{t-1} + 0.9\,Y_{t-1} + \eta_t^C > 0, \\ 0, & \text{otherwise}, \end{cases}$$
$$Z_t = 0.7\,Z_{t-1} + \mathbb{1}(C_{t-1} = 1)\,\big(1.5\,Y_{t-1}\big) + \mathbb{1}(C_t = 0)\,\big(0.5\,X_{t-1}\big) + \eta_t^Z,$$

## G. Assumptions

### G.1. Overview

As mentioned in Sec. 6, we follow Günther et al. (2024) for the main assumptions necessary to extract a set of interpretable causal graphs from the observational data sampled from a system with endogenous context variables. First, the strong sufficiency assumption proposed in Günther et al. (2024) ensures that changes in graphs are due to changes in value of the context variable(s), and not due to support problems, i.e., cases where observations of parents within one context all fall into a range of values where a non-linear mechanism (e.g., a thresholded effect) is constant in an otherwise non-trivial value of the parent. We adapt the strong context sufficiency assumption to the time series case in Asm. G.2. The other (weaker) versions of the sufficiency assumption can be adapted similarly. Technical details are given in the next subsection.

Furthermore, our proposed methods rely on the distinction between context descendants and context ancestors. The distinction does not have to be known a-priori, however, the two cases need to be non-overlapping. Therefore, we need to impose acyclicity assumptions: The strong (descriptive) context-acyclicity assumption of Günther et al. (2024) is adapted differently depending on whether we assume persistence (Asm. 1). Asm. 1 on the persistence of contexts essentially forces the context time series to behave like a single variable within a pre-defined time window. The method we propose is a version of the PCMCI algorithm which exploits this property. The method, we propose a version of the PCMCI algorithm. Thus, potentially problems like in the following example may arise. Consider the following Example G.1 where the context variable is in a feedback loop with other system variables.

*Example* G.1. In the ground truth union graph $C_{t-1}$ is a parent of both $X_t$ and $Y_t$ such that for a certain context value $c$ it

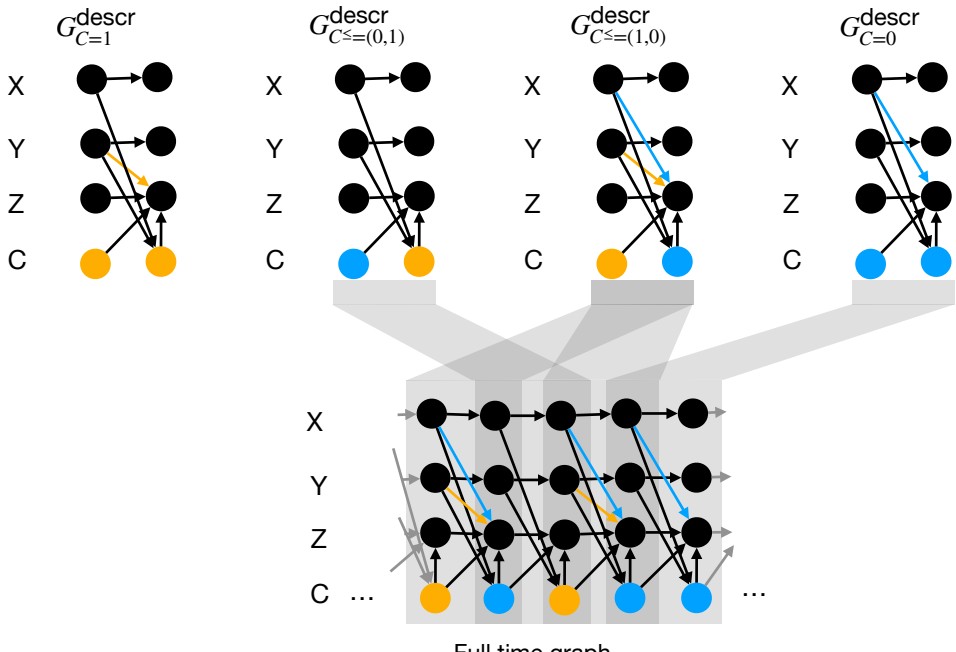

*Figure 8.* Illustration of Ex. F.2. In the top row, the 4 different context window graphs for the SCM given in the example are depicted. The gray areas indicate how they can be combined to infer the full time graph for a specific sequence of context values.

holds that $X_t$ is independent of $Y_t$. However, $X_t$ and $Y_t$ are also parents of $C_t$, thus $X_t \not\perp\!\!\!\perp Y_t | C_{t-1} = c, C_t = c$ by selection bias. Therefore, this context-specific independence will not be found if $C_{t-1}$ and $C_t$ are conditioned on simultaneously.

In the persistent context case, the strong context-acyclicity assumption, Asm. G.6, prohibits such cycles which include the context indicators at the summary graph level. For the case where the persistence assumption is relaxed, and instead at most one true context parent is allowed, the acyclicity assumption Asm. G.7 is formulated using the window graph.

### G.2. Extending the assumptions of Günther et al. (2024) to time series

#### G.2.1. SUFFICIENCY

The following definition extends Günther et al. (2024, Def. C.3).

**Definition G.1.** We call a context indicator $C$ **weakly context-sufficient** for an SCM $M$, if $C_{t-\tau} \notin \mathrm{Pa}^{\mathrm{union}}(Y_t)$ for all $0 \leq \tau \leq \tau_{\max}$ implies $\mathrm{Pa}_{C=c}^{\mathrm{descr}}(Y_t) = \mathrm{Pa}^{\mathrm{union}}(Y_t)$.

The following definition extends Günther et al. (2024, Def. C.1).

**Definition G.2.** We call the indicator time series $C = (C_t)_{t \in \mathcal{T}}$ **strongly time-context-sufficient**, if for an SCM $M$, both

(a) the $f_i$ are continuous, such that:

   (i) If $C_{t-\tau} \notin \mathrm{Pa}(X_t^i)$ for all time steps $t$ and lags $0 \leq \tau \leq \tau_{\max}$ in the mechanism graph, then $f_i$ is injective in each argument, after fixing the other arguments to an arbitrary value.

   (ii) If $C_{t-\tau} \in \mathrm{Pa}(X_t^i)$ for any time lag $\tau$ in the mechanism graph, then $f_i$ is of the form

$$f_i(X, X', C_{t-\tau}, \eta_i) =$$
$$\mathbb{1}(C_{t-\tau} \in A) \times g_i(X, X', C_{t-\tau}, C', \eta_i)$$
$$+ \mathbb{1}(C_{t-\tau} \notin A) \times g_i'(X, C_{t-\tau}, C', \eta_i),$$

   with $A \subset \mathcal{X}_C$, $C'$ are the other context variables, and $g_i$, $g_i'$ injective in each argument, after fixing the other

arguments to an arbitrary value. This could also be generalized to a sum over multiple indicator functions without changing the results.

(b) for every set of variables $V$ not containing $C_{t-\tau}$ for any $\tau$, the distributions $P(V)$, $P(V|\operatorname{do}(C^{\leq} = c\mathbb{1}))$, and $P_M(V|C^{\leq} = \mathbf{c})$ have continuous densities.

Note that in the context-dependent time series setting, the distribution over the endogenous variables $V$ in the time window $[t - \tau_{\max}, t]$ depends on the distribution over their ancestors in the past. This distribution, in turn, is dependent on the distribution over the context values that is observable in data. For instance, deterministically related context values could lead to unexpected behavior, as the following example illustrates.

*Example* G.2. Consider the SCM

$$
\begin{aligned}
X_t &= \mathbb{1}(C_{t-1} = 1)Z_t + \eta_t^X \\
Y_t &= \mathbb{1}(X_t \geq \text{thresh}) + \eta_t^Y, \\
Z_t &= \eta_t^Z, \\
C_t &= \eta_t^C
\end{aligned}
$$

where $C_t$ takes three different values 1, 2, and 3 such that regime 3 only comes after 2, and the other values appear with equal probability. We set $C^{\leq} = (C_{t-1}, C_t)$ to describe the state of the system in context $(3,3)$. When forcing $C^{\leq}$ to take this sequence of values via the intervention $\operatorname{do}(C^{\leq} = (3,3))$, the intervened model is

$$
\begin{aligned}
X_t &= \eta_t^X \\
Y_t &= \mathbb{1}(X_t \geq \text{thresh}) + \eta_t^Y, \\
Z_t &= \eta_t^Z, \\
C_t &= 3.
\end{aligned}
$$

Note that, for example, $X_t$ could take much larger values in a context with $C_{t-1} = 1$ than in the intervened SCM after $\operatorname{do}(C^{\leq} = (3,3))$. The edge $X_t \to Y_t$ is there (by the mechanisms of the intervened SCM) in context $(3,3)$, but since $P(X|C^{\leq} = (3,3))$ potentially never reaches these "large" values of $X_t$ above the threshold $\text{thresh}$, this edge is not visible in context $(3,3)$.

This example shows that in the presence of deterministic relationships between $C_t$ and its past, the support on parents seen in context 3 (in the distribution $P(X|C^{\leq} = (3,3))$) may be smaller than the support in the stationary distribution or in the intervened distribution.

This setting seems interesting in its own right. However it captures a problem different from the one we currently want to study, hence, we exclude it for now by assuming that there is a single, well-defined context-specific graph. We also strengthen this assumption to include that the graph is also well-defined across all distributions we will encounter in observations. A further issue can arise in the finite sample case, especially with persistent contexts: the effective sample count for contexts, i.e., the number of segments between change-points, can be small, such that relevant effects that occur at context boundaries or only in specific context sequences may be difficult to learn. For example, in the case above, if context 3 comes after context 2 in almost all cases, then even on large datasets, the sequence (1,3) may never occur.

The following definition extends Günther et al. (2024, Def. C.12).

**Definition G.3.** For an SCM $M$ at time point $t$, we call a context indicator $C$ **single-graph-sufficient** for $M$, if

$$
G_{C^{\leq}=\mathbf{c}}^{\text{descr}} = G_{C^{\leq}=\mathbf{c}}^{\text{phys}} = G_{C^{\leq}=\mathbf{c}}^{\text{CF}}.
$$

with the graphs $G_{C^{\leq}=\mathbf{c}}^{\text{descr}}, G_{C^{\leq}=\mathbf{c}}^{\text{phys}}, G_{C^{\leq}=\mathbf{c}}^{\text{CF}}$ defined as in Sec. F (also refer to section 4).

*Remark* G.4. In the persistent case, this assumption can be relaxed: it suffices to consider only the graphs for constant contexts over the window, i.e.,

- $\bar{G}_{C=c}^{\text{descr}} := G[\mathcal{F}_{\operatorname{do}(C^{\leq}=c\mathbb{1})}^M, P_M(V|C^{\leq} = c\mathbb{1})]$,

- $G^{\text{phys}}_{C=c} := G[\mathcal{F}^M_{\text{do}(C^\leq=c\mathbb{1})}, P_M(V)]$, and

- $G^{\text{CF}}_{C=c} := G[\mathcal{F}^M_{\text{do}(C^\leq=\mathbf{c})}, P_M(V \mid \text{do}(C^\leq = c\mathbb{1}))]$.

*Remark* G.5. The above definition is not the only possible formalization of the graphical objects in the time series case. Under persistence, the graphica objects could be defined using the stationary distribution conditioned on the entire past of the context variable. Then, it would also be necessry to ensure that the observational distribution resembles the oracle stationary setting, in the sense that the graphs agree.

For an SCM $M$ at time point $t$, we assume that

$$G[\mathcal{F}^M_{\text{do}(C^\leq=c\mathbb{1})}, P_M(V|C^\leq = c\mathbb{1})] = G[\mathcal{F}^M_{\text{do}(C^\leq=c\mathbb{1})}, P_M(V_t|C_s = c \text{ for all } s \leq t)],$$

i.e. the descriptive graph does not depend on $P_M(V|C^\leq = \mathbf{c})$.

We refer to this assumption as **support validity**.

We conjecture that, in the oracle setting, support validity could be replaced by an assumption on the randomness of context values. If the contexts are random enough, e.g. deterministic relations cf. Ex. G.2 between contexts are prohibited, and each combination of context values $\mathbf{c}$ over the time window occurs infinitely often, then presumably no support issues arise. An interesting direction for future work is to determine whether it is sufficient to exclude deterministic relations. It would pe particularly interesting to study the behavior at *context switches*, for example, if certain regions of the system's state space are reached only after prolonged time in a one context, and then a switch occurs, would effects be carried over into the next context?

**Examples of descriptive graphs for different potentially simpler distributions**   In these examples, only fewer (and simpler) distributions have to be considered. This could lead to weaker assumptions or stronger results in these cases.

- The stationary distribution of the system variables for constant context (i.e., non-context-dependent stationary setting) $P_M(V|C_s = c \text{ for all } s \leq t)]$ trivially fulfills support validity.

- $(w, \delta)$-persistent context distribution: If $w$ is large enough, our observable data looks like it is coming from the stationary distribution $P_M(V|C_s = c \text{ for all } s \leq t)$ of the model $M$ with mechanisms $\mathcal{F}^M_{\text{do}(C^\leq=c\mathbb{1})}$.

### G.2.2. ACYCLICITY

Here, we state the two different acyclicity assumptions employed by our two proposed methods, PAC-PCMCI$^+$ and SAC-PCMCI$^+$. We also provide examples of causal systems satisfying these assumptions in Fig. 9.

**Definition G.6** (Strong (descriptive) context-acyclicity for persistence). For all time points $t$, and for each context value $c$ of $C_t$, we assume that the context-specific (c-descriptive) graph $G^{\text{descr}}_{C=c}$ is **strongly context-acyclic**.
That is: $G^{\text{descr}}_{C=c}$ is a DAG, i.e., there are no directed cycles in $G^{\text{descr}}_{C=c}$ for any $c$, and, additionally, no cycles involving any (contemporaneous) ancestors of $C_t$, including $C_t$ itself, are in $G^{\text{union}}$. Furthermore, we require that there are no cycles in the union *summary* graph involving the summary context variable $C$.

**Definition G.7** (Strong (descriptive) context-acyclicity for sparsity). For all time points $t$, and for each context value $\mathbf{c}$ of $C^\leq$, we assume that the context-specific (c-descriptive) graph $G^{\text{descr}}_{C^\leq=\mathbf{c}}$ is **strongly context-acyclic**.
That is: $G^{\text{descr}}_{C=c}$ is a DAG, i.e., there are no directed cycles in $G^{\text{descr}}_{C=c}$ for any $c$, and, additionally, no cycles involving any (contemporaneous) ancestors of $C_t$, including $C_t$ itself, are in $G^{\text{union}}$.

# H. Proofs

We now present the proofs underlying the soundness of the PAC-PCMCI$^+$ and SAC-PCMCI$^+$ methods. We first formulate time series context-specific versions of the Markov property and faithfulness that, together, ensure a one-to-one correspondence between d-separation in the c-descriptive graphs and, potentially context-specific, conditional independence in the population version of the distribution. The specific formulation of these results differs between the persistent and sparse case. We can interpret the two variants of the method as placing different restrictions on the possible set of allowed choices for conditioning sets $Z$.

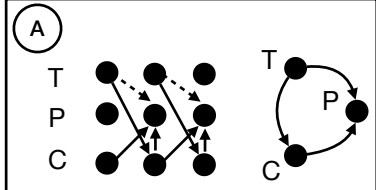 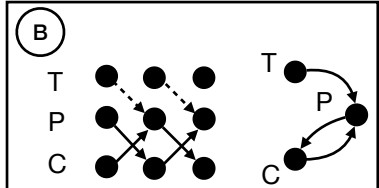 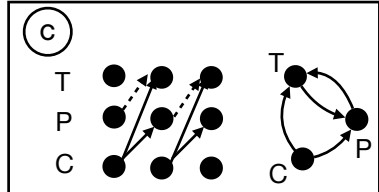

*Figure 9.* Examples of systems where the acyclicity assumption for persistence (Asm. G.6) is satisfied (A and C), and where the acyclicity assumption is not satisfied (B).

The formulation of the Markov condition induces different conditioning sets for each variant. For the persistent case, we search over conditioning sets that fulfill $C^{\leq} \subset Z$ or $C^{\leq} \cap Z = \emptyset$, while for the sparse case, we search $C_{t-\tau} \in Z$ for a single $\tau \leq \tau_{\max}$ or $C^{\leq} \cap Z = \emptyset$. Implicitly, in the persistent case, there are less possible options for $Z$ sets that can d-separate two nodes, thus the conditions we impose in the form of acyclicity assumptions are stronger than in the sparse case.

We start with the Markov and faithfulness lemmas for the persistent case, then prove soundness for PAC-PCMCI$^+$. Afterwards, we state the variants of the Markov and faithfulness lemmas for the sparse case, and prove soundness for SAC-PCMCI$^+$.

The following lemma is a time series adaptation of Günther et al. (2024, Cor. C.2) and ensures that under single graph sufficiency, parent sets that do not include a context variable are indeed not context-dependent.

**Lemma H.1.** *If $M$ is single-graph-sufficient (for $C^{\leq}$), then $M$ is weakly context-sufficient.*

*Proof.* We need to show that if $Y_t$ is not a context variable, i.e., $Y_t \neq C_t$ for all $t$, and no context variable is a parent of $Y_t$, i.e., $C_{t-\tau} \notin \mathrm{Pa}^{\mathrm{union}}(Y_t)$ for all $\tau \leq \tau_{\max}$, then

$$\mathrm{Pa}_{C=c}^{\mathrm{phys}}(Y_t) = \mathrm{Pa}^{\mathrm{union}}(Y_t).$$

(And then by single-graph-sufficiency this is also equal to $\mathrm{Pa}_{C=c}^{\mathrm{descr}}(Y_t)$.)

As with the iid case proved in (Günther et al., 2024), for the time-series extension, by definition of $G^{\mathrm{union}}$ and $G^{\mathrm{phys}}$, the mechanisms $\mathcal{F}$ and $\mathcal{F}_{\mathrm{do}(C^{\leq}=\mathbf{c})}$ differ in $f_{C_{t-\tau}}$ and, due to setting $C^{\leq} = \mathbf{c}$, they differ in those $f_i$ where $C_{t-\tau} \in \mathrm{Pa}^{\mathrm{union}}(X_i)$ for $\tau \leq \tau_{\max}$.

$\square$

## H.1. Soundness of PAC-PCMCI$^+$

**Lemma H.2** (Markov condition). *Adapts Günther et al. (2024, Lemma 4.1). Assume strong context-acyclicity for persistence (Asm. G.6), causal sufficiency, and single-graph-sufficiency (Asm. G.3). If both $X_{t-\tau}, Y_t$ are system variables, $Y_t \notin \mathrm{Anc}_{C=c}^{\mathrm{descr}}(X_{t-\tau})$ (relevant for $\tau = 0$), and $X_{t-\tau}, Y_t$ are not adjacent in $G_{C=c}^{\mathrm{descr}}$, then there is $S \subset \mathrm{Adj}_{C=c}^{\mathrm{descr}}(Y_t)$ (contemporaneous adjacencies) s.t.: If $Y_t$ is neither part of a directed cycle in the union graph (union cycle) nor a child of $C$ at any lag in the union graph (union child), then*

*(a) $X_{t-\tau} \perp\!\!\!\perp Y_t | S, \mathrm{Pa}_t^{\mathrm{union},-}(Y_t), \mathrm{Pa}_{t-\tau}^{\mathrm{union},-}(X_{t-\tau})$,*

*otherwise*

*(b) $X_{t-\tau} \perp\!\!\!\perp Y_t | (S, \mathrm{Pa}_t^{\mathrm{union},-}(Y_t), \mathrm{Pa}_{t-\tau}^{\mathrm{union},-}(X_{t-\tau})) \setminus \mathrm{Pa}^C(X_{t-\tau}, Y_t), C^{\leq} = c\mathbb{1}$.*

*Proof.* The proof is a mostly straightforward adaption of the i.i.d. case, which was treated in (Günther et al., 2024). However, care has to be taken when proving case **(b)**. As in the i.i.d. case we make use of d-separation in the counterfactual graph $G_{C=c}^{\mathrm{CF}}$, we need to ensure that the influence of context variables outside the time window is blocked. This is indeed the case since we are always fixing the values of the parents as well as the noise of $Y_t$ in our counterfactual expression.

For the proof, we consider the two cases corresponding to **(a)** and **(b)**.

**(a)** We use path-blocking arguments in the union graph, and start by showing that $X_{t-\tau}$ and $Y_t$ are also not adjacent in the union graph.

Since $Y_t \notin \mathrm{Anc}^{\mathrm{descr}}_{C=c}(X_{t-\tau})$ and $Y_t$ is not part of a directed union cycle, we get $Y_t \notin \mathrm{Pa}^{\mathrm{union}}(X_{t-\tau})$. Therefore, if a link exists between $X_{t-\tau}$ and $Y_t$ in the union graph, it must be oriented as $X_{t-\tau} \rightarrow Y_t$. Note that this orientation is non-trivial for $\tau = 0$ but the direction is fixed by the hypothesis that $Y_t \notin \mathrm{Anc}^{\mathrm{descr}}_{C=c}(X_{t-\tau})$. However, such an edge is not permitted due to weak context-sufficiency, which is implied by single-graph-sufficiency (see Lemma H.1), and $Y_t$ not being a child of a context variable at any lag in case (a).

Let $Z := \mathrm{Pa}^{\mathrm{union}}(Y_t)$. Since $Y_t$ is not part of a directed cycle in $G^{\mathrm{union}}$, and thus by hypothesis of case (a) $C^{\leq} \cap Z = \emptyset$, we see via standard path-blocking in the union graph that $Z$ in $G^{\mathrm{union}}$ $\sigma$-separates $X_{t-\tau}$ and $Y_t$ which implies the conditional independence $X_{t-\tau} \perp\!\!\!\perp Y_t | Z$ via standard results (Günther et al. (2024), section D.4 cites Peters et al. (2017, Prop. 6.31 (p. 105)), and for cyclic graphs Bongers et al. (2021)).

**(b)** We use path-blocking arguments in the counterfactual graph for $C^{\leq} = c\mathbb{1}$, and call this graph $G^{\mathrm{CF}}_{C=c}$: Like in the iid case, w.l.o.g., $Y_t \notin \mathrm{Anc}^{\mathrm{union}}(C_{t-\tau})$ for any $\tau$, otherwise we would be in case (a). If $Y_t$ were an ancestor of $C_{t-\tau}$, then $Y_t$ could not be part of a directed cycle nor could $C_{t-\tau}$ be a parent, thus ancestor, of $Y_t$, both due to strong context-acyclicity.

Let $Z' := \mathrm{Pa}^{\mathrm{descr}}_{C=c}(Y_t) \cup \mathrm{Pa}^{\mathrm{union},-}_t(Y_t) \cup \mathrm{Pa}^{\mathrm{union},-}_{t-\tau}(X_{t-\tau}) \cup C^{\leq}$. We must show that (i) $\mathrm{Pa}^{\mathrm{CF}}(Y_t) \subset Z'$ and (ii) $Z' \cap \mathrm{Dec}^{\mathrm{CF}}(Y_t) = \emptyset$. This can then allow us to conclude that $Z'$ d-separates $X_{t-\tau}$ from $Y_t$ by standard path-blocking results in $G^{\mathrm{CF}}_{C=c}$.

Statement (i) is clear: by construction of $Z'$, all descriptive parents of $Y_t$ are included. By single graph sufficiency (Asm. G.3), these are equal to the parents in the counterfactual graph.

For statement (ii), note that $\mathrm{Dec}^{\mathrm{CF}}(Y_t) \subset \mathrm{Dec}^{\mathrm{union}}(Y_t)$ by single graph sufficiency and Lemma H.1. We now verify that each component contributing to $Z'$ is disjoint from $\mathrm{Dec}^{\mathrm{CF}}(Y_t)$:

- $\mathrm{Pa}^{\mathrm{descr}}_{C=c}(Y_t) \cap \mathrm{Dec}^{\mathrm{CF}}(Y_t) = \emptyset$: by single-graph-sufficiency, $G^{\mathrm{CF}}_{C=c} = G^{\mathrm{descr}}_{C=c}$, and by weak context-acyclicity, $G^{\mathrm{descr}}_{C=c}$ (hence also $G^{\mathrm{CF}}_{C=c}$) is a DAG, so no parent of $Y_t$ can be its descendant.

- $\mathrm{Pa}^{\mathrm{union},-}_{t-\tau}(X_{t-\tau}) \cap \mathrm{Dec}^{\mathrm{CF}}(Y_t) = \emptyset$: We show the stronger statement that $\mathrm{Pa}^{\mathrm{union},-}_{t-\tau}(X_{t-\tau}) \cap \mathrm{Dec}^{\mathrm{union}}(Y_t) = \emptyset$ which holds since by temporal order, lagged variables cannot be descendants of $Y_t$.

- Finally, $C_t \cap \mathrm{Dec}^{\mathrm{union}}(Y_t) = \emptyset$ holds by the w.l.o.g assumption. This implies (together with lagged variables not being descendants of $Y_t$) that $\mathrm{Pa}^C(X_{t-\tau}, Y_t) \cap \mathrm{Dec}^{\mathrm{CF}}(Y_t) = \emptyset$.

Also note that the counterfactual graph is a causal graph in the usual sense that the Markov property holds automatically, see LemmaD.1 in Günther et al. (2024). Thus, d-separation in the DAG $G^{\mathrm{CF}}_{C=c}$ implies independence

$$X^c_{t-\tau} \perp\!\!\!\perp Y^c_t | Z'^c, \tag{3}$$

writing $V^c$ for a variable that would have been observed in $M_{\mathrm{do}(C^{\leq}=c\mathbb{1})}$ under the same exogenous noises as in $M$.

In the following, we abbreviate

$$\mathrm{Pa}^{\mathrm{union},-}(Y_t, X_{t-\tau}) := \mathrm{Pa}^{\mathrm{union},-}_t(Y_t) \cup \mathrm{Pa}^{\mathrm{union},-}_{t-\tau}(X_{t-\tau}).$$

By definition $C^{\leq} \cup \mathrm{Pa}^{\mathrm{union},-}_t(Y_t, X_{t-\tau}) \subset Z'$, so we can rewrite this as $Z' = Z \cup C^{\leq} \cup \mathrm{Pa}^{\mathrm{union},-}(Y_t, X_{t-\tau})$. Therefore, we rewrite the conditional independence in Eq. (3) as $X^c_{t-\tau} \perp\!\!\!\perp Y^c_t | Z^c, \mathrm{Pa}^{\mathrm{union},-}(Y_t, X_{t-\tau})^c, C^{\leq c}$.

As in the i.i.d. case, we argue using the consistency corollary (Pearl, 2009, cor. 7.3.2 (p. 229)). First note that, by single-graph sufficiency (Asm. G.3), $G^{\mathrm{CF}}_{C=c} = G^{\mathrm{descr}}_{C=c}$. Thus, the parent set $\mathrm{Pa}^{\mathrm{CF}}_{C=c}$ of $Y^c_t$ is the same as $\mathrm{Pa}^{\mathrm{descr}}_{C=c}$ of $Y_t$. Furthermore, we can express $Y^c_t(u) = f_Y(\mathrm{Pa}^{\mathrm{CF}}_{C=c}(Y^c_t)(u) = pa_Y, \eta_Y(u))$. In general, this counterfactual expression is not identifiable. However, using the Pearls consistency assumption, we can identify the expression if we observe the same context value $c$ for $C^{\leq}$ in the model without intervention. More formally, if $C^{\leq}(u) = c\mathbb{1}$, we get $Y^c_t(u) = Y_t(u)$. Since this holds for all $u$, we get that

$$P(Y^c_t | C^{\leq} = c'\mathbb{1}, \mathrm{Pa}^{\mathrm{CF}}_{C=c}(Y^c_t)) = P(Y_t | C^{\leq} = c'\mathbb{1}, \mathrm{Pa}^{\mathrm{descr}}_{C=c'}(Y_t))) \quad \text{when } c' = c. \tag{4}$$

The above implications are easy to interpret when the parent set of $Y_t$ is not context dependent: the intervention on $C^{\leq}$ does not change the model at all. The context-dependent case requires a more detailed explanation.

If the mechanisms in $M$ that lead to $Y_t$ change with $C$, then also $C_\tau \in \mathrm{Pa}^{\mathrm{CF}}_{C=c}(Y^c_t)$ for one or more lags $\tau \leq \tau_{\max}$ by context-sufficiency. We denote this subset of context parents $\mathrm{Pa}^C$. The other system parents are denoted as $\mathrm{Pa}^Y$.

Then, we can express $Y^c_t(u) = f_Y((\mathrm{Pa}^Y(Y^c_t)(u) = pa^Y, \mathrm{Pa}^C(Y^c_t)(u) = c\mathbb{1}, \eta_Y(u))$. By the independence of the noise terms, we can split up $u = (\eta, \eta_Y(u))$. If we first intervene, then apply the same noise as before, and then also condition on $\mathrm{Pa}^Y(Y^c_t)$, we get $Y^c_t(u) = f_Y((\mathrm{Pa}^Y(Y^c_t)(\eta') = pa^Y, \mathrm{Pa}^C(Y^c_t)(u) = c\mathbb{1}, \eta_Y(u))$. Note that, generally, $\eta'$ is different from $\eta$ since the parents are context-dependent, so $\mathrm{Pa}^Y(Y^c_t)(u) \neq \mathrm{Pa}^Y(Y_t)(u)$ (where $\mathrm{Pa}^Y(Y_t)$ are the parents of $Y_t$ in the unintervened model).

Now, if in the unintervened model $C^{\leq}(u) = c\mathbb{1}$, then again by Pearls consistency assumption, $Y^c_t(u) = f_Y((\mathrm{Pa}^Y(Y^c_t)(\eta') = pa^Y, \mathrm{Pa}^C(Y^c_t)(u) = c\mathbb{1}, \eta_Y(u)) = f_Y((\mathrm{Pa}^Y(Y_t)(\eta) = pa^Y, \mathrm{Pa}^C(Y_t)(u) = c\mathbb{1}, \eta_Y(u)) = Y_t(u)$.

We use Eq. (4) to conclude

$$X^c_{t-\tau} \perp\!\!\!\perp Y^c_t | Z^c, \mathrm{Pa}^{\mathrm{union},-}(Y_t, X_{t-\tau})^c, C^{\leq c}$$
$$\Rightarrow \quad X^c_{t-\tau} \perp\!\!\!\perp Y^c_t | Z^c, \mathrm{Pa}^{\mathrm{union},-}(Y_t, X_{t-\tau})^c, C^{\leq c} = c\mathbb{1}$$
$$\Rightarrow \quad X_{t-\tau} \perp\!\!\!\perp Y_t | Z, \mathrm{Pa}^{\mathrm{union},-}(Y_t, X_{t-\tau}), C^{\leq} = c\mathbb{1}.$$

$\square$

**Lemma H.3** (Faithfulness under persistence). *Adapts Günther et al. (2024, Lemma 4.2). Given a context value $c$, assume that both $P_M$ is faithful to $G^{\mathrm{descr}}_{C=c}$, and $P_M(\ldots | C^{\leq} = c\mathbb{1})$ is faithful to $\bar{G}^{\mathrm{descr}}_{C=c}$. Then:*

$$\exists Z \subset \mathrm{Adj}^{\mathrm{descr}}_{C=c} \text{ s.t. } \left\{ \begin{array}{l} X_{t-\tau} \perp\!\!\!\perp Y_t | Z \quad \text{or} \\ X_{t-\tau}, Y_t \neq C_s \text{ for any } s \text{ and} \\ X_{t-\tau} \perp\!\!\!\perp Y_t | Z, C^{\leq} = c\mathbb{1} \end{array} \right\} \quad \Rightarrow \quad X_{t-\tau} \text{ and } Y_t \text{ are not adjacent in } G^{\mathrm{descr}}_{C=c}.$$

*Proof.* First, note that, by definition, $X \not\perp\!\!\!\perp_{P(\ldots|C=c)} Y | Z$ is the same as $X \not\perp\!\!\!\perp_P Y | Z, C = c$.

We omit the time subscript for readability. By the faithfulness assumption of $P_M$, $X \perp\!\!\!\perp_{P_M} Y | Z$ implies the d-separation $X \not\perp\!\!\!\perp_{G^{\mathrm{descr}}_{C=c}} Y | Z$. If $X_{t-\tau} \perp\!\!\!\perp Y_t | Z, C^{\leq} = c\mathbb{1}$, we can apply the second assumption of faithfulness above. This gives us the d-separation $X_{t-\tau} \not\perp\!\!\!\perp_{G^{\mathrm{descr}}_{C=c}} Y_t | Z$. $\square$

The soundness proof of PCMCI$^+$ (Runge, 2020) uses that PC1 returns a superset of the lagged parents for each variable (Runge, 2020, Lemma S1). This is also needed to prove the soundness of both PAC-PCMCI$^+$ and SAC-PCMCI$^+$. In the context-dependent setting, the relevant parents are those of the *union graph*. We adapt this lemma under our assumptions: applying PC1 to context-dependent data yields a superset of the union parents for every variable. These supersets provide the conditioning sets used in the subsequent MCI phase.

**Lemma H.4.** *Adapts Runge (2020, Lemma S1). Under $C$-faithfulness, in the oracle case, the PC1 algorithm (see Runge et al. (2019)) returns a superset $\hat{\mathcal{B}}^{\mathrm{union},-}_t(X_t)$ of lagged union parents, i.e., $\mathrm{Pa}^{\mathrm{union},-}_t(X_t) \subset \hat{\mathcal{B}}^{\mathrm{union},-}_t(X_t)$ for all $X_t$.*

*Proof.* First, note that union faithfulness is implied by C-faithfulness, since $X_{t-\tau} \perp\!\!\!\perp Y_t | Z$ implies $X_{t-\tau}$ and $Y_t$ are not adjacent in $G^{\mathrm{descr}}_{C=c}$ for all $c$, and thus they are not adjacent in the union graph.

We must show that, for any $(X_{t-\tau}, Y_t)$ with $\tau > 0$, $X_{t-\tau} \notin \hat{\mathcal{B}}^{\mathrm{union},-}_t(Y_t)$ implies $X_{t-\tau} \notin \mathrm{Pa}^{\mathrm{union},-}_t(Y_t)$.

The PC1 algorithm removes $X_{t-\tau}$ from $\hat{\mathcal{B}}^{\mathrm{union},-}_t(Y_t)$ iff $X_{t-\tau} \perp\!\!\!\perp Y_t | Z$ for some $Z \subset \hat{\mathcal{B}}^{\mathrm{union},-}_t(Y_t) \setminus \{X_{t-\tau}\}$ in the iterative CI tests.

Then, union faithfulness directly implies that $X_{t-\tau}$ is not adjacent to $Y_t$ and, in particular, $X_{t-\tau} \notin \mathrm{Pa}^{\mathrm{union},-}_t(Y_t)$.

$\square$

The following lemma ensures that additional union parents which are part of the superset discovered during the PC1 phase can be included in the conditioning set without affecting the (in-)dependence relation between $X_{t-\tau}$ and $Y_t$.

**Lemma H.5.** *Let $Z^C \subset C^\leq$ be a not necessarily strict subset of the context variables. For any $(X_{t-\tau}, Y_t)$ with $\tau > 0$, and any sequence of context values* **c**, *we have: If*

*(a)* $X_{t-\tau} \perp\!\!\!\perp Y_t | \operatorname{Pa}_t^{\mathrm{descr}}(Y_t), \operatorname{Pa}_t^{\mathrm{union},-}(Y_t), \operatorname{Pa}_{t-\tau}^{\mathrm{union},-}(X_{t-\tau})$ *or*

*(b)* $X_{t-\tau} \perp\!\!\!\perp Y_t | (\operatorname{Pa}_t^{\mathrm{descr}}(Y_t), \operatorname{Pa}_t^{\mathrm{union},-}(Y_t), \operatorname{Pa}_{t-\tau}^{\mathrm{union},-}(X_{t-\tau})) \setminus \operatorname{Pa}^C(X_{t-\tau}, Y_t), Z^C = \mathbf{c}$

*then, respectively, also*

*(a')* $X_{t-\tau} \perp\!\!\!\perp Y_t | \operatorname{Pa}_t^{\mathrm{descr}}(Y_t), \hat{\mathcal{B}}_t^{\mathrm{union},-}(Y_t) \setminus \{X_{t-\tau}\}, \hat{\mathcal{B}}_t^{\mathrm{union},-}(X_{t-\tau})$, *or*

*(b')* $X_{t-\tau} \perp\!\!\!\perp Y_t | (\operatorname{Pa}_t^{\mathrm{descr}}(Y_t), \hat{\mathcal{B}}_t^{\mathrm{union},-}(Y_t), \hat{\mathcal{B}}_{t-\tau}^{\mathrm{union},-}(X_{t-\tau})) \setminus \operatorname{Pa}^C(X_{t-\tau}, Y_t), Z^C = \mathbf{c}$,

*where $\hat{\mathcal{B}}_t^{\mathrm{union},-}(X)$ denotes the superset of lagged union parents of $X$ discovered during PC1, and $\operatorname{Pa}^C(X)$ denotes the (union) context parents of $X$.*

*Proof.* We first show that the additional union parents included in the superset discovered during the PC1 phase can be included into the conditioning set without affecting the (in-)dependence relation between $X_{t-\tau}$ and $Y_t$. Throughout, we abbreviate the conditioning set as

$$\mathcal{B}(X_{t-\tau}, Y_t) := \operatorname{Pa}_t^{\mathrm{descr}}(Y_t) \cup \operatorname{Pa}_t^{\mathrm{union},-}(Y_t) \cup \operatorname{Pa}_{t-\tau}^{\mathrm{union},-}(X_{t-\tau}),$$

and define

$$W_t^- := \left( \left( \hat{\mathcal{B}}_t^{\mathrm{union},-}(Y_t) \setminus \{X_{t-\tau}\} \right) \cup \hat{\mathcal{B}}_t^{\mathrm{union},-}(X_{t-\tau}) \right) \setminus \mathcal{B}(X_{t-\tau}, Y_t),$$

i.e., the set of all additional variables to the true parents of $X_{t-\tau}$ and $Y_t$.

We start by showing (a) $\implies$ (a').
By definition, $W_t^-$ does not contain parents of $Y_t$, and since all variables in $W_t^-$ are lagged, it also does not contain any descendants of $Y_t$. By assumption, if $X_{t-\tau}$ is not a parent of $Y_t$, it also is not a descendant of $Y_t$. Therefore, via path blocking arguments in the union graph and by Lemma H.2, we obtain that in case (i) and (ii),

$$(X_{t-\tau}, W_t^-) \perp\!\!\!\perp Y_t | \mathcal{B}(X_{t-\tau}, Y_t).$$

Thus, by the contraction and weak union properties, it holds that

$$X_{t-\tau} \perp\!\!\!\perp Y_t | \mathcal{B}(X_{t-\tau}, Y_t), W_t^-$$

which is equivalent to

$$X_{t-\tau} \perp\!\!\!\perp Y_t | \operatorname{Pa}_t^{\mathrm{descr}}(Y_t), \hat{\mathcal{B}}_t^{\mathrm{union},-}(Y_t) \setminus \{X_{t-\tau}\}, \hat{\mathcal{B}}_t^{\mathrm{union},-}(X_{t-\tau}), \tag{5}$$

which we wanted to show for case (a).

In case (b), Lemma H.2 implies $X_{t-\tau} \perp\!\!\!\perp Y_t | \mathcal{B}(X_{t-\tau, Y_t}) \setminus \operatorname{Pa}^C(X_{t-\tau}, Y_t), C^\leq = c\mathbb{1}$. By the same arguments as above, with path-blocking in the counterfactual graph $G_{C=c}^{\mathrm{CF}}$, we obtain

$$(X_{t-\tau}, W_t^-) \perp\!\!\!\perp Y_t | \mathcal{B}(X_{t-\tau}, Y_t) \setminus (\{X_{t-\tau}\} \cup \operatorname{Pa}^C(X_{t-\tau}, Y_t)), C^\leq = c\mathbb{1}.$$

This implies that we can safely add lagged variables to the conditioning set, also in the context-specific case.

$\square$

**Theorem 6.1** (Soundness of PAC-PCMCI$^+$). *Under suitable assumptions (persistence Asm. 1, single graph sufficiency Asm. G.2, strong context-acyclicity for persistence Asm. G.6, with minimal edge flips), the oracle versions of PAC-PCMCI$^+$, Alg. 1, recovers the unoriented edges of $G_{C=c}^{\mathrm{descr}}$ involving at least one variable in $\mathbf{X}_t$.*

*Proof.* We will closely follow the soundness proof of PCMCI$^+$ given in Runge (2020).

We first introduce some notation. The skeleton of the ground truth descriptive time series graph with the added context variable edges from the union graph is denoted by $G_{C=c}^{\text{descr}}$. Furthermore, the skeleton of the time series graph output of the algorithm is denoted by $\mathcal{G}_{alg}$. Again, we abbreviate the conditioning set as

$$\mathcal{B}(X_{t-\tau}, Y_t) := \text{Pa}_t^{\text{descr}}(Y_t) \ \cup \ \text{Pa}_t^{\text{union},-}(Y_t) \ \cup \ \text{Pa}_{t-\tau}^{\text{union},-}(X_{t-\tau}).$$

We focus on edges involving at least one variable in $\mathbf{X}_t$. Thus, w.l.o.g., we fix $Y$ to be a contemporaneous variable $Y_t$ and $X_{t-\tau}$ is not a descendant of $Y_t$.

We need to show that for any $(X_{t-\tau}, Y_t)$ with $\tau > 0$, we have

(1) $X_{t-\tau} - Y_t \notin \mathcal{G}_{alg}$ implies $X_{t-\tau} - Y_t \notin G_{C=c}^{\text{descr}}$

(2) $X_{t-\tau} - Y_t \notin G_{C=c}^{\text{descr}}$ implies $X_{t-\tau} - Y_t \notin \mathcal{G}_{alg}$

We start by proving claim (1). By construction, the algorithm removes an adjacency $X_{t-\tau} - Y_t$ only upon observing a conditional independence $X_{t-\tau} \perp\!\!\!\perp Y_t \mid Z$ or $X_{t-\tau} \perp\!\!\!\perp Y_t \mid Z, C^{\leq} = c\mathbb{1}$ among the tested sets $Z$. By $C$-faithfulness (Lemma H.3), any such empirical CI implies the absence of an edge between $X_{t-\tau}$ and $Y_t$ in $G_{C=c}^{\text{descr}}$.

Now, we show claim (2). Let $X_{t-\tau} - Y_t \notin G_{C=c}^{\text{descr}}$. By the causal Markov condition (Lemma H.2), this implies

(a) $X_{t-\tau} \perp\!\!\!\perp Y_t | \text{Pa}_t^{\text{descr}}(Y_t), \text{Pa}_t^{\text{union},-}(Y_t), \text{Pa}_{t-\tau}^{\text{union},-}(X_{t-\tau})$ if $Y_t$ is neither part of a union cycle nor a union child of $C$ at any lag or

(b) $X_{t-\tau} \perp\!\!\!\perp Y_t | (\text{Pa}_t^{\text{descr}}(Y_t), \text{Pa}_t^{\text{union},-}(Y_t), \text{Pa}_{t-\tau}^{\text{union},-}(X_{t-\tau})) \setminus \text{Pa}^C(X_{t-\tau}, Y_t), C^{\leq} = c\mathbb{1}$ otherwise.

Furthermore, recall that in the MCI phase, the algorithm tests all adjacencies and the superset of lagged parents (by Lemma H.4), thus iteratively testing

$$X_{t-\tau} \perp\!\!\!\perp Y_t | \mathcal{S}, (\hat{\mathcal{B}}_t^{\text{union},-}(Y_t) \setminus \{X_{t-\tau}\}, \hat{\mathcal{B}}_t^{\text{union},-}(X_{t-\tau})), \quad \text{if } C_{t-\tau} \text{ not in the conditioning set, or}$$
$$X_{t-\tau} \perp\!\!\!\perp Y_t | \mathcal{S}, (\hat{\mathcal{B}}_t^{\text{union},-}(Y_t) \setminus \{X_{t-\tau}\}, \hat{\mathcal{B}}_t^{\text{union},-}(X_{t-\tau})), C^{\leq} = c\mathbb{1}, \quad \text{else} \tag{6}$$

for all $\mathcal{S} \in \text{Adj}(Y_t)$ subsets of the estimated contemporaneous adjacencies.

We distinguish three mutually exclusive cases.

**Case 1:** We assume (a) applies and $C_{t-\tau} \notin \text{Pa}_t^{\text{descr}}(Y_t) \cup \hat{\mathcal{B}}_{t-\tau}^{\text{union},-}(X_{t-\tau}) \cup \hat{\mathcal{B}}_t^{\text{union},-}(Y_t)$ for all lags $\tau \leq \tau_{\max}$.

Then, by Lemma H.5, also

$$X_{t-\tau} \perp\!\!\!\perp Y_t | \text{Pa}_t^{\text{descr}}(Y_t), (\hat{\mathcal{B}}_t^{\text{union},-}(Y_t) \setminus \{X_{t-\tau}\}, \hat{\mathcal{B}}_t^{\text{union},-}(X_{t-\tau})).$$

By Lemma H.4, we know $\text{Pa}^{\text{union},-}(Y_t) \subseteq \hat{\mathcal{B}}(Y_t)$ and $\text{Pa}^{\text{union},-}(X_{t-\tau}) \subseteq \hat{\mathcal{B}}(X_{t-\tau})$. Therefore, we only must show that $\text{Pa}_{C=c}^{\text{descr}}(Y_t) \subseteq \text{Adj}(Y_t)$ to prove that, at some point during runtime, the algorithm tests this relationship, see Eq. 6: If $C_{t-\tau} \notin \mathcal{B}(X_{t-\tau}, Y_t)$ for all lag $\tau \leq \tau_{\max}$, then $\text{Pa}^{\text{union}}(Y_t) = \text{Pa}_{C=c}^{\text{descr}}(Y_t)$. The algorithm removes the link between $X_{t-\tau}$ and $Y_t$ iff one of the independencies in Eq. 6 holds for some $\mathcal{S}$. Then, union faithfulness directly implies that $X_{t-\tau}$ and $Y_t$ are not adjacent in $G^{\text{union}}$.

**Case 2.1:** We assume (a) applies and $C_{t-\tau} \in \text{Pa}_t^{\text{descr}}(Y_t) \cup \hat{\mathcal{B}}_{t-\tau}^{\text{union},-}(X_{t-\tau}) \cup \hat{\mathcal{B}}_t^{\text{union},-}(Y_t)$ for some or multiple lags $\tau \leq \tau_{\max}$.

Then, as above by Lemma H.5, also

$$X_{t-\tau} \perp\!\!\!\perp Y_t | \text{Pa}_t^{\text{descr}}(Y_t), (\hat{\mathcal{B}}_t^{\text{union},-}(Y_t) \setminus \{X_{t-\tau}\}, \hat{\mathcal{B}}_t^{\text{union},-}(X_{t-\tau})).$$

(*) First, we consider the sub-case where $C_{t-\tau} \in \mathcal{B}(X_{t-\tau}, Y_t)$ for at least one lag $\tau \leq \tau_{\max}$. In other words, there is at least one true context parent in the conditioning set.

Denote the set of context variables that is a subset of $\mathcal{B}(X_{t-\tau}, Y_t)$ by $\mathrm{Pa}^C(X_{t-\tau}, Y_t)$.

We can also formulate (*) as $\mathrm{Pa}^C(X_{t-\tau}, Y_t)$ is not empty. By the strong context acyclicity under persistence Asm. G.6, we have that, for $\tau$ and any other lag, $s < \tau_{\max}$ $C_{t-s}$ cannot be a child of $Y_t$. Define $Z^C := C^{\leq} \setminus \mathrm{Pa}^C(X_{t-\tau}, Y_t)$. Since $Z^C$ also does not contain any parents of $Y_t$ by definition, we obtain via path blocking in the union graph

$$(X_{t-\tau}, Z^C) \perp\!\!\!\perp Y_t | (\mathrm{Pa}_t^{\mathrm{descr}}(Y_t), (\hat{\mathcal{B}}_t^{\mathrm{union},-}(Y_t) \setminus \{X_{t-\tau}\}, \hat{\mathcal{B}}_t^{\mathrm{union},-}(X_{t-\tau}))), \mathrm{Pa}^C(X_{t-\tau}, Y_t).$$

And thus, by the weak union property,

$$X_{t-\tau} \perp\!\!\!\perp Y_t | (\mathrm{Pa}_t^{\mathrm{descr}}(Y_t), (\hat{\mathcal{B}}_t^{\mathrm{union},-}(Y_t) \setminus \{X_{t-\tau}\}, \hat{\mathcal{B}}_t^{\mathrm{union},-}(X_{t-\tau}))), C^{\leq}.$$

This implies for all $c$

$$X_{t-\tau} \perp\!\!\!\perp Y_t | (\mathrm{Pa}_t^{\mathrm{descr}}(Y_t), (\hat{\mathcal{B}}_t^{\mathrm{union},-}(Y_t) \setminus \{X_{t-\tau}\}, \hat{\mathcal{B}}_t^{\mathrm{union},-}(X_{t-\tau}))) \setminus \mathrm{Pa}^C(X_{t-\tau}, Y_t), C^{\leq} = c\mathbb{1}.$$

which is tested at some point during runtime of our algorithm (see Eq. 6).

**Case 2.2:** Now, we move on to the second sub-case where (a) applies and $C_{t-\tau} \in \mathrm{Pa}_t^{\mathrm{descr}}(Y_t) \cup \hat{\mathcal{B}}_{t-\tau}^{\mathrm{union},-}(X_{t-\tau}) \cup \hat{\mathcal{B}}_t^{\mathrm{union},-}(Y_t)$ for some or multiple lags $\tau \leq \tau_{\max}$, but $C_{t-\tau} \notin \mathcal{B}(X_{t-\tau}, Y_t)$ for all $\tau \leq \tau_{\max}$.

In this case, $C_{t-\tau}$ is always part of the superset of lagged parents and not of the contemporaneous adjacencies, i.e. $\tau > 0$.

By hypothesis, since (a) holds, and $Y_t$ does not have a context parent, we get via path-blocking in the union graph

$$X_{t-\tau} \perp\!\!\!\perp Y_t | \mathcal{S}, (\mathcal{B}_t^{\mathrm{union},-}(X_{t-\tau}, Y_t) \setminus (\{X_{t-\tau}\} \cup C^{\leq}).$$

This, in turn, implies

$$X_{t-\tau} \perp\!\!\!\perp Y_t | \mathcal{S}, (\hat{\mathcal{B}}_t^{\mathrm{union},-}(Y_t) \setminus \{X_{t-\tau}\}, \hat{\mathcal{B}}_t^{\mathrm{union},-}(X_{t-\tau})) \setminus C^{\leq}.$$

By hypothesis, since there is a context variable in the superset of lagged parents, but not in the subset of contemporaneous adjacencies $\mathcal{S}$, we additionally test this relationship.

**Case 3:** We assume (a) does not apply. Now, we must show that if (a) doesn't apply, then we test the context-specific relationship given in (b).

If (a) does not hold, then either

- $Y_t$ is a union child of $C$ at some lag, or

- $Y_t$ is part of a union cycle.

In the first case, the context parent of $Y_t$ will be included in the conditioning set at some point, which means that the context-specific relationship given in (b) is tested using supersets of the lagged parents (which is allowed by Lemma H.5).

If $Y_t$ is part of a union cycle, then, by the minimal edge flips assumption, it is part of a minimal cycle. Since both nodes in a minimal cycle are children of the context variable at possibly different lags by weak context-sufficiency, this means that $Y_t$ is also a union child of $C$ at some lag, and we can argue like above.

$\square$

### H.2. Soundness of SAC-PCMCI$^+$

Throughout the proof, we use the notation for a fixed sequence of context values over a time window $\mathbf{c}$ for a subset of context variables $Z^C \subset C^{\leq}$: $Z^C = \mathbf{c}$, which should be understood as fixing the values of the variables in $Z^C$ to the respective value in $\mathbf{c}$ corresponding to their lag.

**Lemma H.6** (Markov condition under sparsity). *Assume strong context-acyclicity for sparsity (Asm. G.7), causal sufficiency, and single-graph-sufficiency. If both $X_{t-\tau}, Y_t$ are not context variables and both have at most one (possibly different from each other) context parent, with $Y_t \notin \text{Anc}_{C=c}^{\text{descr}}(X_{t-\tau})$ (relevant for $\tau = 0$, this just fixes notation), are not adjacent in $G_{C=c}^{\text{descr}}$, then there is $Z \subset \text{Adj}_{C=c}^{\text{descr}}(Y_t)$ (contemporaneous adjacencies) such that: If $Y_t$ is neither part of a directed cycle in the union graph nor a child of $C$ at any lag in the union graph, then*

*(a) $X_{t-\tau} \perp\!\!\!\perp Y_t | Z, \text{Pa}_t^{\text{union},-}(Y_t), \text{Pa}_{t-\tau}^{\text{union},-}(X_{t-\tau})$,*

*otherwise*

*(b) $X_{t-\tau} \perp\!\!\!\perp Y_t | (Z, \text{Pa}_t^{\text{union},-}(Y_t), \text{Pa}_{t-\tau}^{\text{union},-}(X_{t-\tau})) \setminus \text{Pa}^C(X_{t-\tau}, Y_t), \text{Pa}^C(X_{t-\tau}, Y_t) = \mathbf{c}$.*

*Proof.* **(a)** Case (a) can be proven as for the persistent case.

**(b)** We use path-blocking arguments in the counterfactual graph for $C^{\leq} = \mathbf{c}$. First, note that if there is no context parent of $Y_t$, then we are in case (a). Now, fix the name $C_{t-s}$ for the one context parent of $Y_t$. As in the i.i.d. case, w.l.o.g., $Y_t \notin \text{Anc}^{\text{union}}(C_{t-s})$, otherwise we would be in case (a). If $Y_t$ were an ancestor of $C_{t-s}$, then $Y_t$ could not be part of a directed cycle, nor could $C_{t-s}$ be a parent, thus ancestor, of $Y_t$, both due to strong context-acyclicity (or time order if $s > 0$). Note that, here, the adapted version of acyclicity for sparsity and the assumption that there is at most one context parent of $Y_t$ are needed.

Let $Z' := \text{Pa}_{C=c}^{\text{descr}}(Y_t) \cup \text{Pa}_t^{\text{union},-}(Y_t) \cup \text{Pa}_{t-\tau}^{\text{union},-}(X_{t-\tau}) \cup \text{Pa}^C(X_{t-\tau}, Y_t)$. We need to show that (i) $\text{Pa}^{\text{CF}}(Y_t) \subset Z'$ and (ii) $Z' \cap \text{Dec}^{\text{CF}}(Y_t) = \emptyset$. This can then allow us to conclude that $Z'$ d-separates $X_{t-\tau}$ from $Y_t$ by standard path-blocking results in $G_{C=c}^{\text{CF}}$.

This follows in the same way as in the persistent case.

Also note that the counterfactual graph is a causal graph in the sense that the Markov property holds, see Lemma D.1 in Günther et al. (2024). Thus, d-separation in the DAG $G_{C=c}^{\text{CF}}$ implies independence

$$X_{t-\tau}^c \perp\!\!\!\perp Y_t^c | Z'^c, \tag{7}$$

writing $V^c$ for a variable that would have been observed in $M_{\text{do}(C^{\leq}=\mathbf{c})}$ under the same exogenous noises as in $M$.

In the following, we abbreviate $\text{Pa}^{\text{union},-}(Y_t, X_{t-\tau}) := \text{Pa}_t^{\text{union},-}(Y_t) \cup \text{Pa}_{t-\tau}^{\text{union},-}(X_{t-\tau})$. By definition, $\text{Pa}^C(X_{t-\tau}, Y_t) \cup \text{Pa}_t^{\text{union},-}(Y_t, X_{t-\tau}) \subset Z'$, thus, we can rewrite this as $Z' = Z \cup \text{Pa}^C(X_{t-\tau}, Y_t) \cup \text{Pa}^{\text{union},-}(Y_t, X_{t-\tau})$. Therefore, we rewrite the conditional independence in Eq. 7 as $X_{t-\tau}^c \perp\!\!\!\perp Y_t^c | Z^c, \text{Pa}^{\text{union},-}(Y_t, X_{t-\tau})^c, \text{Pa}^C(X_{t-\tau}, Y_t)^c$. As in the i.i.d. case, and in the persistent case, we argue using the consistency corollary (Pearl, 2009, cor. 7.3.2 (p. 229)), which gives us identifiability of $P(X_t^c | \text{Pa}^Y(Y_t^c), \text{Pa}^C(Y_t^c) = \mathbf{c}')$ (equivalent to $P(X_t | \text{Pa}^Y(Y_t), \text{Pa}^C(Y_t) = \mathbf{c})$) when $\mathbf{c}' = \mathbf{c}$ under the consistency assumption).

$$X_{t-\tau}^c \perp\!\!\!\perp Y_t^c | Z^c, \text{Pa}^{\text{union},-}(Y_t, X_{t-\tau})^c, \text{Pa}^C(X_{t-\tau}, Y_t)^c$$
$$\Rightarrow \quad X_{t-\tau}^c \perp\!\!\!\perp Y_t^c | Z^c, \text{Pa}^{\text{union},-}(Y_t, X_{t-\tau})^c, \text{Pa}^C(X_{t-\tau}, Y_t)^c = \mathbf{c}$$
$$\Rightarrow \quad X_{t-\tau} \perp\!\!\!\perp Y_t | Z, \text{Pa}^{\text{union},-}(Y_t, X_{t-\tau}), \text{Pa}^C(X_{t-\tau}, Y_t) = \mathbf{c}.$$

$\square$

In the sparse case, we require that independence implies d-separation for more conditioning sets than under the persistence assumption. In this sense, we need a stronger faithfulness assumption than in the sparse setting.

**Lemma H.7** (Faithfulness under sparsity). *This lemma adapts Lemma 4.2 in Günther et al. (2024). Given $\mathbf{c}$, assume both $P_M$ is faithful to $G_{C=c}^{\text{descr}}$, and $P_M(\ldots | Z^C = \mathbf{c})$ is faithful to $\bar{G}_{C=\mathbf{c}}^{\text{descr}}$ for all subsets $Z^C \subset C^{\leq}$. Then:*

$$\exists Z \subset \text{Adj}_{C=c}^{\text{descr}} \text{ s.t. } \left\{ \begin{array}{c} X_{t-\tau} \perp\!\!\!\perp Y_t | Z \quad \text{or} \\ X_{t-\tau}, Y_t \neq C_s \text{ for any } s \text{ and} \\ X_{t-\tau} \perp\!\!\!\perp Y_t | Z, \text{Pa}^C(X_{t-\tau}, Y_t) = \mathbf{c} \end{array} \right\} \quad \Rightarrow \quad X_{t-\tau} \text{ and } Y_t \text{ are not adjacent in } G_{C=c}^{\text{descr}}.$$

*Proof.* First, note that, by definition, $X \not\perp\!\!\!\perp_{P(\ldots|C=c)} Y|Z$ is the same as $X \not\perp\!\!\!\perp_P Y|Z, C = c$. We omit the time subscript for readability.

By the faithfulness assumption of $P_M$, $X \perp\!\!\!\perp_{P_M} Y|Z$ implies that they are d-separated, $X \not\perp\!\!\!\perp_{G^{\text{descr}}_{C=c}} Y|Z$.

If $X_{t-\tau} \perp\!\!\!\perp Y_t|Z, \text{Pa}^C(X_{t-\tau}, Y_t) = \mathbf{c}$, we can apply the second faithfulness assumption with a suitable $Z^C$ above. This gives us the d-separation $X_{t-\tau} \not\perp\!\!\!\perp_{G^{\text{descr}}_{C=\mathbf{c}}} Y_t|Z$. $\qquad\square$

Now, we can move on to prove the soundness of SAC-PCMCI$^+$.

**Theorem H.8** (Soundness of SAC-PCMCI$^+$). *Under suitable assumptions (sparsity 2, sufficiency G.2, strong context-acyclicity for sparsity G.7, minimal edge flips), in the oracle case, the algorithm 2 recovers the skeleton of $G^{\text{descr}}_{C^{\leq}=\mathbf{c}}$.*

*Proof.* The proof strategy is the same as in the persistent case, and we highlight here the points and details that diverge.

We first introduce some notation. The skeleton of the ground truth descriptive time series graph with added context variable-edges from the union graph is denoted by $G^{\text{descr}}_{C=c}$. Furthermore, the skeleton of the time series graph output of the algorithm is denoted by $\mathcal{G}_{alg}$. We focus on edges involving at least one variable in the present. Thus, w.l.o.g., we fix $Y$ to be a contemporaneous variable $Y_t$ and $X_{t-\tau}$ is not a descendant of $Y_t$.

We need to show that, for any $(X_{t-\tau}, Y_t)$ with $\tau > 0$, we have

(1) $X_{t-\tau} - Y_t \notin \mathcal{G}_{alg}$ implies $X_{t-\tau} - Y_t \notin G^{\text{descr}}_{C=c}$, and

(2) $X_{t-\tau} - Y_t \notin G^{\text{descr}}_{C=c}$ implies $X_{t-\tau} - Y_t \notin \mathcal{G}_{alg}$.

Claim (1) can be shown in the same way as for the persistent version.

For claim (2), we use the same notation as above. Recall that $\hat{\mathcal{B}}^{\text{union},-}_t$ denotes the superset of lagged union parents of $X$ discovered during PC1 and

$$\mathcal{B}(X_{t-\tau}, Y_t) := \text{Pa}^{\text{descr}}_t(Y_t) \cup \text{Pa}^{\text{union},-}_t(Y_t) \cup \text{Pa}^{\text{union},-}_{t-\tau}(X_{t-\tau}).$$

We also use the Markov condition (Lemma H.6) which implies

(a) $X_{t-\tau} \perp\!\!\!\perp Y_t | \text{Pa}^{\text{descr}}_t(Y_t), \text{Pa}^{\text{union},-}_t(Y_t), \text{Pa}^{\text{union},-}_{t-\tau}(X_{t-\tau})$ if $Y_t$ is neither part of a union cycle nor a union child of $C$ at any lag or

(b) $X_{t-\tau} \perp\!\!\!\perp Y_t | (\text{Pa}^{\text{descr}}_t(Y_t), \text{Pa}^{\text{union},-}_t(Y_t), \text{Pa}^{\text{union},-}_{t-\tau}(X_{t-\tau})) \setminus \text{Pa}^C(X_{t-\tau}, Y_t), \text{Pa}^C(Y_t) = \mathbf{c}$ otherwise.

We distinguish three mutually exclusive cases.

**Case 1:** We assume (a) applies and $C_{t-\tau} \notin \text{Pa}^{\text{descr}}_t(Y_t) \cup \hat{\mathcal{B}}^{\text{union},-}_{t-\tau}(X_{t-\tau}) \cup \hat{\mathcal{B}}^{\text{union},-}_t(Y_t)$ for all lag $\tau \leq \tau_{\max}$.

In this case, the claim that the algorithm removes the link between $X_{t-\tau}$ and $Y_t$ can be shown in the same way as under persistence.

**Case 2:** We assume (a) applies and $C_{t-\tau} \in \text{Pa}^{\text{descr}} \text{Pa}_t(Y_t) \cup \hat{\mathcal{B}}^{\text{union},-}_{t-\tau}(X_{t-\tau}) \cup \hat{\mathcal{B}}^{\text{union},-}_t(Y_t)$ for some lag $\tau \leq \tau_{\max}$.

Denote the set of context variables that is a subset of $\hat{\mathcal{B}}^{\text{union},-}_t(Y_t) \cup \hat{\mathcal{B}}^{\text{union},-}_t(X_{t-\tau})$ by $Z^C$.

Since (a) holds, we get

$$X_{t-\tau} \perp\!\!\!\perp Y_t | \text{Pa}^{\text{descr}}_t(Y_t), \text{Pa}^{\text{union},-}_t(Y_t), \text{Pa}^{\text{union},-}_{t-\tau}(X_{t-\tau}).$$

By Lemma H.5, we can add the other variables from the superset of lagged parents (including the context variables), and obtain

$$X_{t-\tau} \perp\!\!\!\perp Y_t | (\text{Pa}^{\text{descr}}_t(Y_t), (\hat{\mathcal{B}}^{\text{union},-}_t(Y_t) \setminus \{X_{t-\tau}\}, \hat{\mathcal{B}}^{\text{union},-}_t(X_{t-\tau}))).$$

This implies that, for all $\mathbf{c}$,

$$X_{t-\tau} \perp\!\!\!\perp Y_t | (\mathrm{Pa}_t^{\mathrm{descr}}(Y_t), (\hat{\mathcal{B}}_t^{\mathrm{union},-}(Y_t) \setminus \{X_{t-\tau}\}, \hat{\mathcal{B}}_t^{\mathrm{union},-}(X_{t-\tau}))) \setminus Z^C, Z^C = \mathbf{c}.$$

which is tested at some point in our algorithm.

**Case 3:** We assume (a) does not apply.

Now we most prove that, if (a) doesn't apply, then we test the context-specific relationship given in (b).

If (a) doesn't hold, then either

- $Y_t$ is a union child of $C$ at some lag, or
- $Y_t$ is part of a union cycle.

In the first case, the context parent of $Y_t$ will be included in the conditioning set at some point, and thus the context-specific relationship given in (b) using supersets of the lagged parents (thereby using Lemma H.5) will be tested.

If $Y_t$ is part of a union cycle, then, by the minimal edge flips assumption, it is part of a minimal cycle. Since both nodes in a minimal cycle are children of the context variable at possibly different lags by weak context-sufficiency, this means that $Y_t$ is also a union child of $C$ at some lag, and we can argue like above. $\qquad\square$

## I. Data generation

We generate the synthetic structural causal processes for the experiments using the procedure implemented in *Tigramite* (https://github.com/jakobrunge/tigramite). For a given number of variables $N$, a random causal order is first sampled, and auto-lag links with coefficients are drawn as described in the main paper.

The total number of links $L$ across all possible lags up to $\tau_{\mathrm{max}}$ is determined by the sparsity level $s$, which specifies the expected fraction of possible variable pairs. The resulting $L$ links are divided into contemporaneous and lagged connections according to the fraction $f_c$. For each chosen pair $(i \to j)$, the lag $\tau$ is set to zero if the link is contemporaneous, or drawn uniformly from $\{1, \ldots, \tau_{\mathrm{max}}\}$, with $\tau_{\mathrm{max}} = 2$, if lagged. Link strengths are sampled as described in the main paper.

For all experiments in the main paper the sample size is $n = 1000$.

To evaluate the robustness of our algorithms to context misalignment, we include two experiments, where we misspecify the context values in two modes. In the first experiment, we shift the threshold for the context value by a $\Delta \in (0, 1)$ using the minimum $\min$ and the maximum $\max$ values of the continuous values of the context values. We thus compute a new threshold for the categorical context variable $t_n$ by shifting the old thresholds $t_o$ as $t_n = t_o + \Delta \cdot (\max - \min)$. We note that this approach leads to an imbalance in sample sizes between the different contexts. In the second experiment, we flip a fraction $\Delta \in (0, 1)$ of the categorical context values, effectively keeping the sample size the same as before the flip.

## J. Results

### J.1. Synthetic data

### J.2. Exogenous vs. endogenous context variable

Fig. 10 presents results for the setup described in Sec. 7.1 and App. I with either an exogenous context variable (Fig. 10, left) or an endogenous context variable (Fig. 10, right), and all other experiment parameters fixed. We observe similar behavior between the two setups, but there is a slightly higher FPR for all methods under the endogenous context variable setup. The small differences might come due to the fact that, in the time series settings, the overall number of links is much higher compared to the i.i.d. setting (leading to a high denominator), and thus the differences in the rates between the two settings is overall smaller.

### J.3. Increasing the number of changed links

Fig. 11 presents results for the setup described in Sec. 7.1 and App. I with $n_{\mathrm{change}} = 4$ changed links. The results show similar behavior to the setup with $n_{\mathrm{change}} = 2$. However, for the setup with no known context-system adjacencies (solid lines

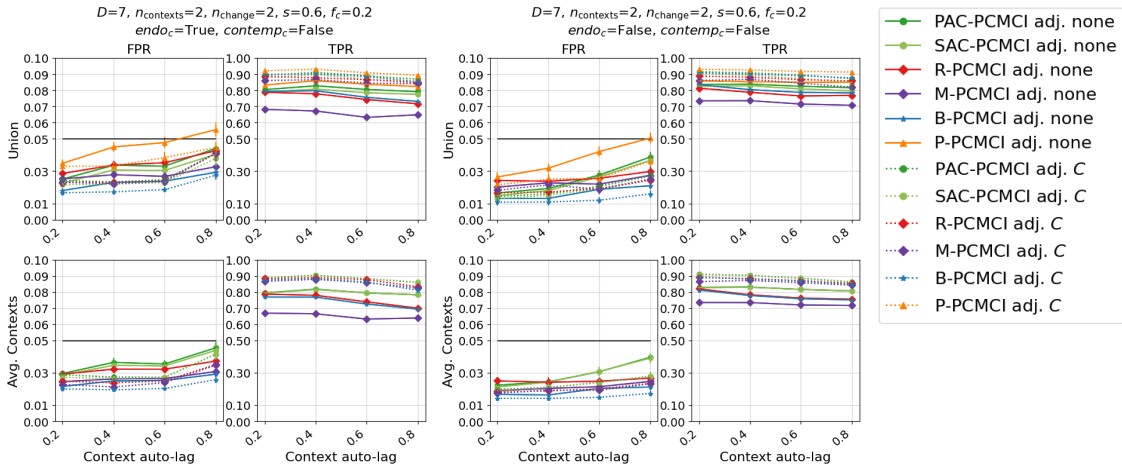

*Figure 10.* Results for PAC-PCMCI$^+$ (persistent), SAC-PCMCI$^+$ (sparse), B-PCMCI (baseline), M-PCMCI (individual datasets) and P-PCMCI (pooled datasets) with an exogenous context variable (left panel) or endogenous context variable (right panel). Solid lines indicate settings where no context-system links are set as link assumptions, while dotted lines indicate settings where links context-system links are known (see Sec. 7).

in Fig. 11), we observe that both the FPR and TPR are lower than in the setting with $n_{change} = 2$, indicating that, overall, less context-system links are found.

### J.4. Varying sample size

Fig. 12 depicts results when varying sample sizes $n \in \{200, 600, 1000, 2000\}$. We can observe that, as the sample size increases, so does the FPR for all methods. This is probably due to overall more links being found. However, the overall trend is consistent with the results from the other setups, and we can observe that PAC-PCMCI$^+$ and SAC-PCMCI$^+$ have the best performance in terms of the trade-off between FPR and TPR.

### J.5. Varying graph size

An evaluation on graphs with varying number of nodes, namely 8, 10 and 15, presented in Fig. 13, show a linear decrease in performance, i.e., an increase in FPR and decrease in TPR, as the number of nodes increases. This behavior is expected, and it is generally known that constraint-based methods struggle with large number of nodes due to repeated CITs. However, we observe that our methods perform best in terms of FPR/TPR trade-off, especially in the case of no known context-system links.

### J.6. Robustness to context misalignment

We evaluate the robustness with the two methods described in App. I and present them in Fig. 14. For the thresholding shift approach, we observe that our methods are robust to misspecifications of the context values, especially for $\Delta$ values of up 0.3. For $\Delta = 0.5$ we observe a higher degraation of the metrics, especially for the context-specific graphs. This is expected, as the sample size for one of the context decreases considerably with such a high $\Delta$, thus impacting performance. We observe similar behavior for the flip approach. However, performance does not drop as much as with the thresholding approach for higher $\Delta$, possibly due to the higher number of samples.

### J.7. Robustness to non-linear system-system links

We evaluated the robustness of our algorithm to non-linear system-system links, presented in Fig. 15. For this, we use the non-linear function $f(x) = x + 5 \cdot x^2 \cdot \exp(\frac{-x^2}{20})$. Due to computational time reasons, we use the same partial correlation CIT as for the linear system. As expected, we observe an incerase in FPR and a decrease in TPR for the non-linear case, due to the incorrect type of CIT used. We note that this is not a flaw of our algorithm, but a general flaw of constraint-based causal discovery: using the wrong test will always lead to worse results, independent of the algorithm used. However, we can observe that among all methods, our approaches have the most balanced results: while they have a higher FPR, they also

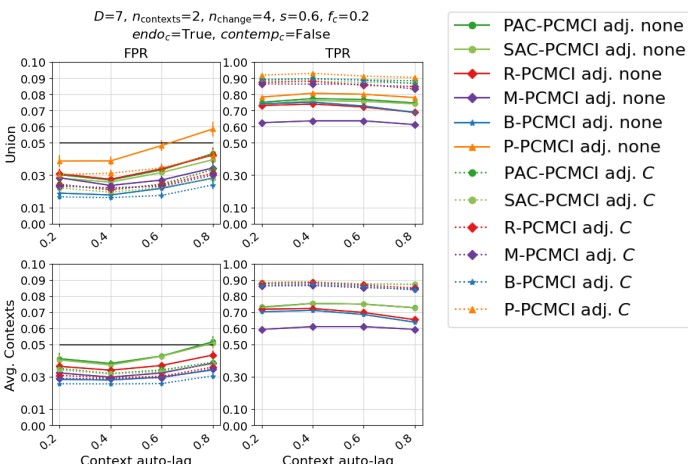

*Figure 11.* Results for PAC-PCMCI$^+$ (persistent), SAC-PCMCI$^+$ (sparse), B-PCMCI (baseline), M-PCMCI (individual datasets), R-PCMCI (Regime-PCMCI with known context), and P-PCMCI (pooled datasets) with an endogenous context variable, either lagged (left panel) or contemporary (right panel) and $n_{change} = 4$ changed links between the different contexts. Solid lines indicate settings where no context-system links are set as link assumptions, while dotted lines indicate settings where links context-system links are known (see Sec. 7).

report the highest TPR. We would recommend using a non-linear CIT such as a nearest-neighbors based approach. However, these CITs are considerably more computationally expensive, and thus we refrain from further experiments with different CITs here.

### J.8. Real-world data

Understanding the causal mechanisms behind climate extremes requires disentangling how the land-atmosphere feedbacks differ under varying moisture contexts. The problem of finding these land-atmosphere feedbacks using context-specific causal discovery has been addressed in Popescu et al. (2024), where the authors applied Regime-PCMCI (Saggioro et al., 2020) on a dataset containing soil moisture and temperature data derived from ERA5 reanalysis data over Western Europe (1993–2022). The dataset includes variables such as precipitation, root-zone soil moisture, latent and sensible heat fluxes, short-wave radiation, and upper-level circulation indices, aggregated to 3-day means. The authors define "moist", "normal" and "dry" contexts via quantiles of the continuous soil moisture variable. They also pre-process to remove seasonality and standardize anomalies. We employ this dataset to learn context-specific causal graphs and examine how soil–atmosphere coupling varies across moisture states using the PAC-PCMCI$^+$ and SAC-PCMCI$^+$ algorithms. We restrict to the following variables: temperature at the 2m level (T2m), short-wave radiation (SW), latent heat (LH), sensible heat (SH), precipitation (Prec), the circulation function (Stream), as well as the soil moisture, which is in our case the context variable (Context). We chose this example as, in their work, the authors present a ground-truth graph hypothesized from expert knowledge, against which we can compare the output of our algorithms.

The results for PAC-PCMCI$^+$ and SAC-PCMCI $^+$(Fig. 16) show that the algorithms detect the expected links between the stream function, temperature, precipitation and short-wave radiation, however, they find a negative instead of a positive relationship between the stream function and short-wave radiation. Furthermore, the algorithms find the expected links between short-wave radiation and latent and sensible heat. The results show a positive precipitation to latent heat link in the dry and normal contexts and a negative latent heat to precipitation link in the moist context. However, this link is spurious, and we believe it is due to the actual link between latent heat fluxes and soil moisture. Since soil moisture is driven by precipitation, the algorithm seems to find the link between latent heat and precipitation directly. There are a few further spurious links, for example, from temperature to latent heat. The soil–moisture–atmosphere interactions form a complex system, which can be difficult to fully capture, and thus lead to the algorithms finding confounding links. Furthermore, as discussed in the main paper, conditional independence testing for mixed-type data is a difficult problem, which can also contribute to some of the erroneous links, e.g., the missing link between soil moisture (Context) and latent heat. Nevertheless, the results demonstrate that our method is capable of uncovering meaningful dependencies in real-world data.

We also compare our results to the ones obtained using Regime-PCMCI in Popescu et al. (2024), as well as to the expert

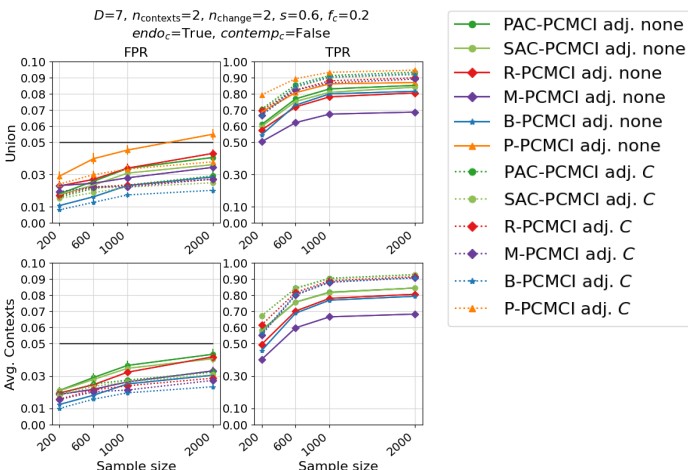

*Figure 12.* Results for PAC-PCMCI$^+$ (persistent), SAC-PCMCI$^+$ (sparse), B-PCMCI (baseline), M-PCMCI (individual datasets), R-PCMCI (Regime-PCMCI with known context), and P-PCMCI (pooled datasets) with endogenous context variables, either lagged (left panel) or contemporary (right panel) and varying sample size $n \in \{200, 600, 1000, 2000\}$. Solid lines indicate settings where no context-system links are set as link assumptions, while dotted lines indicate settings where links context-system links are known (see Sec. 7).

causal graph presented there. A detailed link-wise comparison is provided in table 2. PAC-PCMCI$^+$ and SAC-PCMCI$^+$ recover the key expert-identified pathways more accurately and produce substantially sparser graphs. In particular, they consistently identify the main feedback structures (e.g., Stream → SW, SW → SH/LH, Stream → T2m) with correct signs, while avoiding many unsupported connections. In contrast, Regime-PCMCI yields considerably denser graphs with numerous additional links that are not supported by the expert knowledge. This is likely due the presence of endogenous contexts and sensitivity to the regime partitioning; indeed, obtaining interpretable results in prior work required substantial expert intervention to constrain the model, as Regime-PCMCI does not need expert knowledge of the regime variable, but its optimization strategy of finding regime variables will potentially lead to splitting the data in ways that will introduce many spurious links. Our approaches rely on a different type of expert knowledge: context asignements instead of link constraints, thus helping to avoid confirmation bias. Importantly, PAC-PCMCI$^+$ and SAC-PCMCI$^+$ better disentangle context-specific mechanisms without relying on nearly as much expert knowledge as Regime-PCMCI. While they do not always recover all direct links (e.g., to soil moisture), they identify context-dependent proxy relationships (such as TP → LH) that align closely with the ground truth. In contrast, Regime-PCMCI tends to mix such effects across regimes or obscure them through additional spurious connections. Overall, our methods achieve a more favorable balance between sensitivity and specificity, yielding interpretable graphs that capture both invariant and context-dependent relationships. We will incorporate the proposed table and an expanded discussion in the revision to make this comparison explicit. Table 2 summarizes the results.

### J.9. Computation times

We present the evaluation of computation times in Table 3 for different graph sizes. We observe that our methods have the highest computation times, due to more CITs run due to the found context-system link probably overall more found links, due to larger sample sizes when testing adaptively compared to masking. On the other hand, the masking approaches have the lowest computation times, as they do not find context-system links, and thus generally run less tests. PCMCI$^+$ is also faster than our methods, possibly beacause it only runs once on the pooled data only. However, PCMCI$^+$ does not give the same level of information, returning only the union graph.

## K. Semi-synthetic case study: endogenous regimes in a macro–finance model

The real-world climate application in App. J.8 demonstrates that PAC- and SAC-PCMCI$^+$ can recover meaningful context-dependent structure when the ground-truth graph is only partially known. To complement that study with a setting in which the ground-truth graph is fully specified, but the data-generating process is still motivated by a real application domain, we

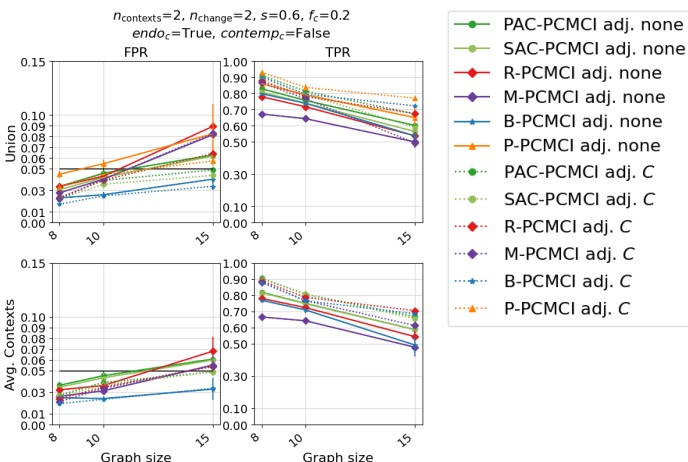

*Figure 13.* Results for PAC-PCMCI$^+$ (persistent), SAC-PCMCI$^+$ (sparse), B-PCMCI (baseline), M-PCMCI (individual datasets), R-PCMCI (Regime-PCMCI with known context), and P-PCMCI (pooled datasets) for varying graph sizes with 8, 10 or 15 nodes. Solid lines indicate settings where no context-system links are set as link assumptions, while dotted lines indicate settings where links context-system links are known (see Sec. 7).

provide a semi-synthetic case study from macroeconomics and finance.[3] At the same time this case-study serves as a guiding example for practitioners who look to apply our proposed algorithms on real data, especially with regard to evaluating if the properties of persistence and sparsity hold on this data. The simulator combines two endogenous Markov-switching regime variables, a *monetary-policy regime* and a *financial-stress regime*, with five continuous macro–finance variables whose structural coefficients depend on the regimes. This appendix is organised as a worked tutorial: we describe the data-generating process (Sec. K.2), specify the ground-truth context window graphs (Sec. K.3), evaluate the persistence (Sec. K.4) and sparsity (Sec. K.5) assumptions on the simulated data, follow the decision rule of App. C to select between PAC- and SAC-PCMCI$^+$ (Sec. K.6), and report the recovered context-specific graphs (Sec. K.7).

### K.1. Motivation

In macro–finance, structural relationships routinely change with the prevailing regime: a monetary-policy response that is active in a *tight* regime may become dormant near the zero lower bound, and a credit-channel link from real activity to spreads may activate only in a *financial-stress* regime. Our simulator is inspired by the Markov-switching VAR analysis of monetary policy and financial stress in Hubrich & Tetlow (2015), with time-varying transition probabilities as in the endogenous-switching extension of Hubrich et al. (2026). The defining feature of this setting is that the regime indicators are *endogenous*: tight policy is triggered by lagged inflation and uncertainty, financial stress emerges from past spreads and uncertainty, and both regimes feed back into the very dynamics they govern. This is precisely the setting that PAC- and SAC-PCMCI$^+$ are designed for, and the simulator below constructs a controlled instance of it.

### K.2. Data-generating process

We simulate $N = 7$ variables: five continuous macro–finance series $\mathrm{gdp\_gap}_t$, $\mathrm{inflation}_t$, $\mathrm{policy\_rate}_t$, $\mathrm{credit\_spread}_t$, $\mathrm{uncertainty}_t$ and two binary context indicators $z_t^{\mathrm{policy}} \in \{0,1\}$ and $z_t^{\mathrm{finance}} \in \{0,1\}$. The two regime variables play the role of the discrete context series $C$ of Def. 3.1 (multiple context variables; here $K = 2$ and the context window is of length $\tau_{\max} + 1 = 2$ in our experiments). Throughout, $\eta_t^{(\cdot)}$ are i.i.d. Gaussian innovations and $\sigma(x) = (1 + e^{-x})^{-1}$.

---

[3]Beyond the broader appeal of an economic example, this also responds to the discussion in the main text on endogenous volatility regimes in financial markets, which is one of the principal non-climate motivations for our framework.

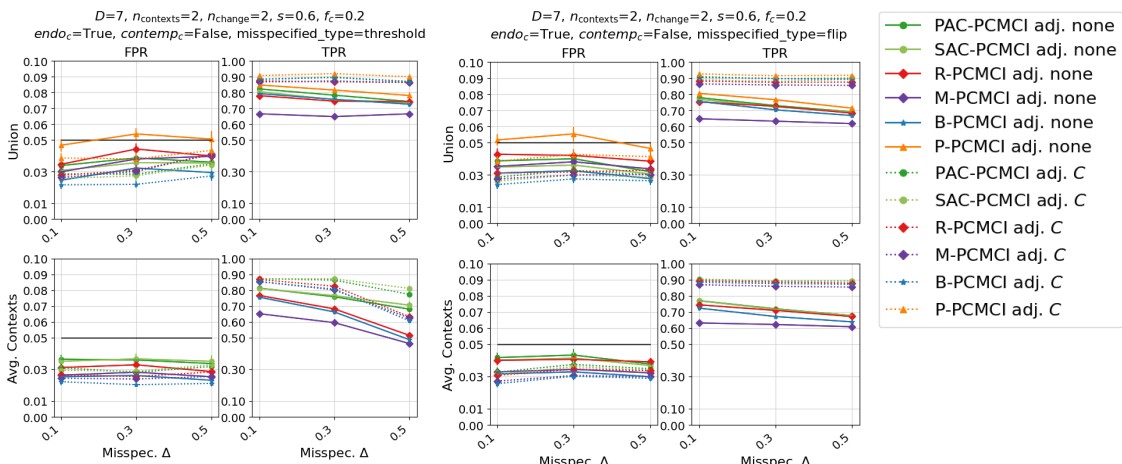

*Figure 14.* Results for PAC-PCMCI$^+$ (persistent), SAC-PCMCI$^+$ (sparse), B-PCMCI (baseline), M-PCMCI (individual datasets), R-PCMCI (Regime-PCMCI with known context), and P-PCMCI (pooled datasets) for misspecified context values with the threshold approach (left panel) and flip approach (right panel), see Sec. I. Solid lines indicate settings where no context-system links are set as link assumptions, while dotted lines indicate settings where links context-system links are known (see Sec. 7).

**Endogenous Markov-switching transitions.** Each regime variable evolves as a first-order Markov chain whose transition probability depends on lagged observed variables, making the indicator *endogenous* in the sense of Def. 3.1:

$$\Pr(z_t^{\text{policy}} = 1 \mid \mathcal{F}_{t-1}) = \begin{cases} \sigma\big(1.3 + 1.0\,\text{inflation}_{t-1} + 0.6\,\text{uncertainty}_{t-1}\big) & \text{if } z_{t-1}^{\text{policy}} = 1, \\ \sigma\big(-1.8 + 1.4\,\text{inflation}_{t-1} + 0.5\,\text{uncertainty}_{t-1}\big) & \text{if } z_{t-1}^{\text{policy}} = 0, \end{cases} \tag{8}$$

$$\Pr(z_t^{\text{finance}} = 1 \mid \mathcal{F}_{t-1}) = \begin{cases} \sigma\big(1.1 + 1.5\,\text{credit\_spread}_{t-1} + 0.7\,\text{uncertainty}_{t-1}\big) & \text{if } z_{t-1}^{\text{finance}} = 1, \\ \sigma\big(-4.0 + 1.7\,\text{credit\_spread}_{t-1} + 0.8\,\text{uncertainty}_{t-1}\big) & \text{if } z_{t-1}^{\text{finance}} = 0. \end{cases} \tag{9}$$

The asymmetry of the intercepts induces *regime-dependent persistence*: each regime is much more likely to remain in state 1 than to switch into it. This mirrors the empirical observation that financial-stress regimes (e.g., the 2008–2009 and 2020 episodes) tend to be stable once entered.

**Regime-modulated structural equations.** The continuous variables follow AR(1)-type updates whose coefficients depend on the contemporaneous regime indicators. The key feature is that several edges in the structural graph are *switched off* when the corresponding regime equals 0, producing genuine *context-specific independencies*:

$$\text{gdp\_gap}_t = 0.5\,\text{gdp\_gap}_{t-1} - 0.30\,z_t^{\text{policy}}\,\text{policy\_rate}_{t-1} - 0.10\,z_t^{\text{finance}}\,\text{credit\_spread}_{t-1} + \eta_t^{(g)}, \tag{10}$$

$$\text{inflation}_t = (0.3 + 0.4\,z_t^{\text{policy}})\,\text{inflation}_{t-1} + (0.30 - 0.3\,z_t^{\text{finance}})\,\text{gdp\_gap}_{t-1} + 0.20\,\text{uncertainty}_{t-1} + \eta_t^{(\pi)}, \tag{11}$$

$$\text{policy\_rate}_t = 0.65\,\text{policy\_rate}_{t-1} + 0.45\,\text{inflation}_{t-1} + (0.10 - 0.3\,z_t^{\text{finance}})\,\text{gdp\_gap}_{t-1} + \eta_t^{(r)}, \tag{12}$$

$$\text{credit\_spread}_t = 0.62\,\text{credit\_spread}_{t-1} - 0.22\,z_t^{\text{finance}}\,\text{gdp\_gap}_{t-1} + 0.33\,z_t^{\text{finance}}\,\text{uncertainty}_{t-1} + \eta_t^{(c)}, \tag{13}$$

$$\text{uncertainty}_t = 0.85\,\text{uncertainty}_{t-1} + 0.14\,z_{t-1}^{\text{policy}} + 0.14\,z_{t-1}^{\text{finance}} + \eta_t^{(u)}. \tag{14}$$

The interactions in Eqs. (10), (13) are products of the form $z_t \cdot x_{t-1}$, which deactivate the corresponding lagged edge whenever $z_t = 0$. This is non-additive in the sense of Def. G.2(ii) and matches the design choice of the main-text synthetic experiments (Sec. 7): the context modifies the graph structure itself rather than entering as an additive term.

We use $T = 5000$ post-burn-in samples (burn-in $= 3000$) and seed 101. All diagnostics and methods in the remainder of this section use this single realisation.

```
1  data, columns = simulate_endogenous_markov_macro(T=5000, burnin=3000, seed=101)
2  df_pd = pd.DataFrame(data, columns=columns)
```

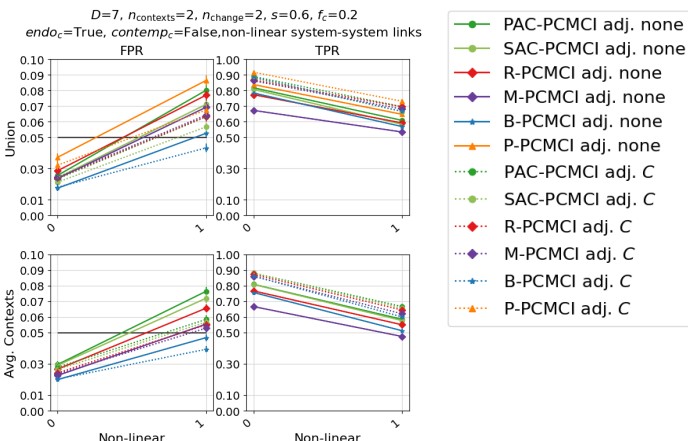

*Figure 15.* Results for PAC-PCMCI$^+$ (persistent), SAC-PCMCI$^+$ (sparse), B-PCMCI (baseline), M-PCMCI (individual datasets), R-PCMCI (Regime-PCMCI with known context), and P-PCMCI (pooled datasets) with either linear or non-linear system-system links. Solid lines indicate settings where no context-system links are set as link assumptions, while dotted lines indicate settings where links context-system links are known (see Sec. 7).

```
3   # columns = ['gdp_gap','inflation','policy_rate','credit_spread',
4   #             'uncertainty','z_policy','z_finance']
```

*Listing 1.* Calling the simulator. The full source is in the companion notebook.

### K.3. Ground-truth context-window graphs

From Eqs. (10)–(14) the structural-parent (mechanism) graph is fully specified and we can enumerate the four context window graphs $G_{C \leq = c}^{\text{descr}}$ for $c \in \{0,1\}^2$ corresponding to $(z^{\text{policy}}, z^{\text{finance}})$. Three categories of edges arise:

- **Always-present (context-invariant) edges:** autoregressive lags for all variables; $\text{inflation}_{t-1} \to \text{policy\_rate}_t$; $\text{gdp\_gap}_{t-1} \to \text{inflation}_t$; $\text{uncertainty}_{t-1} \to \text{inflation}_t$; the context-to-system edges $\text{inflation}_{t-1} \to z_t^{\text{policy}}$, $\text{uncertainty}_{t-1} \to z_t^{\text{policy}}$, $\text{credit\_spread}_{t-1} \to z_t^{\text{finance}}$, $\text{uncertainty}_{t-1} \to z_t^{\text{finance}}$; and the lagged regime effects $z_{t-1}^{\text{policy}} \to \text{uncertainty}_t$, $z_{t-1}^{\text{finance}} \to \text{uncertainty}_t$.

- **Policy-regime-specific edges:** the edge $\text{policy\_rate}_{t-1} \to \text{gdp\_gap}_t$ is active only whenever $z^{\text{policy}} = 1$. There is also a contemporaneous influence $z_t^{\text{policy}} \to \text{inflation}_t$ via the interaction term $z_t^{\text{policy}} \cdot \text{inflation}_{t-1}$.

- **Finance-regime-specific edges:** $\text{credit\_spread}_{t-1} \to \text{gdp\_gap}_t$, $\text{gdp\_gap}_{t-1} \to \text{credit\_spread}_t$, $\text{uncertainty}_{t-1} \to \text{credit\_spread}_t$, and the contemporaneous links $z_t^{\text{finance}} \to \text{inflation}_t$, $z_t^{\text{finance}} \to \text{policy\_rate}_t$, $z_t^{\text{finance}} \to \text{credit\_spread}_t$ are all active only when $z^{\text{finance}} = 1$.

Figure 17 visualises the union graph with regime-specific edges colour-coded.

### K.4. Persistence diagnostic

We assess Property 1 by extracting all maximal constant runs of each regime variable and reporting summary statistics per value. Recall that $(w, \delta)$-persistence requires $\Pr(C_t = C_{t+1} = \cdots = C_{t+w-1} = c) > \delta$. In our experiments $\tau_{\max} = 1$ and hence $w = \tau_{\max} + 1 = 2$.

```python
1  def count_constant_windows_by_value(series):
2      series = np.asarray(series)
3      change_points = np.flatnonzero(series[1:] != series[:-1]) + 1
4      boundaries = np.r_[0, change_points, len(series)]
5      run_values  = series[boundaries[:-1]]
6      run_lengths = np.diff(boundaries)
7      return {v: run_lengths[run_values == v] for v in (0, 1)}
```

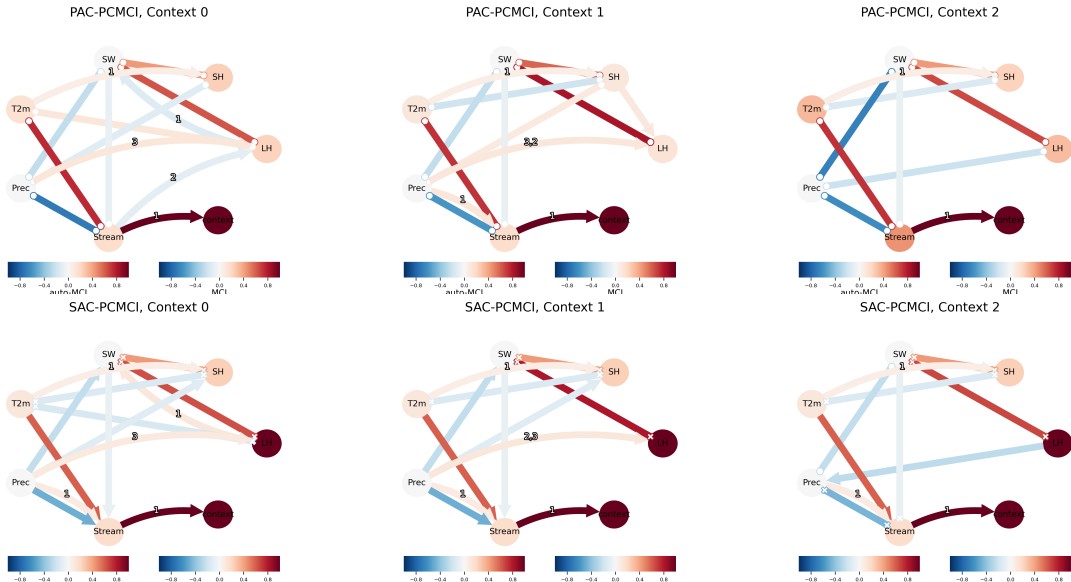

*Figure 16.* Context-specific graphs obtained with the PAC-PCMCI$^+$ algorithm (first row) and the SAC-PCMCI$^+$ algorithm (second row) on the land-atmosphere feedbacks dataset from Popescu et al. (2024) for the dry (Context 0, left), normal (Context 1, middle), and moist (Context 2, right) contexts. Link colors indicate link strengths (MCI), while node colors indicate autocorrelation strength (auto-MCI).

*Listing 2.* Run-length diagnostic (full code in the notebook).

The results on the single realisation used throughout this section are summarised in Table 4.

**Interpretation.** Persistence is comfortably satisfied for the persistent states ($z^{\mathrm{policy}} = 1$, $z^{\mathrm{finance}} = 1$), with $\hat{\delta}_2 \approx 0.85$ for both. The opposite states have markedly shorter runs, particularly $z^{\mathrm{policy}} = 0$, where the median run length of 2 time steps is exactly the window length we use for testing. PAC-PCMCI$^+$ restricts each context-specific independence test to those time points at which the entire $\tau_{\max} + 1$-long window is constant at the queried regime (Sec. 5, PAC-PCMCI$^+$ paragraph). The effective sample size for the $(0, \cdot)$ slice of $z^{\mathrm{policy}}$ is therefore an order of magnitude smaller than for the $(1, \cdot)$ slice. We expect PAC-PCMCI$^+$ to remain sound on this dataset but to lose statistical power on links that change in the short-run regime. SAC-PCMCI$^+$ does not impose persistence and only restricts to the specific context lags that are parents of the target, which preserves more samples per test when sparsity holds.

### K.5. Sparsity diagnostic

Sparsity requires that each system variable has at most one context variable (at a possibly unknown lag) as a parent. In our simulator this property holds exactly for $\mathrm{policy\_rate}$, $\mathrm{credit\_spread}$ and $\mathrm{gdp\_gap}$ (each switched by only one of the two regimes), while $\mathrm{inflation}$ also has only $z_t^{\mathrm{policy}}$ as a contemporaneous parent via the interaction term. The variable $\mathrm{uncertainty}$, however, has *both* $z_{t-1}^{\mathrm{policy}}$ and $z_{t-1}^{\mathrm{finance}}$ as lagged parents (Eq. (14)). The sparsity assumption is therefore *violated* for $\mathrm{uncertainty}$. To verify this from data alone (i.e., without using the ground truth) we run a pooled PCMCI$^+$ to obtain a union graph, then count for each non-regime variable how many regime variables appear as parents at any lag. This is essentially the implicit sparsity test built into the first phase of SAC-PCMCI$^+$ (Sec. 3.2.1, Testability).

```
1  data_type = np.zeros_like(data, dtype=int)
2  for j in regime_idx:
3      data_type[:, j] = 1
4  df_tigramite = pp.DataFrame(data=data, data_type=data_type,
5                              datatime={0: np.arange(len(data))})
6
7  pcmci_pooled = MixedTestPCMCI(
8      dataframe=df_tigramite,
9      cond_ind_test=RobustParCorr(),
```

*Table 2.* Link-wise comparison against ground truth. PAC-PCMCI$^+$ and SAC-PCMCI$^+$ recover correct links and context-dependence more reliably than Regime-PCMCI. We focus on the skeleton.

| Ground Truth | PAC-PCMCI$^+$ | SAC-PCMCI$^+$ | Regime-PCMCI |
|---|---|---|---|
| Stream − SW (all) | Correct, stable | Correct, stable | inconsistent lags |
| Stream − T2m (all) | Correct, stable | Correct, stable | Correct, stable |
| Stream − TP (all) | Correct, stable | Correct, stable | Correct, stable |
| SW − LH (all) | Correct (lag 1 in regime 0) | Correct (lag 1 in regime 0) | Correct, stable |
| SW − SH (all) | Correct, stable | Correct, stable | Correct, stable |
| LH − SH (dry) | Context-dependent (present in middle) | Missing | Context-dependent (present in middle) |
| LH − SM/Context (reversed btw. regimes) | Missing | Missing | Present in all regimes, no reversal |
| TP − SM (all) | Connected via Stream | Connected via Stream | Correct, stable |
| Spurious edges | | | |
| TP − LH (none) | Present in all regimes (lag changes in moist) TP proxy for SM? | Present in all regimes (lag changes and reversed in moist) TP proxy for SM? | No link |
| TP − SH (none) | Inconsistent | Inconsistent | No link |
| TP − SW (none) | Present in all regimes | Present in all regimes | No link |
| T2m − SH (none) | Present in all regimes | Present in all regimes | Present in all regimes |
| Other (Stream − LH, SM − SH, Stream − SH, T2m − SW, T2m − SM) | No link | No link | Present in all or some regimes |

```python
10      disc_cond_ind_test=RegressionCI(),
11      verbosity=0,
12  )
13  pooled_res   = pcmci_pooled.run_pcmciplus(tau_max=1, pc_alpha=0.05)
14  pooled_graph = pooled_res['graph']
15
16  def count_regime_parents(graph, var_names, regime_vars):
17      regime_idx_local = [var_names.index(v) for v in regime_vars]
18      counts = {}
19      for child_idx, child_name in enumerate(var_names):
20          if child_name in regime_vars: continue
21          counts[child_name] = sum(
22              np.any(graph[r, child_idx, :] != '') for r in regime_idx_local
23          )
24      return counts
```

*Listing 3.* Pooled-PCMCI$^+$ sparsity check.

The result on our realisation is shown in Table 5.

**Interpretation.** The data-driven sparsity check correctly identifies a violation of sparsity: uncertainty has two regime parents. There are further violations of sparsity in the ground truth graph, the diagnostic via pooled PCMCI$^+$ is prone to errors in the used causal discovery algorithm.

*Table 3.* Computation time in seconds by method and graph size.

| Method | $D = 7$ | $D = 9$ | $D = 14$ |
|---|---|---|---|
| PAC-PCMCI | 10.5 | 21.7 | 104.4 |
| SAC-PCMCI | 8.8 | 17.0 | 76.4 |
| R-PCMCI | 3.0 | 7.9 | 56.6 |
| M-PCMCI | 2.2 | 5.9 | 44.9 |
| B-PCMCI | 3.0 | 7.9 | 56.6 |
| PCMCI$^+$ | 2.7 | 6.7 | 53.1 |

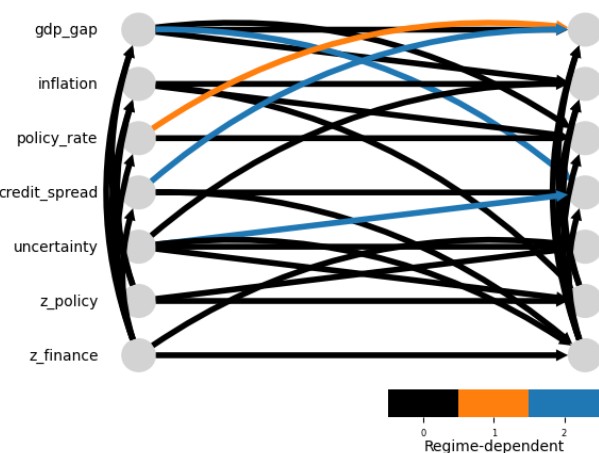

Ground-truth causal graph from simulation equations

*Figure 17.* Ground-truth time-series graph implied by Eqs. (10)–(14). Black edges are present in all four context window graphs; orange edges are active only under $z^{\text{policy}} = 1$; blue edges are active only under $z^{\text{finance}} = 1$.

## K.6. Choosing between PAC- and SAC-PCMCI$^+$

Combining the two diagnostics with the schematic in App. C (Fig. 4):

- Persistence holds strongly for the high-stress states and is weaker but still adequate for the low-stress states; SAC-PCMCI$^+$ does not require persistence.

- Our diagnostic indicates that sparsity is violated for at least one variable, uncertainty; SAC-PCMCI$^+$'s soundness guarantee (Thm. 6.1) requires sparsity globally.

This places the example squarely in the "persistent, not perfectly sparse" quadrant of Fig. 4. The guidance there is to prefer PAC-PCMCI$^+$ when sparsity is violated and persistence holds. We therefore run *both* methods to illustrate the difference, with the expectation that:

- PAC-PCMCI$^+$ is sound for all five system variables and any spurious or missing edges should be attributable to finite-sample effects on short $z^{\text{policy}} = 0$ runs, or conditional independence tests that are not suitable for multiplicative relationships;

- SAC-PCMCI$^+$ may produce additional spurious or missing edges due to the sparsity violation but should otherwise benefit from larger effective per-test sample sizes.

**When SAC-PCMCI$^+$ should still work despite a Property 2 violation.** The Property 2 violation we observe for uncertainty is of a particularly benign form. Inspecting Eqs. (10)–(13), each regime-dependent skeleton change in the simulator is governed by *exactly one* context variable. There is no product of the form $z_t^{\text{policy}} \cdot z_t^{\text{finance}} \cdot x_{t-1}$ anywhere in the

*Table 4.* Run-length statistics for the two regime indicators on the simulated sample ($T = 5000$). $\hat{\delta}_2$ is the empirical fraction of length-2 windows that are constant at value $c$, i.e., a Monte-Carlo estimate of $\Pr(C_t = C_{t+1} = c)$.

| Regime | Value $c$ | # runs | mean length | median length | $\hat{\delta}_2$ |
|---|---|---|---|---|---|
| $z^{\text{policy}}$ | 0 | 180 | 3.91 | 2.0 | 0.10 |
| $z^{\text{policy}}$ | 1 | 181 | 23.73 | 12.0 | 0.86 |
| $z^{\text{finance}}$ | 0 | 34 | 22.06 | 18.5 | 0.14 |
| $z^{\text{finance}}$ | 1 | 34 | 125.00 | 64.0 | 0.84 |

*Table 5.* Number of regime-variable parents per non-regime variable in the pooled-PCMCI$^+$ union graph at $\tau_{\max} = 1$, $\alpha_{\text{PC}} = 0.05$. sparsity property requires this count to be at most 1.

| Variable | # regime parents | sparsity property? |
|---|---|---|
| gdp_gap | 1 | ✓ |
| inflation | 1 | ✓ |
| policy_rate | 1 | ✓ |
| credit_spread | 1 | ✓ |
| uncertainty | 2 | ✗ |

structural equations, so flipping the value of one context variable cannot reactivate an edge that the other context variable has switched off. In other words, while uncertainty has two context parents at the level of its structural equation, the two context indicators *act on disjoint sets of edges* and do not interact at the level of the graph, for example the variable gdp$_g$ap has both $z_t^{\text{policy}}$ and $z_t^{\text{finance}}$ as context parents, but they turn off links to different system parents (policy$_r$ate and credit$_s$pread). This is the kind of assumption a practitioner could plausibly defend from domain knowledge: monetary-policy regimes and financial-stress regimes modulate different transmission channels (the policy-rate channel and the credit channel, respectively), and a graph edge that is broken by tight monetary policy is not expected to be restored by a financial-stress switch. We therefore conjecture that SAC-PCMCI$^+$ will recover the four context window graphs correctly even though we are in a case that is *not* covered by Thm. 6.1.

### K.7. Application of PAC- and SAC-PCMCI$^+$

We treat both binary regimes as context indicators jointly, yielding $K_1 \cdot K_2 = 4$ context contexts $(z^{\text{policy}}, z^{\text{finance}}) \in \{0, 1\}^2$, and run both methods on the simulated time series with $\tau_{\max} = 1$ and $\alpha_{\text{PC}} = 0.05$. We use `RobustParCorr` for purely-continuous tests and `RegressionCI` for tests involving the discrete regime variables, following the mixed-type strategy of the main paper.

```
context_vars   = regime_idx                              # [5, 6]
context_values = [(0, 1), (0, 1)]                        # binary, binary
unique_regimes = list(itertools.product(*context_values))

# PAC-PCMCI+
pac_graphs = {}
for regime in unique_regimes:
    cit = PersistentEndoCIT(mixed_cit=RegressionCI(),
                            cont_cit=RobustParCorr(),
                            context_vars=context_vars,
                            context_values=regime)
    pcmci = PersistentEndoPCMCI(dataframe=df_tigramite,
                                cond_ind_test=cit, verbosity=0)
    pac_graphs[regime] = pcmci.run_pcmciplus_fullpcmci(
        tau_max=1, pc_alpha=0.05)['graph']

# SAC-PCMCI+
sac_graphs = {}
for regime in unique_regimes:
    cit = SparseEndoCIT(mixed_cit=RegressionCI(),
                        cont_cit=RobustParCorr(),
```

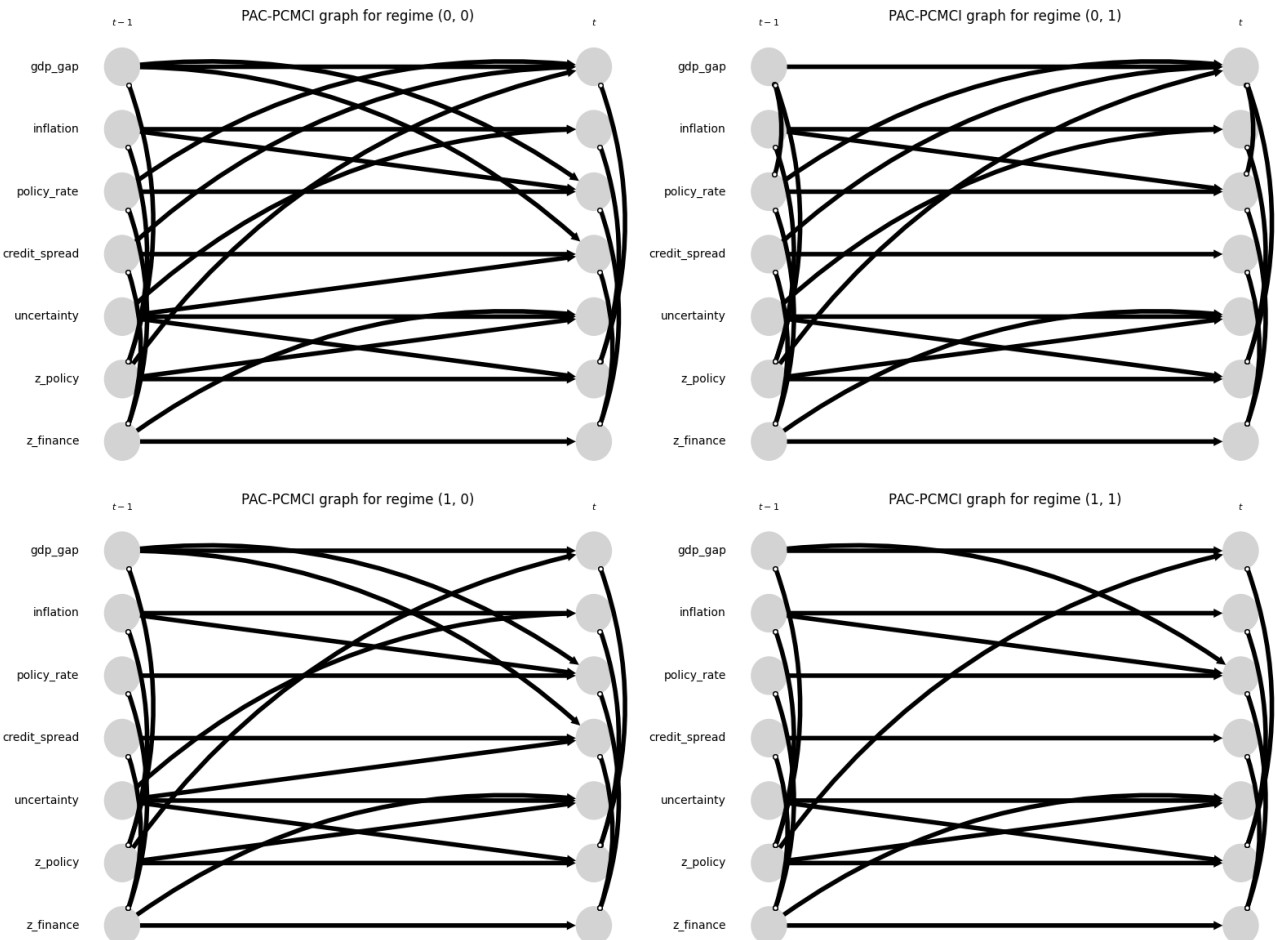

*Figure 18.* PAC-PCMCI$^+$ context window graphs $G_{C=c}^{\mathrm{descr}}$ for $c \in \{0,1\}^2 = (z^{\mathrm{policy}}, z^{\mathrm{finance}})$. Each of the context values is to be understood as persisting over the whole time window. Edge widths encode MCI strength.

```
22                        context_vars=context_vars,
23                        context_values=regime)
24    pcmci = SparseEndoPCMCI(dataframe=df_tigramite,
25                        cond_ind_test=cit, verbosity=0)
26    sac_graphs[regime] = pcmci.run_pcmciplus(
27        tau_max=1, pc_alpha=0.05)['graph']
```

*Listing 4.* Running PAC- and SAC-PCMCI$^+$ on the simulated data.

**Recovered context-specific graphs.** Figures 18 and 19 display the four context window graphs returned by PAC- and SAC-PCMCI$^+$, respectively. Each panel corresponds to one of the four regime combinations.

**Findings.** The qualitative pattern of results matches the prediction of Sec. K.6:

1. Both PAC- and SAC-PCMCI$^+$ recover the context-invariant skeleton reliably across all four contexts, and both correctly identify the regime-flip behaviour of the policy- and finance-specific edges into $\mathrm{gdp\_gap}_t$. This is a substantive improvement over pooled PCMCI$^+$, which captures only the union graph and therefore cannot distinguish a context-dependent edge from a context-invariant one.

2. The longest-run-length finance regime ($z^{\mathrm{finance}} = 1$) produces the cleanest context-specific graphs under PAC-PCMCI$^+$, with both finance-active edges ($\mathrm{gdp\_gap}_{t-1} \to \mathrm{credit\_spread}_t$, $\mathrm{uncertainty}_{t-1} \to \mathrm{credit\_spread}_t$) recovered with

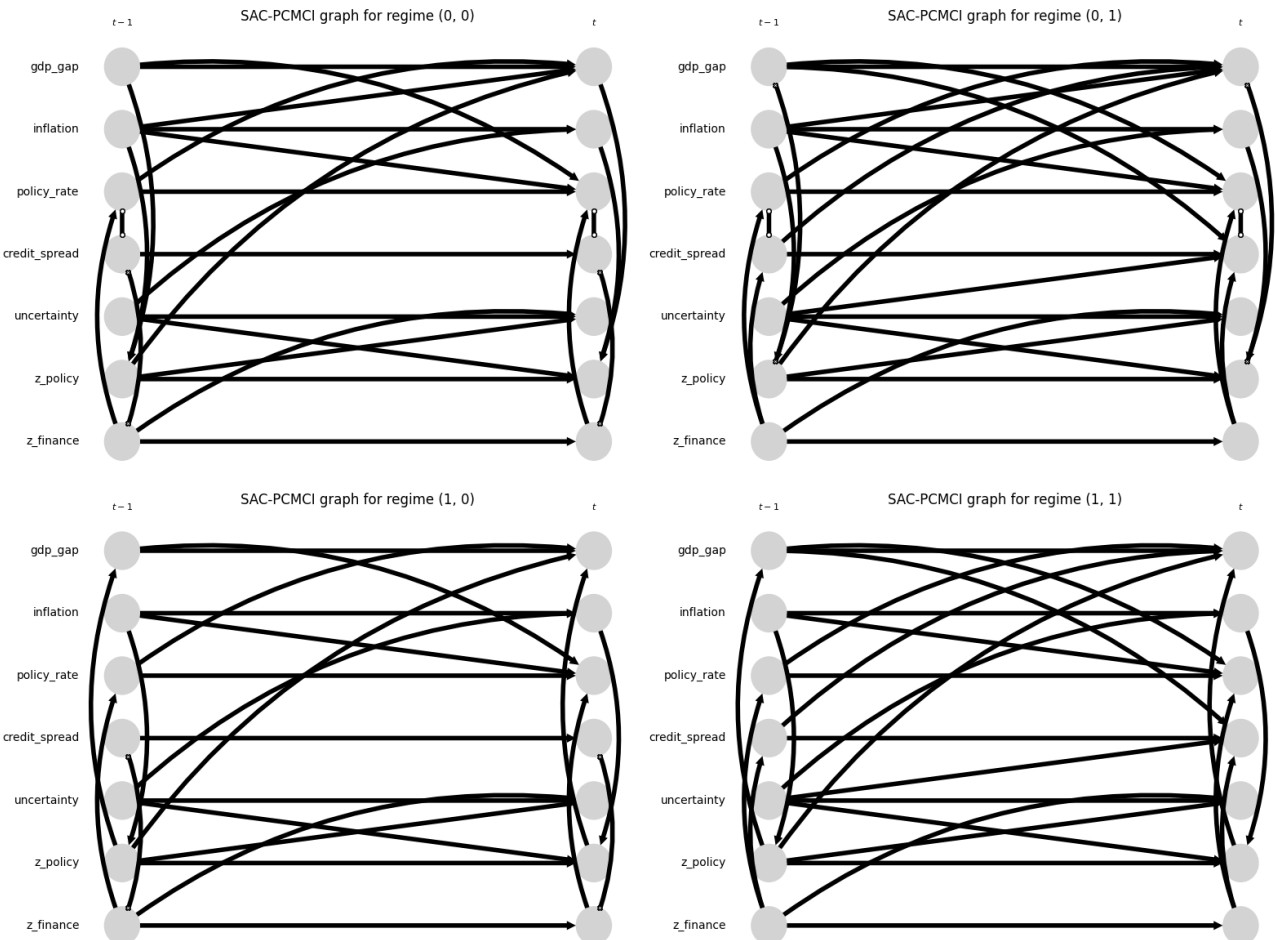

*Figure 19.* SAC-PCMCI$^+$ context window graphs $G_{C^{\leq}=c}^{\mathrm{descr}}$ for the same four contexts as in Fig. 18. We show results for the same contemporaneous context values, however SAC-PCMCI$^+$ uses data where these values do not persist over the time window (the values of $z_{t-1}^{\mathrm{policy}}$, $z_{t-1}^{\mathrm{finance}}$ might be different).

large MCI strengths. Conversely, the short-run-length policy regime ($z^{\mathrm{policy}} = 0$) is the most error-prone slice for PAC-PCMCI$^+$, as anticipated from the persistence diagnostic.

3. The context-system links are not found reliably, this is likely due to their multiplicative nature which RegressionCI is not designed to detect.

## K.8. Discussion

This case study demonstrates that the persistence and sparsity properties can be assessed directly from data using the simple run-length and pooled-PCMCI$^+$ diagnostics described above, before committing to a method. Furthermore, the schematic decision rule of App. C makes a useful suggestion here: PAC-PCMCI$^+$, which is robust to sparsity violations as long as persistence holds, is the more reliable choice on the macro–finance simulator and correctly recovers all regime-dependent edges of interest. Third, even when sparsity is locally violated, SAC-PCMCI$^+$ still recovers most of the true structure correctly, supporting the conjecture in App. C ("Future work") that a relaxed version of Property 2 would suffice to preserve soundness.

**Towards real macroeconomic data.** The simulator is deliberately a toy: it does not capture nominal rigidities, forward-looking expectations, or the full credit-channel mechanics of a quantitative DSGE model. The construction here, however, transfers directly to two natural next steps: (i) replacing the linear AR(1) updates with coefficients estimated from a

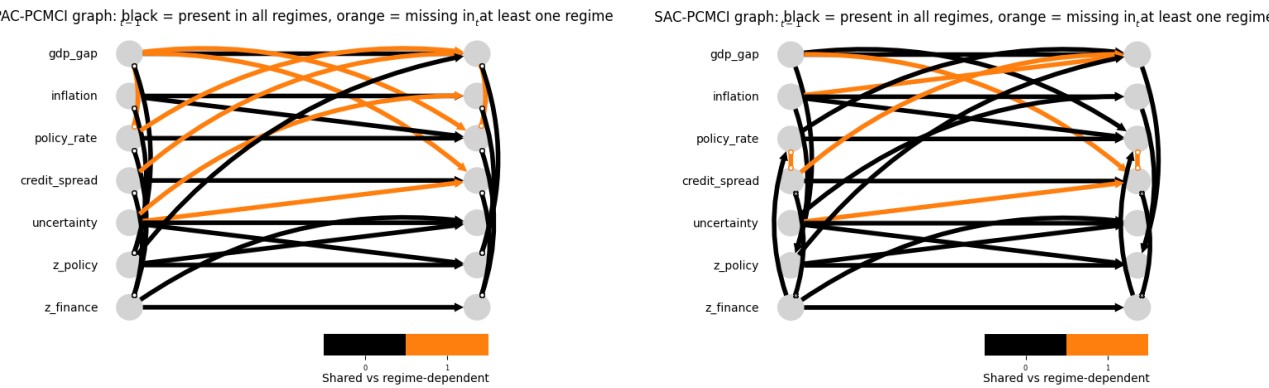

*Figure 20.* Union graphs colour-coded by whether each edge is present in all four context graphs (black) or only in a subset (orange). *Left:* PAC-PCMCI$^+$. *Right:* SAC-PCMCI$^+$. The three known regime-dependent skeleton edges (policy_rate$_{t-1}$ $\rightarrow$ gdp_gap$_t$, credit_spread$_{t-1}$ $\rightarrow$ gdp_gap$_t$, gdp_gap$_{t-1}$ $\rightarrow$ credit_spread$_t$, and uncertainty$_{t-1}$ $\rightarrow$ credit_spread$_t$) should appear in orange in the ideal output.

threshold-VAR fitted to FRED-MD data (financial-stress threshold on the excess bond premium or NFCI), and (ii) replacing the logistic regime transitions with a fitted occasionally binding-constraint mechanism such as a zero-lower-bound indicator. Both extensions preserve the endogenous-context structure, while improving the realism of the data-generating process. We leave a fully data-driven semi-synthetic macroeconomic benchmark to future work.

