# OpenReview forum: "Causal discovery for time series with endogenous context variables"
_ICML.cc/2026/Conference — ICML 2026 regular_

### Official Review · Reviewer_AsxX · 2026-02-26

**Soundness:** 3
**Presentation:** 3
**Significance:** 3
**Originality:** 3
**Overall Recommendation:** 5
**Confidence:** 3

**Summary:**

This paper addresses causal discovery in non-stationary time series where context variables (discrete indicators that modulate causal mechanisms) are endogenous — meaning they are influenced by the system variables they also modulate. This is a meaningful departure from prior work that assumes exogenous contexts. The authors formalize the modeling challenges (context explosion from exponentially many context windows, selection bias, lagged context effects) and identify two complementary structural properties — persistence (contexts change slowly) and sparsity (each variable has at most one context parent) — that render the problem tractable. They propose PAC-PCMCI+ (exploiting persistence) and SAC-PCMCI+ (exploiting sparsity), both extending the PCMCI+ framework with an adaptive conditional independence testing strategy that selects pooled or context-specific data depending on whether context variables appear in the conditioning set. Soundness proofs are provided under suitable assumptions. Evaluation on synthetic benchmarks and a real-world climate dataset (land-atmosphere feedbacks under soil moisture regimes) demonstrates effectiveness relative to several PCMCI+ variants.

**Compliance With Llm Reviewing Policy:**

Affirmed.

**Final Justification:**

I thank the authors for providing concrete preliminary results during the rebuttal on all four open questions.

W1/Q1 (Nonlinear): PAC/SAC outperform baselines under nonlinear mechanisms (TPR: 0.523/0.591 vs. 0.351 for R-PCMCI) with controlled FPR, closing the theory–experiment gap. Results should further improve with CMIknn in the final version.
W6/Q5 (Scalability): Graceful degradation up to N=10 with FPR < 0.05, and PAC/SAC running at roughly half the cost of PCMCI+.
W3/Q4 (Discretization): TPR degrades gradually under misspecification (~0.79 → ~0.67 at 40% flip rate) while FPR stays below 0.03, demonstrating practical robustness.
W2/Q2 (Climate): PAC/SAC produce sparser graphs (4 vs. 7 spurious edges) and recover proxy edges missed by Regime-PCMCI.

Together with the solid theory and practical PAC/SAC guidance, I consider this a well-rounded contribution. Updating from 4 (Weak Accept) to 5 (Accept).

**Key Questions For Authors:**

Nonlinear and non-Gaussian evaluation: Your sufficiency assumptions (Def. F.2) explicitly model nonlinear, threshold-based context effects. Have you evaluated PAC-PCMCI+ and SAC-PCMCI+ on synthetic data with nonlinear mechanisms? If so, how do results change? If not, this seems like a critical gap between the theory and experiments. Positive results here would strengthen the paper considerably.

Comparison with Regime-PCMCI on the climate dataset: Popescu et al. (2024) applied Regime-PCMCI to the same dataset you use. Can you provide a direct comparison showing which expert-hypothesized links each method recovers and which spurious links each produces? This would be the most convincing evidence for the practical value of handling endogenous contexts.

Guidance for choosing between PAC-PCMCI+ and SAC-PCMCI+: In practice, a user may not know whether persistence or sparsity better characterizes their system. Have you evaluated what happens when the "wrong" variant is applied (e.g., SAC-PCMCI+ on highly persistent data, or PAC-PCMCI+ when sparsity holds but persistence does not)? How robust are the methods to partial violations of their respective assumptions?

Sensitivity to discretization: Since the context must be discrete, how sensitive are results to the number of bins and the discretization method (quantiles, domain-based thresholds, etc.)? Even a small ablation on the synthetic data would address this concern.

Scalability: What are the computational costs (wall-clock time, number of CI tests) of PAC-PCMCI+ and SAC-PCMCI+ compared to standard PCMCI+ and M-PCMCI as the number of variables or sample size grows? You claim efficiency advantages but provide no runtime data.

**Limitations:**

The authors provide a thorough and honest discussion of limitations in Section 8, covering the reliance on acyclicity, persistence/sparsity, known discrete contexts, discretization effects, and reduced effective sample sizes. This is commendable. I would additionally suggest discussing the limitation to linear Gaussian evaluation and the absence of scalability analysis.

**Strengths And Weaknesses:**

S1. Important and well-motivated problem. The paper addresses a genuine gap: most causal discovery methods for non-stationary time series treat context as exogenous, yet many real-world systems exhibit endogenous regime dynamics. The dam management example (Figure 1) and the climate science motivation are compelling and make the problem concrete. Overall, the authors analyze a notable area that has received insufficient attention in the time-series causal discovery literature.

S2. Clean problem decomposition. The identification of persistence and sparsity as two complementary, testable properties that tame the context explosion problem is a strong conceptual contribution. The paper clearly explains why each property helps (persistence reduces the number of relevant context-specific graphs to K; sparsity allows conditioning on individual context parents rather than entire context windows) and how they lead to different algorithm variants with different assumption trade-offs.

S3. Rigorous theoretical treatment. The soundness proofs (Theorem 6.1) are carefully developed in the appendix, with separate Markov and faithfulness lemmas for both the persistent and sparse cases (Lemmas G.2, G.3, G.6, G.7). The paper is transparent about the differences in assumption strength between the two variants — persistence requires stronger acyclicity but weaker faithfulness, and vice versa. The extension from the i.i.d. framework of Günther et al. (2024) to time series is non-trivial, particularly the handling of variables outside the time window and lagged context effects.

S4. Principled handling of selection bias. The adaptive testing strategy — choosing pooled vs. context-specific data based on whether context variables appear in the conditioning set — elegantly addresses the selection bias problem that arises when naively masking on endogenous context values. This is well-explained and represents a practical advance even for settings with exogenous contexts (better sample efficiency).

S5. Honest and thorough discussion of limitations. The paper is refreshingly candid about what it can and cannot do: the reliance on known discrete contexts, the exponential hardness when both persistence and sparsity fail, the potential for faithfulness violations from discretization, and reduced effective sample sizes under persistent contexts.

Weaknesses:
W1. Limited synthetic evaluation scope. The synthetic experiments use only linear additive SCMs with Gaussian noise and binary contexts (n_contexts=2). This is a narrow setting that may not reflect the complexity of real-world systems. The paper's motivating examples (soil moisture with 3 regimes, financial volatility) involve more complex dynamics. The absence of nonlinear experiments is concerning because the sufficiency assumptions (Def. F.2) specifically address nonlinear mechanisms (threshold effects), yet the evaluation never tests them. It is unclear how well the methods perform when the data-generating mechanisms include nonlinear context-dependent effects, heterogeneous noise, or more than 2 context values.

W2. Weak real-world evaluation. The climate application (Sec. 7.2, App. I.5) is presented very briefly in the main text and reveals several spurious links (e.g., temperature → latent heat, latent heat → precipitation). The authors attribute these to confounding and mixed-type CI testing difficulties, but this undermines the practical demonstration. There is no systematic quantitative comparison against the expert-hypothesized ground truth from Popescu et al. (2024), and no comparison of the proposed methods against Regime-PCMCI on this same dataset — which is the very method that motivated the work. A table showing which expected links are recovered and which spurious links appear, for each method, would substantially strengthen this section.

W3. The assumption of known, discrete context variables is quite restrictive. While the paper acknowledges this, the practical bottleneck is significant. The paper states that context variables can come from "domain expertise or change-point detection," but provides no guidance on how discretization choices affect performance, nor any sensitivity analysis on the discretization threshold. Given that the paper explicitly discusses the proxy nature of discrete contexts (Sec. 3.1), an experiment varying discretization granularity (e.g., 2 vs. 3 vs. 5 bins) would be informative.

W4. Presentation of experimental results could be improved. Figure 2 is dense and somewhat hard to parse — 6 methods × 2 settings (adj. none / adj. C) × 2 panels (lagged/contemporaneous) × 3 metrics, all on the same figure with many overlapping lines. The key takeaways (FPR/TPR trade-offs) could be communicated more clearly. Additionally, the "adj. none" vs. "adj. C" distinction (whether context-system links are given as prior knowledge) is important but is explained only briefly in Sec. 7.1 and the caption, making it easy to miss that the dotted lines represent a partially-oracle setting.

W5. The relationship between the two algorithm variants could be better clarified. While the paper states that persistence and sparsity are "complementary" and "must not hold both at the same time" (Sec. 3.2.1), it is not entirely clear what happens when neither holds perfectly, or when both hold approximately. The paper does not provide guidance on how a practitioner should choose between PAC-PCMCI+ and SAC-PCMCI+ in a given application, beyond checking the two properties. A practical decision flowchart or empirical comparison under mixed conditions would help.

W6. Scalability considerations are not addressed. All experiments use D=7 system variables and n=1000. The computational complexity of the methods relative to standard PCMCI+ is not analyzed. While the paper claims computational efficiency relative to naive masking (M-PCMCI), runtime comparisons are absent.

W7. The "context explosion" framing, while valid, somewhat overstates novelty. The paper frames context explosion as a key challenge, but the proposed solutions (restrict to persistent windows, or exploit sparsity to condition on single context lags) are relatively straightforward once the properties are identified. This research's notable contribution concerns the formal integration of these ideas into the PCMCI+ framework with theoretical guarantees, rather than the conceptual insights themselves.

---

> ### Author Rebuttal · Authors · 2026-03-31
>
> We thank the reviewer for this thorough and thoughtful review, especially for highlighting the importance and real-world relevance of the problem, the clarity of our persistence–sparsity decomposition, the rigor of our theoretical analysis, and the effectiveness of our adaptive testing strategy.
>
> **Presentation**
> We agree that the figures, particularly Fig. 2, are dense. In the final version, we will split the presentation of the known context-system links setting from the full graph discovery setting, following the reviewer's suggestion.
>
> **Non-linear relationships and non-Gaussian noise**
> Our framework is agnostic to functional form and noise distribution; these affect the CI test rather than PAC/SAC-PCMCI+. We agree that empirical evidence in nonlinear settings would strengthen the paper and will add experiments using a kNN-based CI test (CMIknn).
>
> We also believe the question partly reflects a misunderstanding of our current data generation. The relationship between context and system variables in our experiments is already non-additive: we sample a base graph, select a (possibly endogenous) context variable, generate data for its ancestors, discretize the context via thresholding, and assign children of the context by modifying edges to obtain context-specific graphs while preserving acyclicity. This induces a multiplicative, non-additive effect of the context on the system, consistent with Def. F.2. We will expand the description in Sec. 7.1 and the appendix to make this explicit.
>
> **Climate example evaluation**
> We agree that a more systematic evaluation would strengthen the paper and will add a table comparing PAC-PCMCI+, SAC-PCMCI+, and Regime-PCMCI against the expert ground truth from Popescu et al. (2024), with a condensed version in the main paper.
>
> CD in presence of contexts is challenging, and typically requires expert knowledge. To obtain interpretable results with Regime-PCMCI prior work required substantial expert constraints via expert-identified links, raising concerns about confirmation bias. Our approach aims to leverage different expert-knowledge, about context-assignment.
>
> Our comparison shows that PAC-PCMCI+ and SAC-PCMCI+ recover key expert-identified links without using them as constraints, while producing substantially sparser graphs than unconstrained Regime-PCMCI. They consistently identify the main feedback structures (e.g., Stream $\rightarrow$ SW, SW $\rightarrow$ SH/LH, Stream $\rightarrow$ T2m), while avoiding many unsupported links. They also capture context-dependent proxy relationships such as TP $\rightarrow$ LH that align closely with the ground truth, even if direct links (e.g., to soil moisture) are not always recovered.
>
> In contrast, Regime-PCMCI yields much denser graphs with many unsupported links, likely due to assumption violations under endogenous contexts and its optimization-based regime discovery strategy, which partitions the data to maximize regime separation rather than to respect causal structure. This can introduce spurious links that reflect the partitioning rather than true dependencies.
>
> Overall, PAC/SAC-PCMCI+ achieve a more favorable balance between sensitivity and specificity, recovering both invariant and context-dependent structure without requiring comparable expert supervision. We will incorporate the table and expanded discussion in the revision.
>
> **Scalability in node count, sensitivity to discretization, efficiency and computational cost**
> We refer to the answer to Qtnv for further details.
>
> **Practical guidance on when to use PAC/SAC**
> We agree that guidance on choosing between PAC-PCMCI+ and SAC-PCMCI+ is important, particularly under partially satisfied assumptions.
>
> Persistence can be interpreted as effective sample size per context and estimated via context frequencies. Sparsity can be assessed from the union graph across contexts.
>
> To make the trade-offs explicit, we will add a diagram that maps performance across two axes: degree of persistence and degree of sparsity violation, synthesizing theoretical and empirical findings from our work (see response to gtfK and Qtnv) and indicating the preferable method in each regime.
>
> Our results suggest the following guidance. When sparsity holds, SAC is generally preferable due to its finite-sample efficiency, even under persistence. PAC remains robust when sparsity is violated, particularly with persistent contexts and sufficient data.
>
> An important point is that persistence primarily affects finite-sample performance rather than asymptotic correctness, and PAC can be applied to any subset of context combinations that occur sufficiently often, providing flexibility in sparse-data settings. Moreover, contexts can be redefined, e.g., by grouping variables (or their values), to induce sparsity or persistence, and mitigate context explosion.
>
> Finally, we acknowledge that extending the framework to handle sparsity violations is an important direction for future work.

---

> > ### Author Rebuttal · Reviewer_AsxX · 2026-04-02
> >
> > I read the rebuttal carefully and appreciate the detailed responses. Several points are clarified conceptually, but key evidence is deferred to the revision, which limits how much I can update my assessment.
> >
> > Concerns addressed:
> >
> > W5/Q3 (PAC vs. SAC guidance): The practical guidance is clear and useful—persistence as effective sample size, sparsity from the union graph, and the recommendation to prefer SAC when sparsity holds. The proposed persistence × sparsity diagram would be a valuable addition.
> >
> > W4 (Figure 2): Splitting known-links vs. full-discovery settings should improve clarity.
> >
> > W1/Q1 (Nonlinear evaluation — partial): It is helpful to clarify that context–system relations are already non-additive. However, system–system links remain linear Gaussian, and the promised nonlinear (e.g., CMIknn) results are not yet shown. It would help to confirm whether these will include nonlinear mechanisms and non-Gaussian noise, or only changes to the CI test.
> > Concerns still open:
> >
> > W2/Q2 (Climate comparison): The summary is encouraging (sparser graphs, key links recovered), but no table is provided. Given that this is the main real-world validation, even a compact comparison (TP/FP/missed links vs. ground truth) would be important.
> >
> > W3/Q4 (Discretization sensitivity): The response refers to another thread. A brief summary of sensitivity to the number of bins would be helpful.
> >
> > W6/Q5 (Scalability): Also deferred. Some indication of how runtime scales with the number of variables (beyond D=7) would strengthen the paper.
> >
> > The rebuttal reinforces my view that this is a solid theoretical contribution with well-motivated algorithms. However, the empirical support has not substantially expanded—most additions are promised for the final version. I therefore maintain my score of 4 (Weak Accept), expecting the revision to include nonlinear experiments, the climate comparison, and sensitivity/scalability analyses.

---

> > > ### Author Response · Authors · 2026-04-04
> > >
> > > We report preliminary results (10 runs) for Q1, Q4, Q5, final version will use 100 runs. Errors reported for Q1 only due to space limitations.
> > >
> > > **Q1** We allow linear and non-linear system-system links (drawn randomly), and use Robust Partial Correlation CIT (see Tigramite) due to time reasons. For the final version, we will run these experiments with CMIknn, which should improve TPR considerably while maintaining FPR control.
> > >
> > > *Averaged per-regime*
> > >
> > > | Method | TPR | FPR |
> > > |---|---|---|
> > > | PAC | 0.523 ± 0.027 | 0.032 ± 0.008 |
> > > | SAC | 0.591 ± 0.036 | 0.045 ± 0.012 |
> > > | R-PCMCI | 0.351 ± 0.023 | 0.026 ± 0.007 |
> > > | M-PCMCI | 0.451 ± 0.030 | 0.028 ± 0.005 |
> > > | Baseline | 0.387 ± 0.027 | 0.019 ± 0.007 |
> > >
> > > *Union*
> > >
> > > | Method | TPR | FPR |
> > > |---|---|---|
> > > | PCMCI+| 0.628 ± 0.042 | 0.066 ± 0.016 |
> > > | PAC | 0.554 ± 0.028 | 0.040 ± 0.010 |
> > > | SAC | 0.589 ± 0.033 | 0.051 ± 0.013 |
> > > | R-PCMCI | 0.481 ± 0.037 | 0.042 ± 0.012 |
> > > | M-PCMCI | 0.519 ± 0.034 | 0.043 ± 0.009 |
> > > | Baseline | 0.469 ± 0.029 | 0.025 ± 0.009 |
> > >
> > >
> > > **Q5** Union graph results. FPR increases and TPR decreases with N, but within 0.1 points and FPR < 0.05.
> > >
> > > *FPR*
> > > | Method | N=5 | N=8 | N=10 |
> > > |---|---|---|---|
> > > | Baseline | 0.005 | 0.022 | 0.022 |
> > > | PCMCI+ | 0.039 | 0.069 | 0.050 |
> > > | PAC | 0.019 | 0.035 | 0.031 |
> > > | SAC | 0.041 | 0.039 | 0.039 |
> > > | M-PCMCI | 0.030 | 0.045 | 0.048 |
> > > | R-PCMCI| 0.019 | 0.034 | 0.035 |
> > >
> > > *TPR*
> > >
> > > | Method | N=5 | N=8 | N=10 |
> > > |---|---|---|---|
> > > | Baseline | 0.505 | 0.516 | 0.429 |
> > > | PCMCI+ | 0.670 | 0.627 | 0.611 |
> > > | PAC | 0.604 | 0.558 | 0.529 |
> > > | SAC | 0.681 | 0.595 | 0.529|
> > > | M-PCMCI | 0.505 | 0.493 | 0.439 |
> > > | R-PCMCI | 0.461 | 0.483 | 0.402 |
> > >
> > > The computation times show a clear scaling trend: PAC and SAC scale roughly as half the runtime of PCMCI+ across all N, while masking methods (M-PCMCI, R-PCMCI) are faster but more prone to errors. We will include larger N experiments (N=15, N=20) in the final version to make this scaling behaviour explicit.
> > >
> > > *Computation time*
> > >
> > > | Method | N=5 | N=8 | N=10 |
> > > |---|---|---|---|
> > > | PCMCI+ | 0.528 | 5.629 | 12.402 |
> > > | PAC | 0.569 | 2.244892 | 5.865 |
> > > | SAC | 0.851 | 2.441595 | 6.274 |
> > > | M-PCMCI | 0.124 | 0.589 | 1.570 |
> > > | R-PCMCI| 0.138 | 0.368 | 0.787 |
> > >
> > >
> > > **Q4: Misspecified by flip**
> > >
> > > *Regime-averaged TPR*
> > >
> > > | $\Delta$ | PAC | SAC |R-PCMCI | M-PCMCI | Baseline |
> > > |---|---|---|---|---|---|
> > > | 0.05 | 0.789 | 0.834 | 0.518 | 0.638 | 0.590 |
> > > | 0.10 | 0.754 | 0.802 | 0.515 | 0.611 | 0.611 |
> > > | 0.20 | 0.719 | 0.811 | 0.531 | 0.673 | 0.594 |
> > > | 0.30 | 0.635 | 0.689 | 0.462 | 0.580 | 0.502 |
> > > | 0.40 | 0.669 | 0.693 | 0.445 | 0.603 | 0.469 |
> > >
> > > *Regime-averaged FPR*
> > >
> > > | $\Delta$ | PAC | SAC | R-PCMCI | M-PCMCI | Baseline |
> > > |---|---|---|---|---|---|
> > > | 0.05 | 0.008 | 0.015 | 0.009 | 0.010 | 0.004 |
> > > | 0.10 | 0.015 | 0.033 | 0.006 | 0.022 | 0.007 |
> > > | 0.20 | 0.011 | 0.020 | 0.005 | 0.012 | 0.006 |
> > > | 0.30 | 0.018 | 0.027 | 0.009 | 0.023 | 0.006 |
> > > | 0.40 | 0.017 | 0.025 | 0.008 | 0.019 | 0.006 |
> > >
> > > *Union TPR*
> > >
> > > | $\Delta$ | PCMCI+ | PAC | SAC | R-PCMCI | M-PCMCI | Basline |
> > > |---|---|---|---|---|---|---|
> > > | 0.05 | 0.841 | 0.788 | 0.797 | 0.555 | 0.599 | 0.660 |
> > > | 0.10 | 0.850 | 0.791 | 0.786 | 0.545 | 0.641 | 0.652 |
> > > | 0.20 | 0.842 | 0.810 | 0.805 | 0.636 | 0.626 | 0.694 |
> > > | 0.30 | 0.808 | 0.782 | 0.766 | 0.629 | 0.661 | 0.677 |
> > > | 0.40 | 0.819 | 0.729 | 0.729 | 0.562 | 0.643 | 0.662 |
> > >
> > > *Union FPR*
> > >
> > > | $\Delta$ | PCMCI+ | PAC | SAC |R-PCMCI | M-PCMCI | Baseline |
> > > |---|---|---|---|---|---|---|
> > > | 0.05 | 0.026 | 0.008 | 0.017| 0.014 | 0.024 | 0.006 |
> > > | 0.10 | 0.052 | 0.025 | 0.035 | 0.025 | 0.027 | 0.020 |
> > > | 0.20 | 0.019 | 0.010 | 0.009| 0.016 | 0.010 | 0.002 |
> > > | 0.30 | 0.052 | 0.023 | 0.020 | 0.034 | 0.032 | 0.026 |
> > > | 0.40 | 0.028 | 0.025 | 0.025 | 0.027 | 0.030 | 0.014 |
> > >
> > >
> > > **Q4 - Misspecified by moving threshold** Results for the experiment moving the threshold by $\Delta$×(max⁡(indicator)−threshold GT) shows similar behavior: PAC and SAC algorithms are more robust to changes than other algorithms for regime misspecifications, as they have the smallest decrease in TPR, while FPR is still being controlled (results not reported due to space limitations).
> > >
> > > **Q4 - Sensitivity to bins** Bin sensitivity is equivalent to varying sample size: the number of bins determines the effective sample size per regime. This trade-off is captured by our sample size experiments, making a separate bins experiment redundant. We will run it if the reviewer considers it necessary.
> > >
> > > **Q2** We provide a condensed comparison table (due to space constraints), a link-wise comparison table has also been added to the appendix of the paper:
> > >
> > > | Ground Truth | PAC| SAC| Regime-PCMCI |
> > > |---|---|---|---|
> > > | Ground truth not context-dependent edges | Mostly correct | Mostly correct | Correct |
> > > | Context-dependent edge LH - SM | Missing | Missing | Present in all regimes, no reversal |
> > > | Proxy edge LH - TP | Present in all regimes (lag changes in moist) | Present in all regimes (lag changes and reversed in moist) | No link |
> > > | Spurious edges in summary graph | 4 | 4 | 7 |

---

### Official Review · Reviewer_Qtnv · 2026-03-11

**Soundness:** 3
**Presentation:** 2
**Significance:** 2
**Originality:** 3
**Overall Recommendation:** 5
**Confidence:** 2

**Summary:**

In this paper, the authors study the setting of causal discovery from non-stationary time-series data while treating the context variable as an endogenous variable. The context variable is regarded as a discrete proxy of the underlying continuous context variable. The authors propose two modifications to the PCMCI+ algorithm which attains good recovery performance in the experiments showcased in the paper. Under some regularity constraints on the context variable (persistence and sparsity), the authors show that the oracle version of their method recovers true graph.

**Compliance With Llm Reviewing Policy:**

Affirmed.

**Final Justification:**

The authors have addressed my concerns during the rebuttal, I have increased my score from 4 to 5.

**Key Questions For Authors:**

Q1. How does the proposed method perform under misspecification? What if the persistence or sparsity requirement is relaxed? How would the performance be affected if there is a cycle between the context variable and other endogenous variables?

Q2. How does the proposed method scale with the number of nodes in the graph?

Q3. How does the proposed method compare with the baselines in terms of computational complexity?

Minor comments:
- Typo in line 352, "precesure" instead of "procedure".
- In theorem 6.1, the phrase "Asm 1 or 2" is repeated twice.

**Limitations:**

Yes, the limitations are discussion, although experimental evaluation under misspecified setting would be helpful.

**Strengths And Weaknesses:**

_Strengths_:

1. The paper tackles a challenging problem of performing causal discovery from time-series when the context variable in endogenous. The authors motivate the setup well and is of high importance in applying causal methods to real-world settings.

_Weaknesses_:

1. The paper is quite hard to read, while the problem setup is motivated well, the section containing the technical description of the proposed method is not very easy to follow.
2. Lack of performance analysis under misspecified regime.

---

> ### Author Rebuttal · Authors · 2026-03-31
>
> We thank the reviewer for recognizing the importance and difficulty of causal discovery with endogenous context variables, as well as the strong motivation and real-world relevance of our setup. We thank you as well for spotting the typos, we will correct these in the paper.
>
> **Presentation** We agree that Sec. 5 is currently dense, partly due to length constraints. In the final version, we will restructure it as follows: we will first introduce PCMCI+, then adaptive testing, and then present and clearly delineate the two algorithm variants (PAC and SAC). Verbal explanations will be added for the tests used in each variant to improve readability. We will also incorporate the practitioner guide discussed in our response to Reviewer AssX. We welcome any further suggestions on improving clarity.
>
> **Property violations** For violations of persistence and sparsity, we refer the reviewer to our response to Reviewer gtfK.
>
> Regarding cycles: as stated in Sec. F, PAC-PCMCI+ does not permit cycles involving the context indicator in the summary graph. SAC-PCMCI+ relaxes this to the window level, allowing time-resolved cycles and feedbacks.
>
> The acyclicity assumption between context and system variables is required for our methods to reliably distinguish ancestors from descendants of context variables, an assumption that is foundational for both persistence and sparsity regimes (Sec. 6 and F.2.2). In causal modeling, (contemporaneous) cycles are usually interpreted by finding (unique) solutions, if those exist; this makes definition and interpretation of contextual links difficult. In the presence of larger cycles among system variables, our methods recover all links between the context variable and the system variables involved in the cycle, consistent with the behavior described in Guenther et al. (2024). More generally, cycles in SCMs lead to the discovery of the so-called acyclification of the true graph (Bongers et al., 2021), which bounds what any constraint-based method can recover in this setting. We therefore treat sensitivity to cycle violations as a known limitation and a direction for future work.
>
> **Scaling in node count**
> Experiments are currently running. Preliminary results show a decrease in performance with increasing node count, most notably in TPR, though the magnitude is modest: going from 8 to 10 nodes yields a drop of approximately 0.02 points. The final version will include a scaling experiment extending up to 20 nodes.
>
>
> **Computational complexity**
> We first want to correct a potentially misleading statement in the paper: the claim that our approach requires fewer CI tests than naive masking holds only in the oracle case. In practice, the situation is more nuanced, and we will revise the manuscript accordingly.
>
> Comparing computational complexity across methods is difficult due to differing sample efficiencies and testing strategies. In constraint-based causal discovery, the total number of CI tests depends on early edge deletions: incorrect early deletions (e.g., due to low sample size) can reduce test count and runtime without improving accuracy. Naive masking operates on smaller per-context samples and often deletes edges early, sometimes incorrectly, which lowers its test count. This has been observed in prior work for the i.i.d. case (Guenther et al., 2024), where M-PC shows lower runtime than the adaptive testing variant precisely for this reason.
>
> Now we analyze the oracle case. PAC/SAC explicitly test context-system links, which increases the number of CI tests relative to masking. However, by removing (i) spurious dependencies induced by selection bias upstream of context variables and (ii) context-irrelevant edges, PAC/SAC can produce sparser intermediate graphs (see example below), reducing the number of tests in later stages.
> To illustrate: given system variables X,Y,Z,W and context variable C, where C influences the presence of the edge $Z\rightarrow W$ and induces a spurious (selection-bias) association between X and Y under masking. A union graph retains all edges, masking cannot remove the spurious X-Y association. PAC/SAC test context-system relations, remove the selection-bias-induced edge, and thus recover a sparser intermediate graph, which reduces subsequent test count.
>
> Scaling with node count is mostly determined by the underlying causal discovery algorithm; in the oracle case, this typically means worst-case scaling as $\mathcal{O}(k^d)$ for $k$ nodes and bounded graph-degree $d$. Scaling with sample count depends only on the choice of CIT, and scaling with context count, i.e. context-explosion, is discussed in the main text.
>
> The primary advantage of PAC/SAC lies in improved statistical sample efficiency, not guaranteed runtime reduction. We will add this discussion and a runtime analysis to the final version to make this concrete.

---

> > ### Author Rebuttal · Reviewer_Qtnv · 2026-04-03
> >
> > Thank you for the rebuttal response. I don't have any other questions.

---

### Official Review · Reviewer_gtfK · 2026-03-12

**Soundness:** 3
**Presentation:** 3
**Significance:** 3
**Originality:** 3
**Overall Recommendation:** 5
**Confidence:** 3

**Summary:**

This paper studies causal discovery in time series with endogenous context variables, where contexts both influence and are influenced by the system. The authors model contexts as discrete time-series variables with lagged dependencies, which introduces challenges such as an exponential number of possible context configurations and potential selection bias. To address this, they identify two structural properties (context persistence and sparse context dependence) that make the problem tractable. Based on these assumptions, they propose two extensions of PCMCI+, called PAC-PCMCI+ and SAC-PCMCI+, which use adaptive conditional independence testing to recover context-specific causal graphs. The paper provides theoretical guarantees and empirical evaluations on synthetic data and a climate application, showing improved accuracy and sample efficiency compared to several PCMCI-based baselines.

**Compliance With Llm Reviewing Policy:**

Affirmed.

**Final Justification:**

The authors have addressed the concerns in the rebuttal. I am satisfied and increasing my score from 4 to 5.

**Key Questions For Authors:**

1. The proposed methods rely on either persistence or sparse context dependence. How does the method behave when these assumptions are partially violated? Providing experiments or analysis in such scenarios would help understand the robustness of the approach and could strengthen the evaluation.

2. The framework assumes that discrete context variables are known. In many real-world settings, contexts may need to be inferred. How sensitive are PAC-PCMCI+ and SAC-PCMCI+ to errors in context assignment?

3. The experiments mainly compare against PCMCI-based variants. Could the authors comment on how their methods relate to or compare with other approaches for nonstationary causal discovery?

4. The paper includes a climate science example. Do the authors have additional real-world case studies or qualitative validations (ex: domain expert assessment) that support the discovered context-specific causal relationships?

**Limitations:**

Yes.

**Strengths And Weaknesses:**

Strengths

1. Tackles an important and realistic problem which is causal discovery in time series with endogenous context variables, relaxing the common exogeneity assumption.
2. Proposes two practical extensions of PCMCI+ (PAC-PCMCI+ and SAC-PCMCI+) that mitigate selection bias via adaptive conditional independence testing.
3. Provides theoretical guarantees for recovering context-specific causal graphs under clear assumptions.
4. Well designed synthetic experiments with relevant baselines show improvements in false positive rates.
5. Demonstrates practical relevance through a real-world climate science application.

Weaknesses

1. The framework relies on assumptions such as persistence or sparse context dependence, and behavior when these assumptions fail is not thoroughly studied.
2. Assumes known discrete context variables, which may not always be available in practice.
3. Empirical evaluation is somewhat limited, with only one real-world dataset.
4. Comparisons are mainly with PCMCI based baselines, with limited evaluation against broader nonstationary causal discovery methods.
5. The theoretical guarantees depend on additional assumptions (context sufficiency and acyclicity) that may restrict applicability in some systems.

---

> ### Author Rebuttal · Authors · 2026-03-31
>
> Thank you for your constructive review. We especially appreciate your recognition of the importance of our setting, the practicality of our proposed PCMCI+ extensions, our theoretical guarantees, the strength of our experiments, and the relevance of our real-world application.
>
> **Sensitivity to violations** We thank the reviewer for raising this point. Our experiments already address sensitivity to persistence via the context auto-lag parameter (see Fig. 2), which acts as a proxy for persistence. We therefore consider this dimension covered. Regarding sparsity: our current setup uses a single context variable per graph, thus implicitly enforcing a sparse regime. We agree that explicitly varying sparsity would strengthen the evaluation, and we will add experiments with multiple context variables per graph, where one variable has multiple context parents, to the final version.
>
> **Sensitivity to discretization** Discretization affects results in two distinct ways. First, if contexts are split at the correct threshold but the split is too fine-grained (too many values taken by the context), the number of samples available per context decreases, reducing per-context sample size and implicitly hurting performance. Sensitivity to sample size is already examined in Fig. 11.
> Second, if the threshold is misspecified such that samples from different ground-truth contexts are grouped together, context impurity arises: the algorithm operates on a mixture of regimes instead of individual ones. We did not study this in the original paper, and suggest to add two experiments to the appendix.
> In the first, we shift the context assignment threshold by $\Delta > 0$ from the true threshold after data generation. This experiment captures the effects of context misalignment and reduced per-context sample size. In the second, we isolate the effect of context impurity by keeping the 50/50 sample split fixed and randomly flipping a percentage of context assignments.
>
> Preliminary results show that TPR drops by approximately 0.1–0.15 points as the fraction of misspecified samples increases to 40\%, while FPR increases but remains below 0.05. This confirms that sample size reduction and context impurity do degrade performance, as expected, but the algorithms remain reliable even under substantial misspecification. In the extreme case where enough samples from multiple regimes are mixed, the result approximates the union graph, which yields higher TPR alongside higher FPR. However, the output remains informative as it recovers the union graph of the true structure.
>
> **Comparison against other variants** We position our work in Sec. 1 as the first constraint-based approach for endogenous context-specific causal discovery on time series data. Our adaptive testing procedure can be combined with other constraint-based methods to produce context-specific output, and the goal of our experimental setup was to highlight the advantages of this approach over other strategies such as simple masking.
> We consider a comparison against methods that recover a union graph, such as FCI, as less relevant, because the comparison would mainly reflect performance differences between PCMCI+ and FCI, rather than evaluating our contribution. Our method is an efficient framework for discovering context-specific graphs that operates as a meta-method on top of existing constraint-based approaches.
>
> To our knowledge, the only directly comparable work is Rodas et al. (2024), which takes an optimization-based approach and models context dependence via Markov switching models. At the time of submission, no code was available, and we considered reimplementation out of scope. The code has since been made public. Nevertheless, given the method's higher number of hyperparameters and different overall approach, we prefer to keep the experimental setting focused on the core contribution: improvement through adaptive testing and exploitation of system properties, and consider a direct comparison for future work. Should the reviewers consider such a comparison necessary for acceptance, we propose to add comparisons to other constraint-based methods (e.g., Huang et al., Mameche et al.).
>
>
> **Additional real-world cases** For validation of the existing real-world example (soil moisture–atmosphere feedbacks), we refer to our response to Reviewer AssX. Regarding the request for an additional example, we agree that this would strengthen the paper. We plan to add a fully worked illustrative example from economics to the final version. Endogenous Markov switching models have been widely used in economics to model markets under shifting policies and structural shocks, making them a natural fit for our framework. We will generate time series data from such a model, demonstrate how to assess the key properties of persistence and sparsity, and apply our algorithms. The aim is to provide practitioners with a concrete walkthrough that guides application of the method to real settings.

---

> > ### Author Rebuttal · Reviewer_gtfK · 2026-04-03
> >
> > Thank you for the rebuttal. The authors have mostly addressed my questions. I do share some of the concerns raised by Reviewer AsxX. I will decide whether to maintain or increase the score based on the Authors response to those concerns raised.

---

### Decision · Program_Chairs · 2026-04-30

**Decision:**

Accept (regular)

**Comment:**

This paper studies causal discovery in time series with endogenous context variables, a setting motivated by real-world systems in which context and system variables can influence each other. Reviewers agreed that the problem is well motivated and that the paper makes a meaningful technical contribution by identifying two tractable regimes, persistence and sparse context dependence, and by deriving PAC-PCMCI+ and SAC-PCMCI+, two PCMCI+-based procedures with soundness guarantees under explicit assumptions.

The initial concerns centered on empirical breadth and presentation. The technical section was viewed as dense, the theory relies on assumptions such as known discrete contexts and context sufficiency and acyclicity, and the experiments were initially limited in scope, especially with respect to misspecification, nonlinear settings, scalability and runtime, and the climate case study. The authors addressed these concerns in the rebuttal and follow-up discussion. In particular, the additional evidence reported during discussion, including nonlinear experiments, discretization and misspecification robustness, scaling and runtime results, and a clearer comparison on the climate application, materially strengthens the empirical case and directly addresses the main criticisms. Importantly, the reviewers indicated that these additions addressed the remaining concerns, and they raised their scores after the rebuttal.

The paper is technically sound, addresses a key gap in the field, and offers a useful framework for context-specific causal discovery in time series. I therefore recommend acceptance.